# Disentangled Representation Learning in Non-Markovian Causal Systems

**Adam Li**[*] and **Yushu Pan**[*] and **Elias Bareinboim**

Causal Artificial Intelligence Lab

Columbia University

`{adam.li, yushupan, eb}@cs.columbia.edu`

## Abstract

Considering various data modalities, such as images, videos, and text, humans perform causal reasoning using high-level causal variables, as opposed to operating at the low, pixel level from which the data comes. In practice, most causal reasoning methods assume that the data is described as granular as the underlying causal generative factors, which is often violated in various AI tasks. This mismatch translates into a lack of guarantees in various tasks such as generative modeling, decision-making, fairness, and generalizability, to cite a few. In this paper, we acknowledge this issue and study the problem of causal disentangled representation learning from a combination of data gathered from various heterogeneous domains and assumptions in the form of a latent causal graph. To the best of our knowledge, the proposed work is the first to consider i) non-Markovian causal settings, where there may be unobserved confounding, ii) arbitrary distributions that arise from multiple domains, and iii) a relaxed version of disentanglement. Specifically, we introduce graphical criteria that allow for disentanglement under various conditions. Building on these results, we develop an algorithm that returns a causal disentanglement map, highlighting which latent variables can be disentangled given the combination of data and assumptions. The theory is corroborated by experiments.

## 1 Introduction

Causality is fundamental throughout various aspects of human cognition, including understanding, planning, decision-making. The ability to perform causal reasoning is considered one of the hallmarks of human intelligence [1–3]. In the context of AI, the capability of reasoning with cause-and-effect relationships plays a critical role in challenges of explainability, fairness, decision-making, robustness, and generalizability. One key assumption of most methods currently available in the causal literature is that the set of (endogenous) variables is at the right level of granularity. However, this is not the case in many AI applications, where various modalities, such as images, and text, come into play [4]. For example, images of a park scene capture objects as causal variables, not the pixels themselves. AI must disentangle these latent causal variables to represent the true relationships in the image. Faithfully representing this latent structure impacts downstream AI tasks like image generation and few-shot learning.

In machine learning, the representation learning literature is concerned with finding useful representations from data [5]. One important line of work traces back to linear ICA (independent component analysis) [6], where one attempts to disentangle latent variables assuming a linear mixing function. The literature has also considered settings where the mixing function is nonlinear [7, 8]. It has been understood that nonlinear-ICA is, in general, not identifiable (ID) given only observational data [9].

---

[*]These authors contributed equally to this work, and the author names are listed in alphabetical order.

38th Conference on Neural Information Processing Systems (NeurIPS 2024).

| Work | Input | | | | | Output |
|---|---|---|---|---|---|---|
| | Assumptions | | Data | | | Identifiability Goal |
| | Non-Markovian | Non-parametric | Interventions | Multiple Domains | Distr. Reqs. | |
| [6, 11–14] | ✗ | ✗ | ✓ | ✗ | 1 per node | Scaling, Mixture or Affine Transformation |
| [7, 8, 10, 15, 16] | ✗ | ✗ | ✗/✓ | ✗/✓ | $2\|\mathbf{V}\|+1$ | Scaling |
| [17, 18] | ✗ | ✓ | ✓ | ✗ | 1 per node | Scaling |
| [19, 20] | ✗ | ✓ | ✗/✓ | ✗/✓ | 1 per node | Scaling |
| [21] | ✗ | ✓ | ✓ | ✗ | 1 per node | Scaling or Ancestral Mixture |
| [22] | ✗ | ✓ | ✗ | ✓ | $2\|\mathbf{V}\|+\|M_G\|+1$ | Scaling or Mixture |
| [23] | ✗ | ✗ | ✗/✓ | ✗/✓ | TBD | Scaling & Affine Transformation |
| [24] | ✗ | ✓ | ✗/✓ | ✗/✓ | TBD | Functional Dependency Map[1] |
| **This work** | ✓ | ✓ | ✓ | ✓ | General | Causal Disentanglement Map |

Table 1: A non-exhaustive list of identifiability results given knowledge of the latent graph.

Different routes have been taken to circumvent such impossibilities. For instance, one might assume parametric families (e.g., exponential), and auxiliary variables as input, which can be thought of as non-stationary times-series that may lead to certain invariances that can be exploited [7, 8, 10].

Interestingly, the machinery developed in this context can be applied to causal settings with multimodal data, where there is a mismatch between the causal variables and the granularity at which they are represented in the data. The key observation that links these two worlds is that an underlying causal system generates the data at such granularity (images, texts). Acknowledging this connection leads to various possibilities regarding learning, or disentangling the causal variables from data, similar to the initial ICA-like literature. First, the assumption that the features underlying a signal are independent needs to be relaxed since it is arguably too stringent, *a priori* ruling out almost any interesting causal system. So, we should consider different assumptions regarding the structure of the underlying generative model. One initial relaxation is that this model is Markovian, where the features need not be independent, and causal relationships are allowed across features. In the context

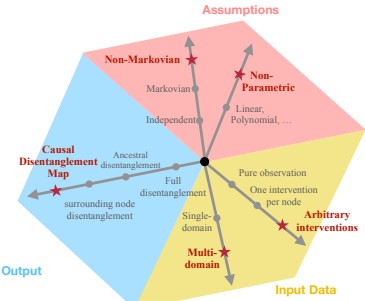

Figure 1: Dimensions of identifiability in causal disentanglement representation learning.

of computer vision, for example, one might assume a specific structure on the latent variables where the style and content of the images are separated and augmented data is leveraged to disentangle these two components [18]. Generalizing this idea to more relaxed causal settings, one can show ID up to certain indeterminancies given observational across multiple domains, or interventional data [21, 22]. Another approach allows for certain parametric mixing functions, which could lead to new ID results [11, 14]. These results have been applied and advanced across various downstream tasks [25–31].

Considering this background, we study three axes within the different types of input and expected outputs of the causal disentanglement representation learning task, which is summarized in Table 1 and Fig. 1. The input can be partitioned into qualitative and quantitative components. Qualitatively, we consider different **assumptions** about the underlying generative processes, including non-parametric models, as well as linear or Gaussian ones. We also account for systems with richer causal topologies than ICA (independent features) and generalize the Markovian setting. Notably, we do not rule out *a priori* the existence of unobserved confounding among features, a pervasive challenge in causal inference in empirical sciences. Quantitatively, we consider **data** gathered from arbitrary combinations of interventions and domains. Recent literature on this distinction acknowledges key differences [32–37], whereas prior literature often assumes data comes from different interventions in the same domain or from various (observational) distributions from different domains. In fact, it is feasible that data spawns various interventions and domains in a less well-structured manner (App. A.3). In terms of the expected **output**, similar to [24], we will consider both full disentanglements as well as a more relaxed type of disentanglement, known as the causal disentanglement map.

For concreteness, consider a hypothetical latent graph depicted in Fig. 2 in the context of epilepsy research [38–46]. In terms of **assumptions**, hospitals in different countries $\Pi^i$ and $\Pi^j$ will differ in the amount of sleep ($V_1$) patients get (represented by the S-node $S^{i,j} \to V_1$). Now suppose sleep ($V_1$) affects the efficacy of the drug treatment ($V_2$), and the drug helps epilepsy patients control their seizures ($V_3$). The quality of sleep and the type of drug treatment are confounded by socioeconomic factors ($V_1 \leftarrow\text{-}\text{-}\text{-}\rightarrow V_2$). Clinicians are then given electroencephalogram (EEG) **data** from each

---

[1]We recently were made aware of the work in [24], where their definition of a functional dependency map is what we define as a causal disentanglement map. However, the disentanglement map that they can achieve is different from ours as discussed in Section F.

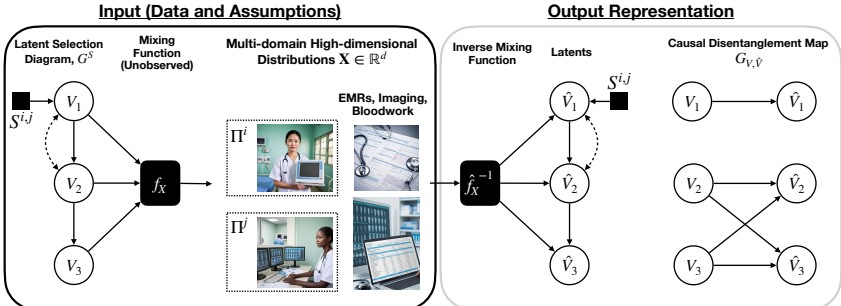

Figure 2: Data generating model and the goal of learning disentangled causal latent representations.

hospital where they know different drug treatments were administered. The EEG $\mathbf{X}$ is a nonlinear (nonparametric) transformation of latent $\mathbf{V} = \{V_1, V_2, V_3\}$ via $f_X$. Their goal is to generate realistic EEG data to understand how different drugs affect EEG patterns. This requires a general **output** representation that disentangles sleep from drug as it is understood that sleep affects EEG [47]. One could leverage state-of-the-art generative modeling techniques and train a self-supervised learning model to learn a representation of the EEG that they then perturb to generate new instances of EEG [48–50]. However, there are no guarantees that the representation, or interventions in the latent space will generate realistic EEG. In this case, drug and sleep might remain entangled in the learned representation, which is potentially harmful, since it may lead to unrealistic EEG data that contains visual differences due to sleep rather than drug. More formally, given an input set of distributions and knowledge of the latent variable causal structure, the goal is to learn the inverse of the mixing function $\widehat{f}_X^{-1}$ and a representation $\widehat{\mathbf{V}} = \{\widehat{V_1}, \widehat{V_2}, \widehat{V_3}\}$, where $V_2$ is disentangled from $V_1$ [2].

In this paper, we develop graphical and algorithmic machinery to determine whether (and how) causal representations can be disentangled from heterogeneous data and assumptions about the underlying causal system, which might help improve various downstream tasks. Our contributions are: [3]:

1. **Graphical criteria for determining the disentangleability of causal factors.** We formalize a general version of the causal representation learning problem and develop methods to determine if a pair of (user-chosen) variables are disentangled in a non-Markovian setting with arbitrary distributions from multiple heterogeneous domains (Props. 3,4, and 5)[4].
2. **An algorithm to learn the causal disentanglement map.** Leveraging these new conditions, we develop an algorithmic called **CRID**, which systematically determines whether two sets of latent variables are disentangleable given their selection diagram and a collection of intervention targets (Thm. 1). The theoretical findings are corroborated with simulations.

All supplementary material (including proofs) is provided in the full technical report.

**Preliminaries.** We introduce basic definitions used throughout the paper. Uppercase letters ($X$) represent random variables, lowercase letters ($x$) signify assignments, and bold letters ($\mathbf{X}$) indicate sets. For a set $\mathbf{X}$, $|\mathbf{X}|$ denotes its dimension. Denote $P(\mathbf{X})$ as a probability distribution over $\mathbf{X}$ and $p(\mathbf{x})$ as its density function. The basic semantic framework of our analysis rests on structural causal models (SCMs) [1, Ch. 7]. An SCM is a 4-tuple $\langle \mathbf{U}, \mathbf{V}, \mathcal{F}, P(\mathbf{U}) \rangle$, where (1) $\mathbf{U}$ is a set of background variables, also called exogenous variables, that are determined by factors outside the model; (2) $\mathbf{V} = \{V_1, V_2, \ldots, V_d\}$ is the set of endogenous variables that are determined by other variables in the model; (3) $\mathcal{F}$ is the set of functions $\{f_{V_1}, f_{V_2} \ldots, f_{V_d}\}$ mapping $\mathbf{U}_{V_j} \cup \mathbf{Pa}_{V_j}$ to $V_j$, where $\mathbf{U}_{V_j} \subseteq \mathbf{U}$ and $\mathbf{Pa}_{V_j} \subseteq \mathbf{V} \backslash V_j$; (4) $P(\mathbf{U})$ is a probability function over the domain of $\mathbf{U}$.

Each SCM induces a causal diagram $G$, which is a directed acyclic graph where every $V_j$ is a vertex. There is a directed arrow from $V_j$ to $V_k$ if $V_j \in \mathbf{Pa}_{V_k}$. There is a bidirected arrow between $V_j$ and $V_k$ if $\mathbf{U}_{V_j}$ and $\mathbf{U}_{V_k}$ are not independent [3]. Variables $\mathbf{V}$ can be partitioned into subsets called *c-components* [57]. The c-component of $X$, denoted as $\mathbf{C}(X)$, is a set of variables connected to $X$ by bidirected paths. The c-component of a set $\mathbf{X}$, denoted as $\mathbf{C}(\mathbf{X})$, is defined as the union of

---

[2]We separate the tasks of disentanglement and structural learning, and consider the latent causal graph as input of our task. Still, there are works in the literature that consider both tasks simultaneously [13, 22, 51–56].

[3]We refer the readers to our full technical report for a more detailed treatment of the problem setting.

[4]All proofs are provided in Appendix C.

the c-component of every $X \in \mathbf{X}$. We will use $\mathbf{Pa}(X)$ or $\mathbf{Pa}_X$ to denote parents of $X$ in $G$. Let $\overline{\mathbf{Pa}}(X) = \mathbf{Pa}(X) \cup X$, which includes $X$ itself. A subgraph over $\mathbf{X} \subseteq \mathbf{V}$ in $G$ is denoted as $G(\mathbf{X})$ and $G_{\overline{\mathbf{X}}}$ denotes the subgraph by removing arrows coming into nodes in $\mathbf{X}$.

A *soft intervention* on a variable $X$, denoted $\sigma_X$, replaces $f_X$ with a new function $f'_X$ of $\mathbf{Pa'} \subset \mathbf{V}$ and variables $\mathbf{U}'_X$ [58, 59]. For interventions on a set of variables $\mathbf{X} \subseteq \mathbf{V}$, let $\sigma_{\mathbf{X}} = \{\sigma_{\mathbf{X}}\}_{X \in \mathbf{X}}$, that is, the result of applying one intervention after the other. Given an SCM $\mathcal{M}$, let $\mathcal{M}_{\sigma_{\mathbf{x}}}$ be a submodel of $\mathcal{M}$ induced by intervention $\mathcal{M}_{\sigma_{\mathbf{x}}}$. A special class of soft interventions, resulting in observational distributions, called *idle* intervention, leaves the function as it is, which means $\sigma_{\mathbf{X}} = \{\}$. Another special class of stochastic soft interventions, called *perfect* interventions [21, 51] and denoted as $\mathrm{perf}(\mathbf{X})$, such that $\mathbf{Pa}(\mathbf{X}) = \emptyset$ and $\mathbf{U}'_X \cap \mathbf{U} = \emptyset$. This implies that the modified diagram induced by $\mathcal{M}_{\sigma_{\mathbf{x}}}$ is $G_{\overline{\mathbf{X}}}$. We assume soft interventions that are not hard do not change the structure of the graph [5]. Namely, the diagram induced by $\mathcal{M}_{\sigma_{\mathbf{x}}}$ is the same with $G$.

## 2 Modeling Disentangled Representation Learning (General Case)

In this section, we formalize the disentangled representation learning task in causal language. We leverage an Augmented SCMs, to model the generative process over *latent* causal variables $\mathbf{V}$.

**Definition 2.1** (Augmented Structure Causal Model). An Augmented Structure Causal Model (for short, ASCM) over a generative level SCM $\mathcal{M}_0 = \langle \{\mathbf{U}_0, \mathbf{V}_0, \mathcal{F}_0, P^0(\mathbf{U}_0)\} \rangle$ is a tuple $\mathcal{M} = \langle \mathbf{U}, \{\mathbf{V}, \mathbf{X}\}, \mathcal{F}, P(\mathbf{U}) \rangle$ such that (1) exogenous variables $\mathbf{U} = \mathbf{U}_0$; (2) $\mathbf{V} = \mathbf{V}_0 = \{V_1, \ldots, V_d\}$ are $d$ latent endogenous variables; $\mathbf{X}$ is an $m$ dimensional mixture variable; (3) $\mathcal{F} = \{\mathcal{F}_0, f_{\mathbf{X}}\}$, where $f_{\mathbf{X}} : \mathbb{R}^d \to \mathbb{R}^m$ is a diffeomorphic [6] function that maps from (the respective domains of) $\mathbf{V}$ to $\mathbf{X}$. $\exists \ \ h = f_{\mathbf{X}}^{-1}$ such that $\mathbf{V} = h(\mathbf{X})$; and (4) $P(\mathbf{U}_0) = P^0(\mathbf{U}_0)$. $\qquad\square$

In words, an ASCM $\mathcal{M}$ describes a two-stage generative process involving latent generative factors $\mathbf{V}$ and high-dimensional mixture $\mathbf{X}$ (e.g., images, or text). First, latent generative factors $\mathbf{V} \in \mathbb{R}^d$ are generated by an underlying SCM. The causal diagram induced by $\mathcal{M}_0$ over $\mathbf{V}$ is called *a latent causal graph* (LCG); denoted as $G$ here. Next, a nonparametric diffeomorphism $f_{\mathbf{X}}$ mixes $\mathbf{V}$ to get the high-dimensional mixture $\mathbf{X} \in \mathbb{R}^m$. An important aspect of $f_{\mathbf{X}}$ is that it is invertible regarding $\mathbf{V}$ which implies that the generative factors $\mathbf{V}$ are recognized in a given $\mathbf{X} = \mathbf{x}$ [7].

The initial disentangled representation learning setting can be traced back at least to linear/nonlinear ICA [7–9], where $G$ is assumed to have no edges ($\mathbf{V}$ are independent of each other) and Markovian (no bidirected edges in the LCG). More recently, allowing latent variables to have edges in the LCG was studied, albeit still under the Markovian assumption [11, 13, 14, 21, 22, 51, 55]. We relax this assumption and allow confounding to exist between $\mathbf{V}$, which we call non-Markovianity[8].

**Domains.** We address the general setting of distributions that arise from multiple domains. Following [32–35, 62, 63], we define the so-called *latent selection diagram* that represents a collection of ASCMs to model the multi-domain setting. Selection diagrams enable us to compactly represent causal structure and cross-domain invariances [9].

**Definition 2.2** (Latent Selection Diagrams). Let $\boldsymbol{\mathcal{M}} = \langle \mathcal{M}_1, \mathcal{M}_2, ..., \mathcal{M}_N \rangle$ be a collection of ASCMs relative to $N$ domains $\boldsymbol{\Pi} = \langle \Pi_1, \Pi_2, ..., \Pi_N \rangle$, sharing mixing function $f_{\mathbf{X}}$ and LCG, $G$. $\boldsymbol{\mathcal{M}}$ defines a **latent selection diagram** (LSD. for short) $G^S$, constructed as follows: (1) every edge in $G$ is also an edge in $G^S$; (2) $G^S$ contains an extra node $S^{i,j}$ and corresponding edge $S^{i,j} \to V_k$ whenever there exists a discrepancy $f_{V_k}^i \neq f_{V_k}^j$, or $P^i(U_k) \neq P^j(U_k)$ between $\mathcal{M}_i$ and $\mathcal{M}_j$. $\qquad\square$

S-nodes indicate possible differences over $\mathbf{V}$ due to changes in the underlying mechanism or exogenous distributions across domains. For example, consider the LSD in Fig. 2. The S-node $S^{i,j}$ implies

---

[5]General soft interventions can arbitrarily change the graph by adding or removing edges. We do not consider this setting, and refer the readers to [1, 58, 59] for a general discussion on soft interventions.

[6]A diffeomorphism is a bijective function $f_{\mathbf{X}}$ such that both $f_{\mathbf{X}}$ and $f_{\mathbf{X}}^{-1}$ are continuously differentiable [27]

[7]Further discussion on the invertibility and non-parametric assumption is provided in Appendix A.2.

[8]To our knowledge, this is the first work in disentangled casual representation learning to relax Markovianity, which we believe is important since a significant challenge in causal inference stems from the existence of confounding bias traced back to Rubin [60], Pearl [1, 61], and more recently data fusion [36].

[9]See [32, 33] and Appendix Sec. A.3 for a more detailed discussion on the fundamental differences between interventions and domains, and why modeling their distinction is important in general.

that $V_1$ possibly changes from domain $\Pi^i$ to $\Pi^j$, while the other variable's mechanisms are assumed to be invariant. Note no S-node points to $\mathbf{X}$ since $f_{\mathbf{X}}$ is shared across $\mathcal{M}$.

**Interventions.** A set of interventions $\boldsymbol{\Sigma} = \{\sigma^{(k)}\}_{k=1}^K$ are applied across domains $\boldsymbol{\Pi}$, where $k$ is an index from 1 to $K$. The corresponding domains that $\boldsymbol{\Sigma}$ are intervened in is denoted as $\boldsymbol{\Pi}^{\boldsymbol{\Sigma}} = \{\Pi^{(k)}\}_{k=1}^K$ (the domains associated with each $\sigma^{(k)} \in \boldsymbol{\Sigma}$). We study a general setting where each intervention can be applied to any subset of nodes and in any domain, which can be seen as a generalization of the more restricted settings in prior work (see Appendix F).

The intervention targets collection of these $K$ interventions $\{\sigma^{(k)}\}_{k=1}^K$ is denoted as $\boldsymbol{\Psi} = \{\mathbf{I}^{(k)}\}_{k=1}^K$. Each intervention target $\mathbf{I}^{(k)}$ is given in the form of $\{V_i^{\Pi^{(k)},\{b\},t}, V_j^{\Pi^{(k)},\{b'\},t'}, \dots\}$, which indicates the intervention $\sigma^{(k)}$ changes the mechanism of $\{V_i, V_j, \dots\}$ in domain $\Pi^{(k)}$. $\{b\}$ indicates the mechanism of the intervention on the same node. The mechanisms of $V_i^1$ and $V_i^2$ are different while the mechanism on different nodes ($V_i^1$ and $V_j^1$) is default different. $t = \text{perf}$ indicates the intervention is perfect. When $\mathbf{I}^{(k)} = \{V_i^{\Pi^{(k)},t}\}$, where $\{b\}$ is omitted, then the intervention is assumed to have a different mechanism. When $\mathbf{I}^{(k)} = \{V_i^{\Pi^{(k)},\{b\}}\}$, where $t$ is omitted, then the intervention is assumed to be a general soft intervention. When $\mathbf{I}^{(k)}$ is an idle intervention in $\Pi^{(n)}$, it is denoted as $\{\}^{\Pi^{(n)}}$. The set $\text{Perf}[\mathbf{I}^{(k)}]$ is a set of variables with perfect interventions in $\sigma^{(k)}$. Thus when $\text{Perf}[\mathbf{I}^{(k)}] = \{\}$, it implies there are no variables with perfect interventions in $\sigma^{(k)}$. $\boldsymbol{\Psi}_{\mathbf{T}}^{\text{perf}}$ is a subset of $\boldsymbol{\Psi}$ such that $\mathbf{T} \subseteq \text{Perf}[\mathbf{I}^{(j)}]$ for every $\mathbf{I}^{(j)} \in \boldsymbol{\Psi}_{\mathbf{T}}^{\text{perf}}$, which implies $\mathbf{I}^{(j)}$ contains perfect interventions on $\mathbf{T}$; see Fig. S1 and Ex. 7 illustrating the notation.

**Given Distributions.** The interventions $\boldsymbol{\Sigma} = \{\sigma^{(k)}\}_{k=1}^K$, induce distributions $\mathcal{P} = \{P^{(k)}\}_{k=1}^K$ in multi-domains, where $P^{(k)} = P^{\Pi^{(k)}}(\mathbf{X}; \sigma^{(k)})$.

**Problem Statement** Suppose the underlying true model $\mathcal{M}$ induces the LSD $G^S$ and a collection of distributions $\mathcal{P}$ over $\mathbf{X}$ is given according to a corresponding collection of interventions $\Sigma$. The goal of this paper is to learn a disentangled representation $\widehat{\mathbf{V}}$ of the latent generative factors $\mathbf{V}$ in $\mathcal{M}$. In the literature, it is common to require every variable $V_i \in \mathbf{V}$ to be disentangled from all other variables [7, 21] or some special subset (e.g. non-ancestors of $V_i$) [21, 22]. However, as illustrated in Fig. 2, sometimes only the target variables ($\mathbf{V}^{tar} \subseteq \mathbf{V}$) is needed to be disentangled from some user-chosen entangled variables ($\mathbf{V}^{en}$). Recent work has also considered a similar goal of general-

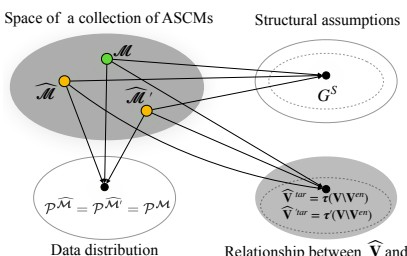

Figure 3: General ID/disentangleability.

ized disentanglement [24]. Our work still differs from theirs in the following ways: i) [assumptions] we model a completely nonparametric non-Markovian ASCM, whereas [24] assumes sparsity and a Markovian ASCM, and ii) [input] we consider arbitrary combinations of distributions from multiple domains, whereas [24] considers only interventions within a single domain (see Appendix F for a detailed comparison). We formally define this type of general indeterminacy next as well as the formal version of our ID task.

**Definition 2.3** (General Identifiability/Disentangleability (ID)). Let $\mathcal{M}$ be the underlying true ASCMs inducing LSD $G^S$, and $\mathcal{P} = \{P^{(k)}\}_{k=1}^K$ a set of distributions resulting from $K$ intervention sets $\Sigma$. Consider target variables $\mathbf{V}^{tar} \in \mathbf{V}$, and $\mathbf{V}^{en} \subseteq \mathbf{V} \backslash \mathbf{V}^{tar}$. The set $\mathbf{V}^{tar}$ is identifiable (disentangled) with respect to (from) $\mathbf{V}^{en}$ if there exists a function $\boldsymbol{\tau}$ such that $\widehat{\mathbf{V}}^{tar} = \boldsymbol{\tau}(\mathbf{V} \backslash \mathbf{V}^{en})$ for any $\widehat{\mathcal{M}}$ that is compatible with $G^S$ and $\mathcal{P}^{\widehat{\mathcal{M}}} = \mathcal{P}$. For short, $\mathbf{V}^{tar}$ is said to be ID w.r.t. $\mathbf{V}^{en}$. □

To illustrate, consider a target variable $\mathbf{V}^{tar}$ such that one wants its representation to be disentangled from another subset variables $\mathbf{V}^{en}$. The above definition states that $\mathbf{V}^{tar}$ is disentangled from $\mathbf{V}^{en}$ (or is ID w.r.t. $\mathbf{V}^{en}$) if the learned representations $\widehat{\mathbf{V}}^{tar}$ in $\widehat{\mathcal{M}}$ is only a function of $\mathbf{V} \backslash \mathbf{V}^{en}$ for *any* $\widehat{\mathcal{M}}$ that matches with the LSD $G^S$ and distribution $\mathcal{P}$[10]. Def 2.3 is illustrated in Fig. 3. Following the example illustrated in Fig. 2, suppose the user wants $V_3$ to be disentangled from $V_1$ while considering the entanglement between $V_2$ and $V_3$ acceptable. If $\widehat{V}^3 = \tau(V^2, V^3)$ for any ASCM $\widehat{\mathcal{M}}$ matches the

---

[10]In general, this definition is defined after a permutation of variables. We illustrate more in Sec. A.4.

distributions and LSD, $V_3$ is ID w.r.t. $V_1$. Def. 2.3 is more relaxed since one is free to choose any target $\mathbf{V}^{tar}$ and $\mathbf{V}^{en}$. It can be reduced to existing identifiability definitions (Appendix F.2).

**Example 1** (Example of an ID task). *Suppose the pair of underlying ASCMs $\langle \mathcal{M}_1, \mathcal{M}_2 \rangle$ induces the LSG $G^S$ in Fig. 2 and distributions $\mathcal{P} = \{P^{(1)}, P^{(2)}, P^{(3)}, P^{(4)}\} = \{P^{\Pi_1}(\mathbf{X}), P^{\Pi_2}(\mathbf{X}), P^{\Pi_2}(\mathbf{X}; \sigma_{V_3}), P^{\Pi_1}(\mathbf{X}; \sigma_{V_4})\}$ from interventions $\Sigma = \{\sigma^{(1)}, \sigma^{(2)}, \sigma^{(3)}, \sigma^{(4)}\} = \{\{\}, \{\}, \sigma_{V_3}, \text{perf}(V_2)\}$. Given intervention targets $\boldsymbol{\Psi} = \{\mathbf{I}^{(1)}, \mathbf{I}^{(2)}, \mathbf{I}^{(3)}, \mathbf{I}^{(4)}\} = \{\{\}^{\Pi_1}, \{\}^{\Pi_2}, V_3^{\Pi_2}, V_2^{\Pi_1,\text{perf}}\}$ and $G^S$, the task is to determine whether (and how) $\{V_2, V_3\}$ is ID w.r.t. $V_1$, and $V_1$ is ID w.r.t $\{V_2, V_3\}$. The answer to this is provided in Ex. 6.* □

**Assumptions (Informal) and Modeling Concepts**  Before discussing the main theoretical contributions, we restate important assumptions and remarks (discussed in this section) here to ground the ASCM model [11].

**Assumption 1** (Soft interventions without altering the causal structure). *Interventions do not change the causal diagram. Hard interventions cut all incoming parent edges, and soft interventions preserve them [59]. However, more general interventions may arbitrarily change the parent set for any given node [59]. We do not consider such interventions and leave this general case for future work.*

**Assumption 2** (Known-target interventions). *All interventions occur with known targets, reducing permutation indeterminacy for intervened variables.*

**Assumption 3** (Sufficiently different distributions). *Each pair of distributions $P^{(j)}$ and $P^{(k)} \in \mathcal{P}$ are sufficiently different, unless stated otherwise. This is naturally satisfied if ASCMs and interventions are randomly chosen [51]. Similar assumptions include the "genericity" [51], "interventional discrepancy" [21], and "sufficient changes" assumptions [10, 22].*

**Remark 1** (Mixing is invertible). As a consequence of Def. 2.1, the mixing function $f_{\mathbf{X}}$ is invertible, ensuring that latent variables are uniquely learnable [9, 10, 17, 64].

**Remark 2** (Confounders are not part of the mixing function). According to Def. 2.1, latent exogenous variables $\mathbf{U}$ influence the high-dimensional mixture $\mathbf{X}$ only through latent causal variables $\mathbf{V}$, so unobserved confounding $\mathbf{U}$ does not directly affect the mixing function.

**Remark 3** (Shared causal structure). As a consequence of Def. 2.2, each environment's ASCM shares the same latent causal graph, with no structural changes among latent variables [12].

## 3  Graphical Criterion for Causal Disentanglement

In this section, we study a general form of identifiability given general assumptions and input distributions. More specifically, we build the connection from $\mathbf{V}$ and representation $\widehat{\mathbf{V}}$ through comparing distributions and then introduce three graphical criteria (Prop. 3, 4 and 5) to check ID.

First, we introduce a factorization of distributions induced by non-Markovian models [3, Def. 15]. Specifically, consider $P_{\mathbf{T}}(\mathbf{V})$ induced by an ASCM $\mathcal{M}$ after a perfect intervention on $\mathbf{T}$. Then, given a topological order $<$ of $G$, $P_{\mathbf{T}}(\mathbf{V})$ can be factorized as follows:

$$P_{\mathbf{T}}(\mathbf{V}) = \prod_{V_i \in \mathbf{V}} P_{\mathbf{T}}(V_i | \mathbf{Pa}_i^{\mathbf{T}+}) \tag{1}$$

where $\mathbf{Pa}_i^{\mathbf{T}+} = \overline{\mathbf{Pa}}(\{V \in \mathbf{C}(V_i) : V \leq V_i\}) \setminus \{V_i\}$ is the extended parents set of $V_i$ in $G_{\overline{\mathbf{T}}}$. The factorization form for $P_{\mathbf{T}}(\mathbf{V})$ will be different according to the choice given order.

**Example 2.** *Consider a collection of $\boldsymbol{\mathcal{M}}$ inducing the LSD shown in Fig. 4(c). Given order A: $V_1 < V_2 < V_3 < V_4$, $P(\mathbf{V})$ can be factorized as: $P(V_1)P(V_2 | V_1)P(V_3 | V_2, V_1)P(V_4 | V_3)$ Notice that the conditioning part of $V_3$ includes $\{V_2, V_1\}$, which are not parents of $V_3$. Choosing order B: $V_1 < V_3 < V_2 < V_4$, $P(\mathbf{V})$ can be factorized as $P(V_1)P(V_3)P(V_2 | V_1, V_3)P(V_4 | V_3)$. The conditioning part of $V_2$ and $V_3$ are different given different orders.* □

---

[11]For a formal discussion on the assumptions, we refer the readers to Appendix A.2)

[12]The assumption that there are no structural changes between domains can be relaxed and is considered in the context of inference when causal variables are fully observed, as discussed in [33]. This is an interesting topic for future explorations, and we do not consider this avenue here.

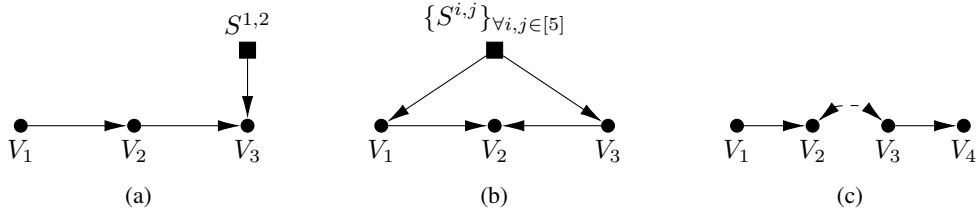

Figure 4: LSDs in Ex. and Exps. (a) chain, (b) collider and (c) non-markovian graphs.

Armed with this factorization, the representation $\widehat{V}$ in $\widehat{\mathcal{M}}$ and the true underlying variables $\mathbf{V}$ in $\mathcal{M}$ can be related by comparing distributions as follows.

**Proposition 1** (**Distribution Comparison**). *Consider a pair of collections ASCMs $\mathcal{M}$ and $\widehat{\mathcal{M}}$ that matches with the distribution $\mathcal{P}$ resulting from interventions $\Sigma$ and LSD $G^S$. Consider two distributions $P^{\Pi^{(j)}}(\mathbf{X}; \sigma^{(j)})$ and $P^{\Pi^{(k)}}(\mathbf{X}; \sigma^{(k)})$. Suppose $\mathrm{perf}(\mathbf{T})$ is in both intervention sets, then,*

$$\sum_i^d \log p_{\mathbf{T}}^{(j)}(v_i \mid \mathbf{pa}_i^{\mathbf{T}+}) - p_{\mathbf{T}}^{(k)}(v_i \mid \mathbf{pa}_i^{\mathbf{T}+}) = \sum_i^d \log p_{\mathbf{T}}^{(j)}(\widehat{v}_i \mid \widehat{\mathbf{pa}}_i^{\mathbf{T}+}) - \log p_{\mathbf{T}}^{(k)}(\widehat{v}_i \mid \widehat{\mathbf{pa}}_i^{\mathbf{T}+}), \quad (2)$$

*where $p_{\mathbf{T}}^{(j)}(\cdot)$ and $p_{\mathbf{T}}^{(k)}(\cdot)$ are density functions.* □

To illustrate, Prop.1 shows when the intervention and domain changes from $\sigma^{(k)}$ to $\sigma^{(j)}$ and $\Pi^{(k)}$ to $\Pi^{(j)}$, the change comes from factors $p_{\mathbf{T}}(v_i \mid \mathbf{pa}_i^{\mathbf{T}+})$ both in $\mathcal{M}$ and $\widehat{\mathcal{M}}$.

However, not all factors necessarily contribute to Eq (2). For example, in the Markovian setting, only one factor $p_{\mathbf{T}}(v_i \mid \mathbf{pa}_i)$ possibly changes when comparing the observational to a singleton interventional distribution in the same domain. Other invariant factors will be canceled out in Eq. (2). The following result generalizes finding invariant factors when comparing distributions from different domains and interventions in non-Markovian settings.

**Proposition 2** (**Invariant Factors**). *Consider two distributions $P^{(j)}, P^{(k)} \in \mathcal{P}$ with intervention targets $\sigma^{(j)}$ and $\sigma^{(k)}$ containing $\mathrm{perf}(\mathbf{T})$. Construct the changed variable set $\Delta \mathbf{V}[\mathbf{I}^{(j)}, \mathbf{I}^{(k)}, G^S]$ (for short $\Delta \mathbf{V}$) with target sets $\mathbf{I}^{(j)}, \mathbf{I}^{(k)}$ as follows: (1) $V_l \in \Delta \mathbf{V}$ if $V_l^{\pi_l, \{b_l\}, t_l} \in \mathbf{I}^{(j)}$ but $V_l^{\pi_l', \{b_l\}, t_l'} \notin \mathbf{I}^{(k)}$, or vice versa; (2) $V_l \in \Delta \mathbf{V}$ if i) $S^{\Pi^{(j)}, \Pi^{(k)}}$ point to $V_l$ and ii) $V_l^{\pi_l, \{b_l\}, t_l} \notin \mathbf{I}^{(j)} \cup \mathbf{I}^{(k)}$. If $V_i \in \mathbf{V} \backslash \mathbf{C}(\Delta \mathbf{V})$, then $p_{\mathbf{T}}^{(j)}(v_i \mid \mathbf{pa}_i^{\mathbf{T}+}) = p_{\mathbf{T}}^{(k)}(v_i \mid \mathbf{pa}_i^{\mathbf{T}+})$ (denoted invariant factors).* □

Prop. 2 states that factors $p_{\mathbf{T}}(v_i \mid \mathbf{pa}_i^{\mathbf{T}+})$ are guaranteed to be invariant if $V_i$ is not in the C-component of the changed variable set $\Delta \mathbf{V}$. $\Delta \mathbf{V}[\mathbf{I}^{(j)}, \mathbf{I}^{(k)}, G^S]$ contains variables that are intervened differently in $\mathbf{I}^{(j)}, \mathbf{I}^{(k)}$ and the variables pointed by S-node, $S^{j,k}$ [13].

**Example 3.** *Consider the diagram in Fig. 4(c) and two distributions $P^{(1)}, P^{(2)} \in \mathcal{P}$ with intervention targets $\mathbf{I}^{(1)} = \{\}^{\Pi_1}$ and $\mathbf{I}^{(2)} = \{V_2^{\Pi_1}\}$. The changed variable set $\Delta \mathbf{V}^{(2),(1)} = \{V_2, V_3\}$ since $V_2 \in \mathbf{I}^{(2)}, V_2 \notin \mathbf{I}^{(1)}$, and $\mathbf{C}(V_2) = \{V_2, V_3\}$. Thus, comparing $P^{(2)}$ with $P^{(1)}$ (order A in Ex. 2), factors $p(v_1), p(v_4 \mid v_2, v_1)$ are invariant, whereas $p(v_2 \mid v_1), p(v_3 \mid v_2, v_1)$ can change.* □

With Prop. 2, Eq. (2) naturally keeps factors only in the C-component of $\Delta \mathbf{V}$, i.e.,

$$\sum_{V_i \in \tilde{\mathbf{V}}} \log p_{\mathbf{T}}^{(j)}(v_i \mid \mathbf{pa}_i^{\mathbf{T}+}) - p_{\mathbf{T}}^{(k)}(v_i \mid \mathbf{pa}_i^{\mathbf{T}+}) = \sum_{V_i \in \tilde{\mathbf{V}}} \log p_{\mathbf{T}}^{(j)}(\widehat{v}_i \mid \widehat{\mathbf{pa}}_i^{\mathbf{T}+}) - \log p_{\mathbf{T}}^{(k)}(\widehat{v}_i \mid \widehat{\mathbf{pa}}_i^{\mathbf{T}+}) \quad (3)$$

where $\tilde{\mathbf{V}} = \mathbf{C}(\Delta \mathbf{V}[\mathbf{I}^{(j)}, \mathbf{I}^{(k)}, G^S])$. This factorization hints that $\widehat{\mathbf{V}}$ (RHS of Eq. (3)) is only related to variables that appear on the LHS.

**Definition 3.1** ($\Delta \mathbf{Q}$ Set). *Given two distributions $P^{(j)}, P^{(k)}$ with interventions targets $\sigma^{(j)}$ and $\sigma^{(k)}$ containing perfect interventions on $\mathbf{T}$, the $\Delta \mathbf{Q}[\mathbf{I}^{(j)}, \mathbf{I}^{(k)}, \mathbf{T}, G^S]$ set (for short: $\Delta \mathbf{Q}^{(j),(k)}$, or $\Delta \mathbf{Q}$ if index not needed) of the target sets $\mathbf{I}^{(j)}, \mathbf{I}^{(k)}$ is the remaining variables after comparison (i.e. Eq. 3), $\Delta \mathbf{Q}[\mathbf{I}^{(j)}, \mathbf{I}^{(k)}, \mathbf{T}, G^S] = \tilde{\mathbf{V}} \cup \mathbf{Pa}^{\mathbf{T}+}(\tilde{\mathbf{V}})$, where $\tilde{\mathbf{V}} = \mathbf{C}(\Delta \mathbf{V}[\mathbf{I}^{(j)}, \mathbf{I}^{(k)}, G^S])$.* □

---

[13]Notice that the same intervention mechanism will dominate the domain changes, which means when the intervened mechanism of $V_l$ is the same between $\mathbf{I}^{(j)}$ and $\mathbf{I}^{(k)}$, the discrepancy of $V_l$ due to the change of domain between $\Pi^{(j)}$ and $\Pi^{(k)}$ will be canceled. See Appendix A.3 for an example.

To illustrate, the $\Delta\mathbf{Q}$ set involves all variables in LHS of Eq. (3), including $\tilde{\mathbf{V}}$ and its extended parents. These variables come from factors that possibly change and are kept in Eq. (3). We call $\mathbf{V}\backslash\Delta\mathbf{Q}$ *canceled variables* since invariant factors are canceled out from the comparison. Continuing Ex. 3, $\Delta\mathbf{Q} = \{V_1, V_2, V_3\}$ given either topological order.

Leveraging the comparisons among distributions in $\mathcal{P}$ (Eq. 3), we next develop three criterions for disentanglement. First, we can disentangle canceled variables from $\Delta\mathbf{Q}$ set since the difference of density over representations $\widehat{\mathbf{V}}$ in the $\Delta\mathbf{Q}$ set (RHS of Eq. (3)) is irrelevant to canceled variables (LHS of Eq. (3)).

**Proposition 3** (**ID the $\Delta\mathbf{Q}$ set w.r.t Canceled Variables**). *Consider variables $\mathbf{V}^{tar} \subseteq \mathbf{V}$. Let $\mathcal{P}_{\mathbf{T}} = \{P^{(a_0)}, P^{(a_1)}, \ldots, P^{(a_L)}\} \subseteq \mathcal{P}$ be a collection of distributions such that (1) $\forall\ l \in [L]$, $\mathbf{T} = \mathrm{Perf}[\mathbf{I}^{(a_0)}] \subseteq \mathrm{Perf}[\mathbf{I}^{(a_l)}]$ [14]; (2) $\bigcup_{l \in [L]} \Delta\mathbf{Q}[\mathbf{I}^{(a_l)}, \mathbf{I}^{(a_0)}, \mathbf{T}, G^S] = \mathbf{V}^{tar}$; (3) there exists $\{a'_1, \ldots, a'_{d'}\} \subseteq \{a_1, \ldots, a_L\}$ such that for all $V_i^{tar} \in \mathbf{V}^{tar}, V_i^{tar} \in \Delta\mathbf{Q}[\mathbf{I}^{(a'_i)}, \mathbf{I}^{(a_0)}, \mathbf{T}, G^S]$, where $d' = |\mathbf{V}^{tar}|$. Then, $\mathbf{V}^{tar}$ is ID w.r.t $\mathbf{V}\backslash\mathbf{V}^{tar}$.* □

Prop. 3 disentangles target variables $\mathbf{V}^{tar}$ (as a union of $\Delta\mathbf{Q}$ sets) from canceled variables according to Eq. (3). To illustrate, it considers to find a collection of $L$ distribution $\{P^{(a_1)}, \ldots, P^{(a_L)}\}$ to compare with the baseline $P^{(a_0)}$ such that (1) the perfect intervention variables sets of $\{\mathbf{I}^{(a_1)}, \ldots, \mathbf{I}^{(a_L)}\}$ contain the perfect intervention set of the baseline $\mathbf{I}^{(a_0)}$, (2) the union of $\Delta\mathbf{Q}$ is equivalent to $\mathbf{V}^{tar}$, and (3) each $V_i^{tar}$ changes at least once. Then, $\mathbf{V}^{tar}$ can be ID wrt $\mathbf{V}\backslash\mathbf{V}^{tar}$.

**Example 4.** *(Ex. 3 continued.) Suppose $\mathcal{P} = \{P^{(1)}, P^{(2)}, P^{(3)}, P^{(4)}\}$ with intervention targets $\mathbf{I}^{(1)} = \{\}^{\Pi_1}, \mathbf{I}^{(2)} = \{V_2\}^{\Pi_1}, \mathbf{I}^{(3)} = \{V_3\}^{\Pi_1}, \mathbf{I}^{(4)} = \{V_1\}^{\Pi_1}$. Consider $\mathbf{V}^{tar} = \{V_1, V_2, V_3\}$ and $\mathbf{V}^{en} = \mathbf{V}\backslash\{V_1, V_2, V_3\} = \{V_4\}$. Comparing $\{\mathbf{I}^{(2)}, \mathbf{I}^{(3)}, \mathbf{I}^{(4)}\}$ with the baseline $\mathbf{I}^{(1)}$, the perfect intervention variables are $\mathbf{T} = \mathrm{Perf}[\mathbf{I}^{(1)}] = \{\}$. Then we have $\Delta\mathbf{Q}$ sets: $\{V_1, V_2, V_3\}, \{V_1, V_2, V_3\}$ and $V_1$. Thus, these three comparisons satisfy the three conditions in Prop. 3. Then $\mathbf{V}^{tar}$ is ID w.r.t $\mathbf{V}^{en}$ by Prop. 3. See Appendix Ex. 17 for a derivation.* □

According to Prop. 3, a disentanglement corollary leveraging the comparison of interventions and observational distributions can be derived.

**Corollary 1** (**ID intervened variables**). *Given an observational distribution and $L$ distributions resulting from interventions on the same target $\mathbf{W}$ but with different mechanisms (in the same domain), if $L \geq |\mathbf{Pa}^+_{\mathbf{W}} \cup \mathbf{W}|$, $\mathbf{Pa}^+_{\mathbf{W}} \cup \mathbf{W}$ is ID w.r.t $\mathbf{V}\backslash\{\mathbf{Pa}^+_{\mathbf{W}} \cup \mathbf{W}\}$, where $\mathbf{Pa}^+_{\mathbf{W}} = \cup_{W_i \in \mathbf{w}}\mathbf{Pa}^+_{W_i}$.* □

The second result disentangles variables within $\Delta\mathbf{Q}$ sets.

**Proposition 4** (**ID of variables within $\Delta\mathbf{Q}$ sets**). *Consider the variables $\mathbf{V}^{tar} \subseteq \mathbf{V}$, $\mathcal{P}_{\mathbf{T}}$ that satisfies conditions (1) in Prop. 3 and $\Delta\mathbf{Q}^{(a_l),(a_0)} = \mathbf{V}^{tar}$, for $l \in [L]$. For any pair of $V_i, V_j \in \mathbf{V}^{tar}$ such that $V_i \perp\!\!\!\perp V_j | \mathbf{V}^{tar}\backslash\{V_i, V_j\}$ in $G_{\overline{\mathbf{T}}}(\mathbf{V}^{tar})$, $V_i$ is ID w.r.t. $V_j$ if $L \geq 2|\mathbf{V}^{tar}| + \delta_{\not\perp}$, where $\delta_{\not\perp}$ is the number of pair $V_k, V_r \in \mathbf{V}^{tar}$ such that $V_k$ and $V_r$ are connected given $\mathbf{V}^{tar}\backslash\{V_k, V_r\}$ in $G_{\overline{\mathbf{T}}}(\mathbf{V}^{tar})$.* □

Prop. 4 disentangles target variables $V_i$ and $V_j$ both in $\Delta\mathbf{Q}$ sets. To illustrate, consider a set of distributions that satisfies conditions (1) as Prop. 3. This proposition suggests that if (1) $V_i, V_j \in \mathbf{V}^{tar}$ are conditionally independent given all other variables in $\mathbf{V}^{tar}$, (2) $L$ is not smaller than $2|\mathbf{V}^{tar}| + \delta_{\not\perp}$, then $V_i$ can be disentangled from $V_j$.

**Example 5.** *Suppose LSD $G^S$ is a collider graph shown in Fig. 4(b). Suppose the intervention targets are $\mathbf{\Psi} = \{\{\}^{\Pi_1}, \{\}^{\Pi_2}, \{\}^{\Pi_3}, \{\}^{\Pi_4}, \{\}^{\Pi_5}\}$, which means that observational distributions are available in each domain. Consider $\mathbf{T} = \{\}$. Let $\mathbf{V}^{tar} = \{V_1, V_3\}$. We have $V_1 \perp\!\!\!\perp V_3$ in $G(V_1, V_3)$. Based on Def. 3.1, $\Delta\mathbf{Q}[\mathbf{I}^{(j)}, \mathbf{I}^{(1)}, \mathbf{T}, G^S] = \{V_1, V_3\}$ for $j = 2, 3, 4, 5$. Then the number of distributions used for comparing (i.e., four) is not smaller than the required ($2 \times 2 + 0$), which means $V_1$ is ID w.r.t. $V_3$ and $V_3$ is ID w.r.t. $V_1$ by Prop. 4. See App. Ex. 18 for a derivation.* □

With these existing disentanglements from Props. 3 and 4, the following Proposition considers an inverse direction, which identifies canceled variables w.r.t. $\Delta\mathbf{Q}$ sets [15].

**Proposition 5** (**ID of canceled variables w.r.t. $\Delta\mathbf{Q}$ sets**). *Suppose $\mathbf{\Psi}$ contains $\mathrm{perf}(\mathbf{T})$. Given $\mathbf{V}\backslash V^{tar}$ is ID w.r.t. a single variable $V^{tar}$, $V^{tar}$ is ID w.r.t. $\mathbf{V}\backslash V^{tar}$ if $V^{tar} \perp\!\!\!\perp \mathbf{V}\backslash V^{tar}$ in $G_{\overline{\mathbf{T}}}$.* □

---

[14]Recall we use the notation $\mathrm{Perf}[\mathbf{I}]$ to denote that all variables that have perfect interventions on $\mathbf{I}$.

[15]Recall that ID is one-way. ID of $V_i$ wrt $V_j$ does not imply $V_j$ is ID wrt $V_i$.

**Algorithm 1 CRID: Algorithm for determining causal representation identifiability** - $G^S$ is the LSD; $\Psi$ is the intervention target sets; $G_{\mathbf{V},\widehat{\mathbf{V}}}$ is the output bipartite graph (i.e. CDM).

---

**Input:** $G_{\mathbf{V}}$, and intervention target sets $\Psi$.
**Output:** CDM $G_{\mathbf{V},\widehat{\mathbf{V}}}$

1: $G_{\mathbf{V},\widehat{\mathbf{V}}} \leftarrow$ **FullyConnectedBipartiteGraph**$(\mathbf{V},\widehat{\mathbf{V}})$      $\triangleright$ Initialize $G_{\mathbf{V},\widehat{\mathbf{V}}}$ with Alg. F.2
2: **while** $G_{\mathbf{V},\widehat{\mathbf{V}}}$ is updated in the last epoch **do**
3:      $\mathbf{Perf} = \{\mathbf{T}_1, \mathbf{T}_2, \ldots, \mathbf{T}_s\} \leftarrow \Psi$      $\triangleright$ Get perfect intervention variables sets.
4:      **for** all $\mathbf{T} \in \mathbf{Perf}$ **do**
5:          $\Psi_{\mathbf{T}}^{\mathrm{perf}} \leftarrow \Psi$    $\triangleright$ Collect intervention targets that contain hard intervention variables $\mathbf{T}$
6:          **for** all $\mathbf{I} \in \Psi_{\mathbf{T}}^{\mathrm{perf}}$ such that $\mathrm{Perf}[\mathbf{I}] = \mathbf{T}$ **do** $\triangleright$ Iterate intervention targets as the baseline
7:              $\mathcal{Q} = \{\mathbf{Q}_1, \ldots, \mathbf{Q}_{|\Psi_{\mathbf{T}} \backslash \mathbf{I}|}\}$, where $Q_k \leftarrow \Delta\mathbf{Q}[\mathbf{J}^{(k)}, \mathbf{I}, \mathbf{T}, G^S]$    $\triangleright$ Construct $\Delta\mathbf{Q}$ sets.
8:              **for** all $\mathbf{Q}$ such that $\mathbf{Q} = \bigcup_{\mathbf{Q}_l \in \mathcal{Q}} \mathbf{Q}_l \subset \mathbf{V}$ **do**      $\triangleright$ Iterate the union of $\Delta\mathbf{Q}$ factors
9:                  $G_{\mathbf{V},\widehat{\mathbf{V}}} \leftarrow$ **Dis$\Delta$QFromCancel**$(\mathbf{Q}, G_{\mathbf{V},\widehat{\mathbf{V}}}, G_{\overline{\mathbf{T}}}, \Psi_{\mathbf{T}}^{\mathrm{perf}}, \mathbf{I}, \mathcal{Q})$    $\triangleright$ Alg. F.3, Prop. 3
10:                  $G_{\mathbf{V},\widehat{\mathbf{V}}} \leftarrow$ **DisWithin$\Delta$Q**$(\mathbf{Q}, G_{\mathbf{V},\widehat{\mathbf{V}}}, G_{\overline{\mathbf{T}}}, \Psi_{\mathbf{T}}^{\mathrm{perf}}, \mathbf{I}, \mathcal{Q})$       $\triangleright$ Alg. F.6, Prop. 4
11:      **for** all $\mathbf{T} \in \mathbf{Perf}$ **do**
12:          $G_{\mathbf{V},\widehat{\mathbf{V}}} \leftarrow$ **DisCancelFrom$\Delta$Q**$(G_{\mathbf{V},\widehat{\mathbf{V}}}, G_{\overline{\mathbf{T}}})$             $\triangleright$ Alg. F.8, Prop. 5
13: **return** $G_{\mathbf{V},\widehat{\mathbf{V}}}$

---

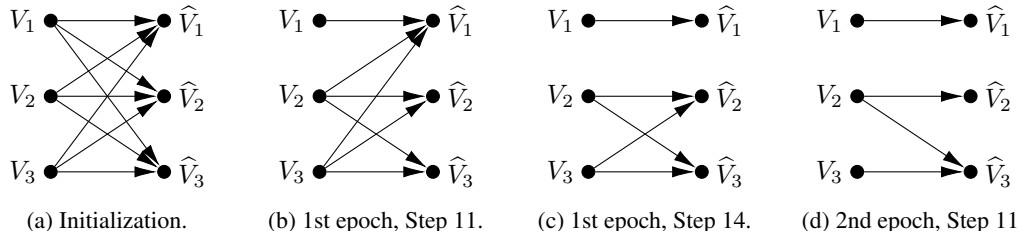

(a) Initialization.      (b) 1st epoch, Step 11.      (c) 1st epoch, Step 14.      (d) 2nd epoch, Step 11.

Figure 5: Process of removing edges from CDM $G_{\mathbf{V},\widehat{\mathbf{V}}}$ using Alg. 1 in Ex. 6. (d) is the final output.

To illustrate, Prop. 5 states: if $\mathbf{V} \backslash \{V^{tar}\}$ is already disentangled from $V^{tar}$, then $V^{tar}$ is ID wrt $\mathbf{V} \backslash \{V^{tar}\}$ if a perfect intervention on $\mathbf{T}$ exists to separate $V^{tar}$ and $\mathbf{V} \backslash \{V^{tar}\}$ in $G_{\overline{\mathbf{T}}}$. Prop. 5 does not compare distributions but relies on existing disentanglements. See Ex. 19 for details.

## 4 Algorithmic Disentanglement of Causal Representations

In this section, we develop an algorithmic procedure for determining whether $\mathbf{V}^{tar}$ and $\mathbf{V}^{en}$ are disentangleable given the LSD $G^S$ and interventions sets $\Psi$. The whole algorithm **CausalRepresentationID** (**CRID**, for short) is described in Alg. 1. We start by introducing a bipartite graph $G_{\mathbf{V},\widehat{\mathbf{V}}}$, called *Causal Disentanglement Map (CDM)* (which was informally shown in Fig 2 (right)). In words, the absence of the edge $V_i \not\rightarrow \widehat{V}_j$ implies $V_j$ is ID w.r.t $V_i$. If each $\widehat{V}_i$ is only pointed by $V_i$, then we have full disentanglement of $\mathbf{V}$. If $\mathbf{V} \subset V_i$ points to $\widehat{V}_i$, then we have partial disentanglement of $V_i$.

**CRID** proceeds by first constructing the fully connected CDM in Step 1. In each iteration, the hard intervention set $\mathbf{T}$ and the baseline intervention target set $\mathbf{I}$ (Steps 4 and 6) are enumerated. For each $\mathbf{T}$ and baseline, all $\Delta\mathbf{Q}$ sets are constructed based on Def. 3.1 and put into a collection $\mathcal{Q}$ (Steps 7). After the union of $\Delta\mathbf{Q}$ sets (denoted as $\mathbf{Q}$) is chosen (Step 8) iteratively, Props. 3 and 4 are leveraged in two procedures (Step 9 and 10) to check the identification of $\mathbf{Q}$ w.r.t. $\mathbf{V} \backslash \mathbf{Q}$ and the identification within $\mathbf{Q}$. The disentanglements in CDM at the current stage are leveraged to reduce the required number of distributions (see details in Alg. F.3 and F.6). At the end of the iteration, Prop. 5 is used for identifying $\mathbf{V} \backslash \mathbf{Q}$ from $\mathbf{Q}$ leveraging current disentanglement in CDM (Step 11-12).

**Example 6.** *(Ex. 1 continued.) Consider the selection diagram (Fig. 2) and the set up in Ex. 1 The perfect intervention variable sets are the empty set $\{\}$ and $\{V_2\}$. First, $\mathbf{T}$ is chosen as $\{\}$ and then $\Psi_{\mathbf{T}}^{\mathrm{perf}} = \Psi$. Choosing the baseline $\mathbf{I} = \mathbf{I}^{(1)}$, the $\Delta\mathbf{Q}$ collection: $\mathcal{Q} = \{\mathbf{Q}_1, \mathbf{Q}_2, \mathbf{Q}_3\} = \{\{V_2, V_3\}, \{V_2, V_3\}, \{V_1, V_2\}\}$. We consider the $\mathbf{Q}$ as $\{V_2, V_3\}$ and $\{V_1, V_2\}$. For $\mathbf{Q} = \{V_2, V_3\}$,*

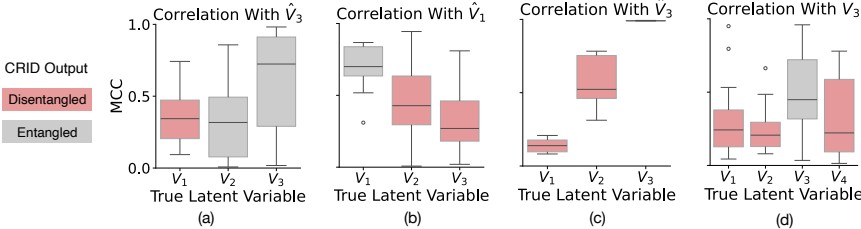

Figure 6: Correlation of learned latent representations with true latent variables from Fig. 4.

*leveraging Step 9 (Prop. 3), the edges from $V_1$ to $\{\widehat{V}_2, \widehat{V}_3\}$ are removed (See Ex. 15 for details) and Step 10 (Prop. 4) does not remove further edges. However, for $\mathbf{Q} = \{V_1, V_2\}$, no edge can be removed, since it at least needs two comparisons for claiming disentanglement.*

*Choosing $\{\}^{\Pi_2}$ or $V_3^{\Pi_2}$ or $V_2^{\Pi_1, \mathrm{do}}$ as the baseline, no new $\mathbf{Q}$ can be constructed, so no further edges are removed. When $\mathbf{T}$ is chosen as $\{V_2\}$, the comparison does not work since no other distribution is available. At the end of this iteration, with the fact that $\{V_2, V_3\}$ is ID wrt $V_1$ and $V_1 \perp\!\!\!\perp \{V_2, V_3\}$ in $G_{\overline{V_2}}$, Step 12 (Prop. 5) removes edges from $V_2$ to $\widehat{V}_1$ and $V_3$ to $\widehat{V}_1$. In the second iteration, the algorithm repeats the choice of $\mathbf{T}$ and the baseline. At this iteration, for $\mathbf{Q} = \{V_1, V_2\}$, the edge from $V_3$ to $\widehat{V}_2$ is removed since $V_3$ to $\widehat{V}_1$ has already been removed in CDM and only 1 comparison is needed now. At the end of this epoch no further can be removed by Alg. F.8. In the third epoch, $G_{\mathbf{V}, \widehat{\mathbf{V}}}$ is not updated and the process of CDM returned is shown in Fig. 5.* □

The following theorem indicates the soundness of CRID.

**Theorem 1** (**Soundness of CRID**). *Consider a LSD $G^S$ and intervention targets $\boldsymbol{\Psi}$. Consider the target variables $\mathbf{V}^{tar}$ and $\mathbf{V}^{en} \subseteq \mathbf{V} \backslash \mathbf{V}^{tar}$. If no edges from $\mathbf{V}^{tar}$ points to $\widehat{\mathbf{V}}^{en}$ in the output causal disentanglement map (CDM) from **CRID**, $G_{V, \widehat{V}}$, then $\mathbf{V}^{tar}$ is ID w.r.t $\mathbf{V}^{en}$.* □

## 5 Experiments

We corroborate the theoretical findings through simulations and MNIST dataset. For full details, see Appendix Section G. In simulations, we consider LSDs shown in Fig. 4 with different collection of distributions $\mathcal{P} = \{P^{(k)}(\mathbf{X}; \sigma^{(k)})\}_{k=1}^{K}$ and the results are presented in Fig. 6. For the evaluation, we follow a standard evaluation protocol in prior work [18], where we take the latent representations $\widehat{\mathbf{V}}$ and compute their mean correlation coefficient (MCC) wrt the latent $\mathbf{V}$. We compare MCC with what is expected from **CRID**. Fig. S9 shows the full MCC comparisons of $\mathbf{V}$ and $\widehat{\mathbf{V}}$.

**Chain Graph Fig. 4(a).** Fig. 6(a) shows ID of $V_3$ wrt $\{V_1\}$ using input distributions $\mathcal{P}$ with interventions $\Sigma = \{\sigma_{\{\}}, \sigma_3^{\{1\}}, \sigma_3^{\{2\}}\}$ because $MCC(\widehat{V}_3, V_1)$ is relatively low compared to $MCC(\widehat{V}_3, V_3)$, which is consistent with CRID. The ID results of [21] states $V_3$ would still be entangled with $V_1$ because $V_1 \in \overline{Anc}(V_3)$. Fig. 6(b) shows ID of $V_1$ wrt $\{V_2, V_3\}$. Interestingly, we do not even have to intervene on $V_1$ to obtain full disentanglement.

**Collider Graph Fig. 4(b).** Fig. 6(c) shows $V_1$ and $V_3$ are ID wrt $V_2$ and each other because $MCC(\widehat{V}_3, V_3) > MCC(\widehat{V}_3, V_i)$ and $MCC(\widehat{V}_2, V_2) > MCC(\widehat{V}_2, V_i)$, which is consistent with CRID. There are distributions from four domains that have a change-in-mechanism on $\{V_1, V_3\}$ (represented by the S-node). According to [22], since $V_1$ and $V_3$ are adjacent in the Markov Network, $V_1$ and $V_3$ are not disentangleable.

**Non-Markovian Graph Fig. 4(c).** Fig. 6(d) shows $V_3$ is ID wrt $\{V_1, V_2, V_4\}$ with interventions $\Sigma = \{do^{\{1\}}(V_3), do^{\{2\}}(V_3)\}$, which is consistent with CRID. No prior results achieve disentanglement with confounding among $\mathbf{V}$.

## 6 Conclusions

This work introduces theory and a practical ID algorithm for determining which latent variables are disentangleable from a given set of assumptions in the form of a LSD, and input distributions from heterogenous domains. This brings us one step closer to building robust AI that can reason causally over high-level concepts when only given low-level data.

## Acknowledgements

AL was supported by the NSF Computing Innovation Fellowship (#2127309). YP and EB were supported in part by the NSF, ONR, AFOSR, DoE, Amazon, JP Morgan, and The Alfred P. Sloan Foundation.

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

# Appendix

## Contents

# A  Background and Assumptions

## A.1   Notations

| Symbol | Description |
|---|---|
| $[d]$ | $\{1, 2, \ldots, d\}$ |
| $G$ | Latent Causal Graph (LCG) over $\mathbf{V}$ induced by an $\mathcal{M}$ |
| $\mathcal{M}$ | An ASCM (Def. 2.1) describes the data generation process of $d$ latent variables $\mathbf{V} \in \mathbb{R}^d$ and an observed high-dimensional mixture $\mathbf{X} \in \mathbb{R}^m$. |
| $G$ | Latent Causal Diagram (LCG) over $\mathbf{V}$ induced by an $\mathcal{M}$ |
| $\overline{\mathbf{Pa}}(\mathbf{V}), \overline{\mathbf{Pa}}_{\mathbf{V}}$ | The union of parents of $\mathbf{V}$ and $\mathbf{V}$ itself |
| $\mathbf{C}(\mathbf{V})$ | C-Component of $\mathbf{V}$ (Def.6.2). |
| $\boldsymbol{\mathcal{M}}$ | A set of $N$ ASCMs $\langle \mathcal{M}_1, \ldots, \mathcal{M}_N \rangle$ (shared mixing function $f_{\mathbf{X}}$) relative to domains $\mathbf{\Pi} = \langle \Pi_1, \ldots, \Pi_N \rangle$ |
| $G^S$ | Latent Selection Diagram (LSG, Def 2.2) induced by $\boldsymbol{\mathcal{M}}$ |
| $\Sigma = \{\sigma^{(k)}\}_{k=1}^K$ | A set of $K >= N$ interventions applied to $\boldsymbol{\mathcal{M}}$. Each intervention $\sigma^{(k)}$ can be idle, perfect, or other soft interventions that do not alter the structure of $G$ |
| $\mathbf{\Pi}^{\Sigma} = \{\Pi^{(k)}\}_{k=1}^K$ | The corresponding domains of interventions $\Sigma$. $\sigma^{(k)}$ is applied in $\Pi^{(k)}$ |
| $\mathbf{\Psi} = \{\mathbf{I}^{(k)}\}_{k=1}^K$ | The collection of intervened target sets of the intervention collection $\Sigma$. |
| $\mathbf{I}^{(k)} = \{V_i^{\Pi^{(k)}, \{b\}, t}, \ldots\}$ | The intervention target set of $\sigma^{(k)}$ |
| $\{b\}$ | The mechanism of intervention. Default as interventions have different mechanisms if $b$ is ignored. Also, the mechanism od different variables are different. The mechanism of $V_1^{\{1\}}$ is not equal to $V_2^{\{1\}}$. |
| $t$ | Whether an intervention is perfect or not. $t = do$ means it is perfect. Default as not perfect if $t$ is ignored. |
| $\mathrm{Perf}[\mathbf{I}^{(k)}]$ | Variables that are perfectly intervened on in $\sigma^{(k)}$. |
| $\mathbf{\Psi}_{\mathbf{T}}^{\mathrm{perf}}$ | The collection of intervention target sets that contain a perfect intervention on $\mathbf{T}$. |
| $\mathcal{P} = \{P^{(k)}\}_{k=1}^K$ | Set of distributions induced by $\boldsymbol{\mathcal{M}}$ resulting from collection of interventions $\Sigma$. $P^{(k)} = P^{\Pi^{(k)}}(\mathbf{X}; \sigma^{(k)})$ |
| $\mathbf{Pa}^{\mathbf{T}+}(\mathbf{V}), \mathbf{Pa}_{\mathbf{V}}^{\mathbf{T}+}$ | Extended parents from factorization Eq. (1). |
| $\Delta\mathbf{V}[\mathbf{I}^{(j)}, \mathbf{I}^{(k)}, G^S]$ | Changed variable sets constructed in Proposition 2. For short, $\Delta\mathbf{V}$ or $\Delta\mathbf{V}^{(j),(k)}$ when index is needed. |
| $\tilde{\mathbf{V}}$ | The C-Component of $\Delta\mathbf{V}[\mathbf{I}^{(j)}, \mathbf{I}^{(k)}, G^S]$. The factor $P(v_i \mid \mathbf{pa}^{\mathbf{T}}+)$ for $V_i \in \mathbf{V} \backslash \tilde{\mathbf{V}}$ remains invariant in Eq.( 2). |
| $\Delta\mathbf{Q}[\mathbf{I}^{(j)}, \mathbf{I}^{(k)}, \mathbf{T}, G^S]$ | $\Delta\mathbf{Q}$ set defined in Def. 3.1. Variables in $\Delta\mathbf{Q}$ set remains from Eq.( 2) to Eq.( 3). For short, $\Delta\mathbf{Q}$ or $\Delta\mathbf{Q}^{(j),(k)}$ when index is needed. |
| Canceled variables | The complement of $\Delta\mathbf{Q}$, which is $\mathbf{V} \backslash \Delta\mathbf{Q}$. |

Figure S1: Table of Notations

We gave an example to illustrate the notation of a collection of intervention target sets $\Psi$ and each intervention target set $\mathbf{I}^{(k)}$.

**Example 7.** *Let an intervention target collection be*

$$\Psi = \{\mathbf{I}^{(1)} = \{\{\}^{\Pi_1}\}, \mathbf{I}^{(2)} = \{V_1^{\Pi_1,\{1\}}\}, \mathbf{I}^{(3)} = \{V_1^{\Pi_2,\{2\}}, V_2^{\Pi_2,\{1\},\mathrm{perf}}\}, \mathbf{I}^{(4)} = \{V_1^{\Pi_2,\{1\}}, V_2^{\Pi_2,\mathrm{perf}}\}\} \tag{4}$$

*In words, $\Psi$ indicates 4 different interventions $\Sigma = \{\sigma^{(k)}\}_{k=1}^4$:*

$\sigma^{(1)}$; *an idle intervention is applied resulting in an observational distribution in the domain $\Pi^1$.*

$\sigma^{(2)}$; *a soft intervention with mechanism $\{1\}$ is applied to $V_1$ in domain $\Pi^1$.*

$\sigma^{(3)}$; *an intervention is applied to $V_1$ and $V_2$ in domain $\Pi^2$ where the mechanism of $V_1$ is different from $\sigma^{(2)}$ and the intervention on $V_2$ is perfect.*

$\sigma^{(4)}$; *an intervention is applied to $V_1$ and $V_2$ in domain $\Pi^2$ where the mechanism of $V_1$ is the same with $\sigma^{(2)}$ and the mechanism of $V_2$ is different from $\sigma^{(3)}$.*

$\mathrm{Perf}[\mathbf{I}^{(3)}] = \{V_2\}$ *means that $\sigma^{(3)}$ perfectly intervenes on $\{V_2\}$.*

$\Psi_{V_2}^{\mathrm{perf}} = \{\mathbf{I}^{(3)}, \mathbf{I}^{(4)}\}$ *means that the interventions targets that contain perfect interventions on $V_2$.*

$\Psi_{\{\}} = \Psi$. *Also, **the mechanism of $V_1^{\Pi_1,\{1\}}$ is different from the mechanism of $V_2^{\Pi_2,\{1\}}$** since variables are different.* $\qquad\square$

## A.2 Assumptions and Remarks

In this paper, we make a few key assumptions about interventions and the differences in domains. We leverage many similar assumptions to the setting proposed in the literature related to causal representation learning, and handling of multiple domains and interventions [21, 22, 32, 51]. We discuss those assumptions and their implications here.

**Remark 1** (Mixing is invertible). As a consequence of Def. 2.1, the mixing function $f_{\mathbf{X}}$ is invertible, ensuring that latent variables are uniquely learnable [9, 10, 17, 64].

The mapping from generative factors $\mathbf{V}$ to high dimensional mixture $\mathbf{X}$ is a one-to-one mapping. Consider images. In one direction, $\mathbf{V}$ constructs the image through a mixing tool $f_{\mathbf{X}}$ (such as a camera lens). In the reverse direction, these generative factors $\mathbf{V}$ can be uniquely labeled through $f_{\mathbf{X}}^{-1}$. We take images example in Sec. D.1 as an example. The generative factors $Gender$, $Age$ and $Haircolor$ are directly expressed through pixels in images. Given an image, the values of these generative factors are uniquely determined. This assumption is commonly used in non-linear ICA and representation learning literature [9, 10, 17, 64].

**Remark 2** (Confounders are not part of the mixing function). According to Def. 2.1, latent exogenous variables $\mathbf{U}$ influence the high-dimensional mixture $\mathbf{X}$ only through latent causal variables $\mathbf{V}$, so unobserved confounding $\mathbf{U}$ does not directly affect the mixing function.

An example of when this can occur in the real world is when modeling high-dimensional T1 MRI scans. Let the LCG comprise of Drug Treatment $\rightarrow$ Outcome, but they are confounded by socioeconomic status (Drug Treatment $\leftrightarrow$ Outcome). The drug treatment and outcome are assumed to be visually discernable on the MRI. However, socioeconomic status does not directly impact how the MRI appears, except through how it impacts the drug treatment efficacy or outcome. In addition, in EEG data, sleep quality and drug treatment may influence EEG appearance, while socioeconomic status may confound sleep and drug treatment but not directly affect EEG. This idea is also present in prior work, such as nonlinear ICA, where independent exogenous variables $U_i$ each point to a single $V_i$. [7].

**Remark 3** (Shared causal structure). As a consequence of Def. 2.2, each environment's ASCM shares the same latent causal graph, with no structural changes among latent variables [16].

This means that the S-nodes will not represent structural changes such as when $V_i$ has a different parent set across domains [17].

---

[16]The assumption that there are no structural changes between domains can be relaxed and is considered in the context of inference when causal variables are fully observed, as discussed in [33]. This is an interesting topic for future explorations, and we do not consider this avenue here.

[17]The assumption that there are no structural changes between domains can be relaxed and is considered in the context of inference, as discussed in [33]. This is an interesting topic for future explorations, and we do not consider this avenue here.

**Remark 4** (Mixing function is shared across all domains). By Def. 2.1, the mixing function $f_{\mathbf{X}}$ is the same for all ASCMs $\mathcal{M}^i \in \mathcal{M}$, enabling cross-domain analysis. If the mixing function varied across distributions, the latent representations would not be identifiable from iid data alone [9, 51]. □

Sharing of the mixing function is needed for the multi-domain setting because if everything may change across environments, the domains can only be analysed in isolation, and thus unable to leverage the changes (and similarities) across domains.

**Assumptions for Interventions**   We discuss assumptions related to interventions here.

**Assumption 1** (Soft interventions without altering the causal structure). *Interventions do not change the causal diagram. Hard interventions cut all incoming parent edges, and soft interventions preserve them [59]. However, more general interventions may arbitrarily change the parent set for any given node [59]. We do not consider such interventions and leave this general case for future work.*

This assumption precludes any soft interventions that modify the graphical structure of the causal diagram. This work does allow both perfect interventions that cut all incoming parent edges, and soft interventions that preserve all parent edges. However, more general interventions may arbitrarily change the parent set for any given node [59]. We do not consider such interventions and leave this general case for future work. Note Assumption 1 does not mean that interventions cannot occur with the same mechanism across domains. For example, consider two hospitals $\Pi^1$ and $\Pi^2$. Treating epilepsy in each of these hospitals can have outcomes that differ vastly due to the differences in domains [38, 39, 41]. This is represented graphically in $G^S$ with $S^{1,2} \to$ outcome. However, if a neurologist who controls every aspect of his treatment procedure treats patients in both hospitals herself for the purposes of an experiment, then the outcomes will not differ in distribution. This is represented graphically as $S^{1,2} \not\to$ outcome with the S-node being removed from the "outcome" variable. Thus if a pair of interventions occurring in different domains are deemed to have the same mechanism, then the S-node (if one is pointing to the intervened variable) is removed when comparing these two distributions.

Another assumption we make is that all interventions have known targets.

**Assumption 2** (Known-target interventions). *All interventions occur with known targets, reducing permutation indeterminacy for intervened variables.*

That is, for each interventional distribution we have, we know the interventions that occurred and at which node(s) they occurred. This assumption allows us to reduce the permutation indeterminacy that would arise if we did not know the intervention targets. In this work, we also are not concerned with permutation indeterminacy for variables we do not necessarily intervene on because we will mostly be concerned with disentanglement wrt the intervened variables (see Appendix Section A.4). It would be interesting for future work to consider unknown intervention targets.

**Assumptions for Distributions**   In Sec. 2, we discuss that each distribution resulting from an intervention is sufficiently distinct from another distribution Assumption 4. Here we formally define and illustrate what is "change sufficiently".

**Assumption 4** (Changing Sufficiently). *Consider a collection of ASCMs $\mathcal{M}$ and a set of distribution $\mathcal{P}$ induced by $\mathcal{M}$ from a collection of interventions $\Sigma$. Let the LSD induced by $\mathcal{M}$ be $G^S$. Let $\mathcal{P}_{\mathbf{T}} = \{P^{(a_0)}, P^{(a_1)}, \ldots, P^{(a_L)}\} \subseteq \mathcal{P}$ be any collection of distributions such that $\mathbf{T} = do[\mathbf{I}^{(a_0)}] \subseteq do[\mathbf{I}^{(a_l)}]$ for $l \in [L]$, meaning for the baseline distribution all perfect interventions must be exactly on $\mathbf{T}$, and all other distributions must at least contain $\mathbf{T}$ in their perfect interventions. Let $\mathbf{Q} = \bigcup_{l \in [L]} \Delta\mathbf{Q}[\mathbf{I}^{(a_l)}, \mathbf{I}^{(0)}, \mathbf{T}, G^S]$ (Def. 3.1). It is assumed:*

1. *The probability density function of $\mathbf{V}$ is smooth and positive, i.e. $p_{\mathbf{T}}^{(a_l)}(\mathbf{v})$ is smooth and $p_{\mathbf{T}}^{(a_l)}(\mathbf{v}) > 0$ almost everywhere.*

2. *First-order discrepancy. If there exists $\{a_1', \ldots, a_{|\mathbf{Q}|}'\} \subseteq \{a_1, \ldots, a_L\}$ such that $\forall \ V_q \in \mathbf{Q}, V_q \in \Delta\mathbf{Q}[\mathbf{I}^{(a_q')}, \mathbf{I}^{(a_0)}, \mathbf{T}, G^S]$, then $\{\boldsymbol{\omega}_1(\mathbf{v}, a_1), \boldsymbol{\omega}_1(\mathbf{v}, a_2), \ldots, \boldsymbol{\omega}_1(\mathbf{v}, a_L)\}$ are linearly independent, where*

$$\boldsymbol{\omega}_1(\mathbf{v}, a_l) = \left( \oplus \left( \frac{\partial \log p_{\mathbf{T}}^{(a_l)}(\mathbf{v}) - \log p_{\mathbf{T}}^{(a_0)}(\mathbf{v})}{\partial v_q} \right)_{V_q \in \mathbf{Q}} \right) \tag{5}$$

3. *Second-order discrepancy. Let a set $\boldsymbol{\mathcal{E}}$ consist of pairs of $(V_p, V_q)$ such that $(V_p, V_q)$ appears at least in one $\Delta\mathbf{Q}$ and $V_p$ is connected with $V_q$ conditioning on $\mathbf{V}\backslash\{V_p, V_q\}$ in $G_{\overline{T}}(\mathbf{Q})$. Namely,*

$$\boldsymbol{\mathcal{E}} = \{\epsilon_j = \{V_k, V_r\} \mid$$
$$\text{(i) } \exists a_l, \{V_p, V_q\} \in \Delta\mathbf{Q}^{(a_l),(a_0)}; \tag{6}$$
$$\text{(ii) } V_p \text{ is d-connected to } V_q \text{ conditioned on } \mathbf{V}^{tar}\backslash\{V_p, V_q\} \text{ in } G_{\overline{\mathbf{T}}}(\mathbf{V}^{tar})\},$$

*If there exists $\{a'_1, \ldots, a'_{2|\mathbf{Q}|+|\boldsymbol{\mathcal{E}}|}\} \in \{a_1, \ldots, a_L\}$ such that $\forall \ V_q \in \mathbf{Q}, V_q \in \Delta\mathbf{Q}^{(a'_i),(a_0)}], V_q \in \Delta\mathbf{Q}^{(a'_{|Q|+i}),(a_0)}$ and for all $\epsilon_j \in \boldsymbol{\mathcal{E}}, \epsilon_j \subseteq \Delta\mathbf{Q}^{(a'_{2|\mathbf{Q}|+j}),(a_0)}$, then $\{\boldsymbol{\omega}_2(\mathbf{v}, a_1), \boldsymbol{\omega}_2(\mathbf{v}, a_2), \ldots, \boldsymbol{\omega}_2(\mathbf{v}, a_L)\}$ are linearly independent, where*

$$\boldsymbol{\omega}_2(\mathbf{v}, a_l) = \left( \oplus \left( \frac{\partial \log p_{\mathbf{T}}^{(a_l)}(\mathbf{v}) - \log p_{\mathbf{T}}^{(a_0)}(\mathbf{v})}{\partial v_q} \right)_{V_q \in \mathbf{Q}},\right.$$
$$\oplus \left( \frac{\partial^2 \log p_{\mathbf{T}}^{(a_l)}(\mathbf{v}) - \log p_{\mathbf{T}}^{(a_0)}(\mathbf{v})}{\partial v_q^2} \right)_{V_q \in \mathbf{Q}},$$
$$\left. \oplus \left( \frac{\partial^2 \log p_{\mathbf{T}}^{(a_l)}(\mathbf{v}) - \log p_{\mathbf{T}}^{(a_0)}(\mathbf{v})}{\partial v_p v_q} \right)_{(V_p, V_q) \in \mathcal{E}(G_{\overline{T}}(\mathbf{Q}))} \right) \tag{7}$$

$\square$

At a high level, this assumption will be naturally satisfied if the ASCMs and interventions are randomly chosen and only will be violated if the probability density of $P^{(j)}$ and $P^{(k)}$ are fine-tuned to each other [51]. This kind of assumption is generally included in the causal representation learning literature, such as the "genericity" assumption [51], the "interventional discrepancy" assumption [21], and the "sufficient changes" assumption [10, 22].

To illustrate, the assumptions contain two linear independence constraints. Specifically, the first-order and second-order partial derivatives of the log discrepancy from $P^{(a_l)}$ to $P^{(a_0)}$ should be independent of each other. Specifically, The two conditions are made because of necessity, since the linear independence constraints can hold only if these conditions hold. The following example illustrates the necessity of the first-order condition:

**Example 8** (Distributions do not change sufficiently). *Consider $\Delta\mathbf{Q}$ obtained after comparisons as*

$$\Delta\mathbf{Q}^{(1),(0)} = \{V_1\}, \Delta\mathbf{Q}^{(2),(0)} = \{V_1\}, \Delta\mathbf{Q}^{(1),(0)} = \{V_1, V_2, V_3\}, \tag{8}$$

*Let $\mathbf{Q} = \{V_1, V_2, V_3\}$. We have*

$$\frac{\log p_{\mathbf{T}}^{(1)}(\mathbf{v}) - \log p_{\mathbf{T}}^{(0)}(\mathbf{v})}{\partial v_2} = 0 \tag{9}$$

*Since $V_2 \notin \Delta\mathbf{Q}^{(1),(0)}$. Similarly, we know*

$$\boldsymbol{\omega}_1(v_1, v_2, v_3, 1) = \left( \frac{\partial \log p_{\mathbf{T}}^{(1)}(\mathbf{v}) - \log p_{\mathbf{T}}^{(0)}(\mathbf{v})}{\partial v_1}, 0, 0 \right)$$
$$\boldsymbol{\omega}_1(v_1, v_2, v_3, 2) = \left( \frac{\partial \log p_{\mathbf{T}}^{(2)}(\mathbf{v}) - \log p_{\mathbf{T}}^{(0)}(\mathbf{v})}{\partial v_1}, 0, 0 \right)$$
$$\boldsymbol{\omega}_1(v_1, v_2, v_3, 3) = \left( \frac{\partial \log p_{\mathbf{T}}^{(3)}(\mathbf{v}) - \log p_{\mathbf{T}}^{(0)}(\mathbf{v})}{\partial v_1}, \frac{\partial \log p_{\mathbf{T}}^{(3)}(\mathbf{v}) - \log p_{\mathbf{T}}^{(0)}(\mathbf{v})}{\partial v_2}, \frac{\partial \log p_{\mathbf{T}}^{(3)}(\mathbf{v}) - \log p_{\mathbf{T}}^{(0)}(\mathbf{v})}{\partial v_3} \right)$$
$$\tag{10}$$

*And this implies $\boldsymbol{\omega}_1(v_1, v_2, v_3, 1), \boldsymbol{\omega}_1(v_1, v_2, v_3, 2), \boldsymbol{\omega}_1(v_1, v_2, v_3, 3)$ are for sure not linearly independent.* $\square$

On the other perspective, violating these assumptions is like stating the probability densities are fine-tuned to each other [51]. Here we give an example of how this assumption can be violated.

| Domain | Observational | Interventional | | | |
|---|---|---|---|---|---|
| $\Pi^1$ | $P^1_{\{\}}(\mathbf{X})$ | $P^1_{v_i}(\mathbf{X})$ | $P^1_{v_j}(\mathbf{X})$ | $P^1_{v_i,v_j}(\mathbf{X})$ | $\ldots$ |
| $\Pi^2$ | $P^2_{\{\}}(\mathbf{X})$ | $P^2_{v_i}(\mathbf{X})$ | $P^2_{v_k}(\mathbf{X})$ | $P^2_{v_i,v_k}(\mathbf{X})$ | $\ldots$ |
| $\vdots$ | $\vdots$ | $\vdots$ | $\vdots$ | $\vdots$ | $\vdots$ |
| $\Pi^N$ | $P^N_{\{\}}(\mathbf{X})$ | $P^N_{v_l}(\mathbf{X})$ | $P^N_{v_m}(\mathbf{X})$ | $P^N_{v_l,v_j}(\mathbf{X})$ | $\ldots$ |

Table S1: **Possible distributions observed for any given causal representation learning task** - Each domain $\mathbf{\Pi} = \{\Pi^1, \Pi^2, ..., \Pi^N\}$ may contain observational and interventional distributions over latent variables $\mathbf{V}$, which are mixed via $f_{\mathbf{X}}$ to generate $\mathbf{X} \in \mathbb{R}^m$. The first row and column are studied in the existing literature under the lens of the multi-domain intervention exchangeability assumption [32]. Prior work also requires distributions across the entire column (i.e. many domains must be observed), or entire row (i.e. an intervention per latent variable). This paper discusses a more general disentangled representation learning setting when an arbitrary combination of distributions from interventions and domains can be input (i.e. any combination of cells in yellow, and green).

**Example 9** (Distributions do not change sufficiently). *Consider intervention targets*

$$\mathbf{\Psi} = \{\mathbf{I}^{(1)} = \{\{\}^{\Pi_1}\}, \mathbf{I}^{(2)} = \{V_1^{\Pi_1,\{1\}}\}, \mathbf{I}^{(3)} = \{V_2^{\Pi_1,\{2\}}\}, \mathbf{I}^{(4)} = \{V_1^{\Pi_1,\{1\}}, V_2^{\Pi_1,\{2\}}\}\} \quad (11)$$

*Choosing $\mathbf{I}^{(1)}$ as the baseline, $\mathbf{T} = \{\}$. The corresponding $\Delta\mathbf{Q}$ sets are $\{\{V_1\}, \{V_2\}, \{V_1, V_2\}\}$. Let $\mathbf{Q}$ be the union of $\Delta\mathbf{Q}$ sets, which is $\{V_1, V_2\}$. One can verify*

$$\boldsymbol{\omega}_1(\mathbf{v}, 2) + \boldsymbol{\omega}_1(\mathbf{v}, 3) = \boldsymbol{\omega}_1(\mathbf{v}, 4) \quad (12)$$

*since $\mathbf{I}^{(4)}$ is designed as a combination of $\mathbf{I}^{(2)}$ and $\mathbf{I}^{(3)}$.* □

We provide the following Lemma to justify Assumption 4 formally.

**Lemma 1.** *Assumption 4 almost surely holds.* □

### A.3 Domains vs Interventions

In previous studies, there has been a tendency to conflate the notions of interventions and domain shifts [65–69]. However, it is essential to recognize their distinctiveness, particularly when considering various real-world examples spanning different scientific domains that utilize observational and interventional data. The differentiation between interventions and domains is not only conceptually significant but also holds implications for causal inference and the characterization of corresponding causal structures as noted by [32]. Moreover, it is crucial to avoid conflating these qualitatively distinct concepts of interventions and domains, as highlighted in transportability analysis [62]. Pearl and Bareinboim have introduced clear semantics for (S) nodes (environments), presenting a unified representation in the form of selection diagrams [33, 35, 36].

By recognizing these differences, this work leverages any combination of observational and/or interventional data arising from multiple domains to present a general approach to disentanglement learning compared to prior work (see Table S1). Prior work generally considered either interventions in a single domain (top row in $\Pi^1$), where there must be an intervention per latent variable [14, 21], or observational distributions from many domains $\Pi^1, \Pi^2, ..., \Pi^N$ (first column under "Observational"). However, this paper considers a general setting where we may have an arbitrary collection of interventions, or observations from any combination of domains (green section).

Here, we illustrate some examples of the CRID algorithm using distributions from multiple domains.

**Example 10** (Example illustrating CRID with domains). *Consider the LSD shown in Fig. 4(a). We have the following distributions $\mathcal{P} = \{P^{(1)}, P^{(2)}\} = \{P^{\Pi_1}(\mathbf{X}), P^{\Pi_2}(\mathbf{X})$ from interventions $\Sigma = \{\sigma^{(1)}, \sigma^{(2)}\} = \{\{\}, \{\}\}$. Applying CRID algorithm, we can determine that $V_1$ is ID wrt $V_2$ and $V_3$.* □

This example illustrates that observational data in two domains can help disentangle a root variable ($V_1$) from all its descendants.

**Example 11** (Example illustrating CRID with interventions across domains with different mechanisms)**.** *Consider the LSD shown in Fig. 4(a). We have the following distributions* $\mathcal{P} = \{P^{(1)}, P^{(2)}\} = \{P^{\Pi_1}(\mathbf{X}), P^{\Pi_2}(\mathbf{X})$ *from interventions* $\Sigma = \{\sigma^{(1)}, \sigma^{(2)}\}$ *with targets* $\mathbf{\Psi} = \{\{V_2\}^{\Pi_1}, \{\}^{\Pi_2}$. *Applying CRID algorithm, we can determine that $V_2$ and $V_1$ is ID wrt $V_3$.* $\square$

This example demonstrates that when comparing observational data from domain $\Pi_1$ with interventional data from a different domain $\Pi_2$, the only invariant factor is $P(V_3|V_2)$, with $\Delta V[\{\{V_2\}^{\Pi_1}, \{\}^{\Pi_2}, G^S] = \{V_1, V_2\}$. The canceled variable is $V_3$, and thus we achieve our identifiability result.

**Example 12** (Example illustrating CRID with interventions across domains with the same mechanisms)**.** *Consider the LSD shown in Fig. 4(a). We have the following distributions* $\mathcal{P} = \{P^{(1)}, P^{(2)}, P^{(3)}\}$ *from interventions* $\Sigma = \{\sigma^{(1)}, \sigma^{(2)}, \sigma^{(3)}\}$ *with targets* $\mathbf{\Psi} = \{\{V_1^{[i]}, V_2\}^{\Pi_1}, \{\}^{\Pi_2}, \{V_1^{[i]}\}^{\Pi_2}$. *Applying CRID algorithm, we can determine that $V_1$ is ID wrt $\{V_2, V_3\}$, and $V_2$ is ID wrt $\{V_3\}$.* $\square$

Even with an intervention that changes both $V_1, V_2$. When comparing the distributions $P^{(1)}$ and $P^{(3)}$, the $P(V_1)$ term becomes an invariant factor because the intervention has the same mechanism. This removes the possible difference encoded by the S-node on $V_1$ between domains $\Pi^1, \Pi^2$.

These examples further demonstrates the importance of distinguishing domains and interventions because a difference in mechanism is present when comparing all distributions between a pair of domains, $\Pi_i \neq \Pi_j$. This in principle, results in additional variables in the $\Delta\mathbf{Q}$ set. However, interventions may allow us to remove variables from this set by increasing the number of invariant factors.

### A.4 Permutation Indeterminacy

In the context of causal representation learning, permutation indeterminacy is a significant challenge that arises when attempting to identify latent variables from observed data. This phenomenon occurs when the ordering of latent variables is not uniquely determined, leading to multiple equivalent representations (i.e. permutations of the latent variables) that can explain the observed data equally well.

In the earliest results of disentangled representation learning, linear ICA was known to be identifiable only up to permutation and scaling indeterminacies [6]. Permutation indeterminacy is still present in nonlinear ICA [7], since the independent components may be permuted arbitrarily.

Interestingly, when generalizing the problem to the Markovian setting where latent variables have causal structure (i.e. edges in a causal graph), permutation indeterminacy can be reduced to a graph isomorphism in certain cases. That is, latent variables are exchangeable with other latent variables that preserve the topological ordering of the latent causal graph (rather than permuted with any arbitrary latent variable) [13, 22, 51]. When the interventions occur with known targets on the latent space, and intervention occurs uniquely on every latent variable, then there is no permutation indeterminacy [21].

In this work, we assume intervention targets are known, but do not necessarily occur on all latent variables, and they may occur on multiple variables at once. For variables that are intervened on uniquely (i.e. one intervention applied on only that variable), there is no permutation ambiguity. For variables that are intervened on in groups, or not intervened on at all, there still exists permutation ambiguity:

1. (Grouped variables) These variables are all intervened on in the same group. In the context of our paper, these variables are consistently in the same $\Delta\mathbf{Q}$ set. For example, consider the following LCG $V_1 \rightarrow V_2 \leftarrow V_3$. If we have distributions arising only from interventions on $\{V_1, V_3\}$ and the observational distribution, and assume the learned representation is fully disentangled, then the learned representation still has a permutation indeterminacy wrt $\{V_1, V_3\}$. That is, $\hat{V}_1$ could be the representation for $V_1$, or $V_3$ and similarly for $\hat{V}_3$ (See why permutation can hold for details in Example 18).

2. (Non-intervened variables) These variables do not contain any interventions. Then there is still permutation ambiguity among these variables. However, instead of a graph isomorphism ambiguity, these variables form a subgraph isomorphism problem because there may be other

variables that change across distributions (i.e. via interventions, or changes in domains), which are not permutable with respect to these invariant variables.

Specifically, the identifiability we talk about (Def. 2.3) is considered after a subgraph isomorphism permutation. For example, in the collider example setting where permutation can happen between $V_1$ and $V_3$. The "$V_1$ is ID w.r.t $\{V_2, V_3\}$" should implies there exists a function $\tau$ such that $\pi(\mathbf{V})[V_1] = \tau(\pi(\mathbf{V})[V_1])$, where $\pi(\mathbf{V})[V_i]$ means variable $V_i$ after the permutation on $\mathbf{V}$ and $\pi$ denotes a permutation only in this text. In our paper, we are primarily concerned with disentanglement and determining if the learned representation is disentangled in some general sense, and the permutation part is out of our scope.

## B  CRID Algorithm Details

Here, we provide additional pseudocode for the CRID Alg. 1.

First, the following algorithm illustrates how to initialize a fully connected bipartite graph $G_{\mathbf{V}, \widehat{\mathbf{V}}}$. In the initial $G_{\mathbf{V}, \widehat{\mathbf{V}}}$, the true underlying factors $\mathbf{V}$ points to representations each $\widehat{V}_i \in \widehat{\mathbf{V}}$, which means each variable $V_i \in \mathbf{V}$ is entangled with all other variables.

---

**Algorithm F.2 FullyConnectedBipartiteGraph: Initialization step** - Initialize a fully connected bipartite graph.

---

**Input:** $\mathbf{V}, \widehat{\mathbf{V}}$
**Output:** $G_{\mathbf{V}, \widehat{\mathbf{V}}}$
 1: Initialize an empty graph $G_{\mathbf{V}, \widehat{\mathbf{V}}}$
 2: **for** $V_i$ in $\mathbf{V}$ **do**
 3:     **for** $V_j$ in $\widehat{\mathbf{V}}$ **do**
 4:         Add edge $(V_i, V_j)$ to $G_{\mathbf{V}, \widehat{\mathbf{V}}}$

---

Then, after constructing $Q$ from comparisons of distributions, the Alg. F.3 illustrates the details to check whether $\mathbf{V} \backslash \mathbf{Q}$ can be disentangled from $\mathbf{Q}$ according to Proposition 3. To illustrate, each variable $Z \in \mathbf{V} \backslash \mathbf{Q}$ is checked one by one. The variables that have already been disentangled from $Z$ are collected in the list **Mem** through procedure **CheckMemoize**. Next, check if there is a sub-collection of $\mathcal{Q}$ that satisfy the [1-3] conditions in Proposition 3. The checking procedure is shown in Alg.F.5. If conditions are satisfied the edges from $Z$ to $\widehat{\mathbf{Q}}$ are removed to demonstrate disentanglement. Based on the Lemma 2, the condition [3] in Prop. 3 can be reduced to a weaker condition [4] leveraging existing disentanglements in CDM.

**Lemma 2.** *Consider variables $\mathbf{V}^{tar} \subseteq \mathbf{V}$ and $Z \in \mathbf{V} \backslash \mathbf{V}^{tar}$. Suppose $\mathbf{Mem} = \{V_j \in \mathbf{V}^{tar} \mid V_j \text{ is ID w.r.t. } Z\}$. Consider, $\mathcal{P}_{\mathbf{T}}$ and its corresponding intervention targets that hold conditions [1-2] in Prop. 3. If the new version of the condition [3] is also satisfied:*

*[4] there exists $\{a'_1, \ldots, a'_{|\mathbf{V}^{tar}|}\} \subseteq \{a_1, \ldots, a_L\}$ such that for all $V_i^{tar} \in \mathbf{V}^{tar} \backslash \mathbf{Mem}, V_i^{tar} \in \Delta\mathbf{Q}[\mathbf{I}^{(a'_i)}, \mathbf{I}^{(a_0)}, \mathbf{T}, G^S]$.*

*then $\mathbf{V}^{tar}$ is ID w.r.t $Z$.* $\qquad\square$

To illustrate, the above lemma indicates not all variables in $\mathbf{V}^{tar}$ needed to be covered uniquely. Variables that have been already disentangled (in **Mem**) do not need to be considered.

**Example 13.** *Consider the LSD $G^S$ and intervention targets $\mathbf{I}^{(1)} = \{\}$ and $\mathbf{I}^{(4)} = \{V_2^{\Pi_1, do}\}$. Comparing $\mathbf{I}^{(4)}$ and $\mathbf{I}^{(1)}$ taking $\mathbf{T} = \{\}$, $\Delta\mathbf{Q} = \{V_1, V_2\}$. Based on Prop. 3, we cannot get $V_2$ is ID w.r.t $V_3$ since to cover $V_1$ and $V_2$ separately, at least two $\Delta\mathbf{Q}$ sets are needed.*

*Now assume it is known that $V_1$ is ID w.r.t. $V_3$, namely $\mathbf{Mem} = \{V_1\}$. $\Delta\mathbf{Q}$ sets only need to cover $V_2$ and does not need to cover $V_1$ from condition [4] in Lemma 2. Then $V_2$ is ID w.r.t. $V_3$.* $\qquad\square$

**Algorithm F.3 Dis△QfromCancel** - Check whether canceled variables $\mathbf{V}\backslash\mathbf{Q}$ can be disentangled from the LQ factors $\mathbf{Q}$. $G_{\mathbf{V},\widehat{\mathbf{V}}}$ is the current bipartite graph; $G_{\overline{\mathbf{T}}}$ is the LCG after the perfect intervention on $\mathbf{T}$; $\mathbf{\Psi}_{\mathbf{X}}^{\mathrm{perf}}$ is the intervened sets that contains perfect interventions on $\mathbf{X}$; $\mathbf{I} \in \mathbf{\Psi}_{\mathbf{T}}^{\mathrm{perf}}$ is the chosen baseline distribution; $\mathcal{Q}$ is the collection of $\Delta\mathbf{Q}$ sets after comparing intervention targets $\mathbf{J} \in \mathbf{\Psi}_{\mathbf{X}}^{\mathrm{perf}}\backslash\mathbf{I}$ with the baseline.

**Input:** $\mathbf{Q}, G_{\mathbf{V},\widehat{\mathbf{V}}}, G_{\overline{\mathbf{X}}}, \mathbf{\Psi}_{\mathbf{X}}^{\mathrm{perf}}, \mathbf{I}, \mathcal{Q}$
**Output:** $G_{\mathbf{V},\widehat{\mathbf{V}}}$
1: **for** all $Z \in \mathbf{V}\backslash\mathbf{Q}$ **do**
2:     $\mathbf{Mem} \leftarrow CheckMemoize(G_{\mathbf{V},\widehat{\mathbf{V}}}, Z, \mathbf{Q})$    ▷ Variables in $\mathbf{Q}$ has been already ID w.r.t. $Z$.
3:     **if** $CheckConsition3(\mathcal{Q}, \mathbf{Q}, \mathbf{Mem})$ **then**    ▷ Check conditions in Prop. 3 and Lem. 3
4:         remove edge $Z \to \widehat{\mathbf{Q}}$ in $G_{\mathbf{V},\widehat{\mathbf{V}}}$
5: **return** $G_{\mathbf{V},\widehat{\mathbf{V}}}$

---

**Algorithm F.4 CheckMemoize: Memoization step** - The variables in $\mathbf{Q}$ is ID w.r.t $Z$ already.

**Input:** $G_{V,\hat{V}}, Z, \mathbf{Q}$
**Output:** $\mathbf{Mem}$
1: $\mathbf{Mem} \leftarrow \{\}$
2: **for** all $\widehat{V} \in \mathbf{Q}$ **do**
3:     **if** $Z \to \widehat{V} \notin G_{\mathbf{V},\widehat{\mathbf{V}}}$ **then**
4:         $\mathbf{Mem}.append(V)$
5: **return** $\mathbf{Mem}$

---

**Algorithm F.5 CheckCondition3**: Check conditions in Proposition 3 and Lemma 2. $\mathcal{Q}$ is the collection of $\Delta\mathbf{Q}$ sets; $\mathbf{Q}$ are target variables; $\mathbf{Mem}$ are variables in $\mathbf{Q}$ have already been disentangled.

**Input:** $\mathcal{Q}, \mathbf{Q}, \mathbf{Mem}$
**Output:** $True$ or $False$
1: $\mathbf{L} \leftarrow \{\}$
2: **for** $\mathbf{Q}_k \in \mathcal{Q}$ **do**
3:     **if** $\mathbf{Q}_k \subseteq \mathbf{Q}$ **then**
4:         $\mathbf{L}.append(\mathbf{Q}_k)$
5: $\mathbf{Q}^{re} = \{Q_1, \dots, Q_{d'}\} \leftarrow \mathbf{Q}\backslash\mathbf{Mem}, d' \leftarrow |\mathbf{Q}^{re}|$
6: **if** $Q_1 \in \mathbf{L}_1, Q_2 \in \mathbf{L}_2, \dots, Q_{d'} \in \mathbf{L}_{d'}$ after a permutation of $\mathbf{L}$ **then**
7:     **return** $True$
8: **return** $False$

**Algorithm F.6 DisWithin$\Delta$Q** - Check the disentanglement of variables within $\mathbf{Q}$. $G_{\mathbf{V},\widehat{\mathbf{V}}}$ is the current bipartite graph; $G_{\overline{\mathbf{T}}}$ is the LCG after the perfect intervention on $\mathbf{T}$; $\Psi_{\mathbf{T}}^{\text{perf}}$ is the intervened sets that contains perfect interventions on $\mathbf{X}$; $\mathbf{I} \in \Psi_{\mathbf{T}}^{\text{perf}}$ is the chosen baseline distribution; $\mathcal{Q}$ is the collection of $\Delta\mathbf{Q}$ sets after comparing intervention targets $\mathbf{J} \in \Psi_{\mathbf{X}}^{\text{perf}} \backslash \mathbf{I}$ with the baseline.

---

**Input:** $\mathbf{Q}, G_{\mathbf{V},\widehat{\mathbf{V}}}, G_{\overline{\mathbf{T}}}, \Psi_{\mathbf{T}}^{\text{perf}}, \mathbf{I}, \mathcal{Q}$
**Output:** $G_{\mathbf{V},\widehat{\mathbf{V}}}$
1: **for** for all pair $V_i, V_j \in \mathbf{Q}$ **do**
2:     **if** $V_i \perp V_j \mid \mathbf{Q} \backslash \{V_i, V_j\}$ **then**
3:         $\mathbf{Mem}_i \leftarrow CheckMemoize(G_{\mathbf{V},\widehat{\mathbf{V}}}, V_i, \mathbf{Q})$       $\triangleright$ Variables in $\mathbf{Q}$ is ID w.r.t $V_i$ already.
4:         $\mathbf{Mem}_j \leftarrow CheckMemoize(G_{\mathbf{V},\widehat{\mathbf{V}}}, V_j, \mathbf{Q})$       $\triangleright$ Variables in $\mathbf{Q}$ is ID w.r.t $V_j$ already.
5:         **if** $CheckConsition4(\mathcal{Q}, \mathbf{Q}, \mathbf{Mem}_i, \mathbf{Mem}_j, G_{\overline{\mathbf{T}}})$ **then**   $\triangleright$ Check conditions in Prop. 4 and Lem. 3
6:             remove edge $Z \to \widehat{\mathbf{Q}}$ in $G_{\mathbf{V},\widehat{\mathbf{V}}}$
7: **return** $G_{\mathbf{V},\widehat{\mathbf{V}}}$

---

Next, the Alg. F.6 illustrates the details to check whether $V_i, V_j \in \mathbf{Q}$ such that $V_i$ and $V_j$ are independent of each other conditioning on other variables in $\mathbf{Q}$ can be disentangled according to Proposition 4. To illustrate, two lists of variables that have already been disentangled from $V_i$ and $V_j$ are constructed as $\mathbf{Mem}_i$ and $\mathbf{Mem}_j$ respectively through **CheckMemoize**. Next, check if there is a sub-collection of $\mathcal{Q}$ that satisfy the [1-3] conditions in Proposition 3. The checking procedure is shown in Alg.F.7. If conditions are satisfied the edges from $Z$ to $\widehat{\mathbf{Q}}$ are removed to demonstrate disentanglement. Based on the Lem. 3, the condition [3'] in Prop. 4 can be reduced to a weaker condition [4'] leveraging existing disentanglements in CDM.

**Lemma 3** (**ID of variables within $\Delta$Q sets**). *Consider variables $\mathbf{V}^{tar} \subseteq \mathbf{V}$. For any pair of $V_i, V_j \in \mathbf{V}^{tar}$ such that $V_i \perp\!\!\!\perp V_j | \mathbf{V}^{tar} \backslash \{V_i, V_j\}$ in $G_{\overline{\mathbf{T}}}(\mathbf{V}^{tar})$, let $\mathbf{Mem}_i$ be a list of variables in $\mathbf{Q}$ that have been ID w.r.t. $V_i$ and let $\mathbf{Mem}_j$ be a list of qvariables in $\mathbf{Q}$ that have been ID w.r.t. $V_j$. If there exists $\mathcal{P}_{\mathbf{T}}$ that satisfies conditions [1-2] in Prop. 3 and the following condition [4'].*

> *[4'] (Enough changes occur across distributions) Let $\mathbf{Q}^{re} = \mathbf{V}^{tar} \backslash (\mathbf{Mem}_i \bigcup \mathbf{Mem}_j)$ and $d' = |\mathbf{Q}^{re}|$. And*

$$\boldsymbol{\mathcal{E}}_{ij} = \{ \epsilon_j = \{V_k, V_r\} \mid i) \; \exists a_l, \{V_k, V_r\} \in \Delta\mathbf{Q}^{(a_l),(a_0)};$$
$$ii) \; V_k \text{ is connected to} V_r \text{ conditioning } \mathbf{V}^{tar} \backslash \{V_k, V_r\} \text{ in } G_{\overline{\mathbf{T}}}(\mathbf{V}^{tar}) \quad (13)$$
$$iii) \; V_k, V_r \notin \mathbf{Mem}_i \cup \mathbf{Mem}_j \}$$

> *there exists $\{a'_1, \ldots, a'_{2d'+|\boldsymbol{\mathcal{E}}|}\} \in \{a_1, \ldots, a_L\}$ such that for all $Q_i \in \mathbf{Q}^{re}, Q_i \in \Delta\mathbf{Q}^{(a'_i),(a_0)}], Q_i \in \Delta\mathbf{Q}^{(a'_{d'+i}),(a_0)}$ and for all $\epsilon_l \in \boldsymbol{\mathcal{E}}_{ij}, \epsilon_l \subseteq \Delta\mathbf{Q}^{(a'_{2d'+l}),(a_0)}$.*

*, then $V_i$ is ID w.r.t $V_j$.* $\hfill\square$

**Algorithm F.7 CheckCondition4**: Check conditions in Proposition 4 and 3. $\mathcal{Q}$ is the collection of $\Delta\mathbf{Q}$ sets; $\mathbf{Q}$ are target variables;$\mathbf{Mem}_i$ are variables in $\mathbf{Q}$ have already been disentangled with $V_i$;$\mathbf{Mem}_j$ are variables in $\mathbf{Q}$ have already been disentangled with $V_j$; $G_{\overline{\mathbf{T}}}$ is the diagram after removing incoming edge to $\mathbf{T}$.

---

**Input:** $\mathcal{Q}, \mathbf{Q}, \mathbf{Mem}_i, \mathbf{Mem}_j, G_{\overline{\mathbf{T}}}$
**Output:** $True$ or $False$
 1: $\mathbf{L} \leftarrow \{\}$
 2: **for** $\mathbf{Q}_k \in \mathcal{Q}$ **do**
 3:     **if** $\mathbf{Q}_k \subseteq \mathbf{Q}$ **then**
 4:         $\mathbf{L}.append(\mathbf{Q}_k)$
 5: $\mathcal{E} \leftarrow \{\}$
 6: **for** $\{V_k, V_r\} \subseteq \mathbf{Q}$ **do**
 7:     **if** (i) $\exists L \in \mathbf{L}$ such that $\{V_k, V_r\} \subseteq L$ (ii) $V_k$ is conditionally connected to $V_l$ (iii) $\{V_k, V_r\} \nsubseteq$ $\mathbf{Mem}_i \cup \mathbf{Mem}_j$ **then**
 8:         $\mathcal{E}.append((V_k, V_r))$             ▷ Construct $\mathcal{E}$ according to Lem. 3
 9: $\mathbf{Q}^{re+} = \{Q_1, \ldots, Q_{d'}\} \leftarrow (\mathbf{Q}\backslash(\mathbf{Mem}_i \cup \mathbf{Mem}_j)) \cup \mathcal{E}, d^+ \leftarrow |\mathbf{Q}^{re}|$
10: **if** $Q_1 \in \mathbf{L}_1, Q_2 \in \mathbf{L}_2, \ldots, Q_{d'} \in \mathbf{L}_{d'}$ after a permutation of $\mathbf{L}$ **then**
11:     **return** $True$
12: **return** $False$

---

Lastly, we leverage the independence and current disentangled results stored in $G_{\mathbf{V}, \widehat{\mathbf{V}}}$. Canceled variables with $\mathbf{V}\backslash\mathbf{Q}$ can be disentangled with each other according to Proposition 5. The following algorithm illustrates this step.

---

**Algorithm F.8 Dis$\Delta$QFromCancel** - Disentangle canceled variables from $\Delta\mathbf{Q}$. $G_{\mathbf{V}, \widehat{\mathbf{V}}}$ is the current bipartite graph; $G_{\overline{\mathbf{T}}}$ is the LCG after the perfect intervention on $\mathbf{T}$.

---

**Input:** $\mathbf{Q}, G_{\mathbf{V}, \widehat{\mathbf{V}}}, G_{\overline{\mathbf{X}}}$
**Output:** $G_{\mathbf{V}, \widehat{\mathbf{V}}}$
 1: **for** for all $Z$ such that $Z \perp \mathbf{V}\backslash Z$ in $G_{\overline{\mathbf{T}}}$ **do**
 2:     **if** there are no edges from $\mathbf{V}\backslash Z$ to $\mathbf{Z}$ **then**
 3:         remove edges from $Z$ to $\mathbf{V}\backslash Z$
 4: **return** $G_{\mathbf{V}, \widehat{\mathbf{V}}}$

---

# C Proofs

Here, we provide detailed proofs of theoretical results in the main paper.

## C.1 "Distribution Change Sufficiently" - Proof of Lemma 1

We assume "distributions changes sufficiently" in Sec. 2. This assumption is formally defined in Assumption 4 and will be used as a technique assumption in the proof of propositions in this work. Lemma 1 provides the justification of this assumption. It suggests Assumption 4 almost surely holds. We first provide proof here.

**Assumption 4** (Changing Sufficiently). *Consider a collection of ASCMs $\mathcal{M}$ and a set of distribution $\mathcal{P}$ induced by $\mathcal{M}$ from a collection of interventions $\Sigma$. Let the LSD induced by $\mathcal{M}$ be $G^S$. Let $\mathcal{P}_{\mathbf{T}} = \{P^{(a_0)}, P^{(a_1)}, \ldots, P^{(a_L)}\} \subseteq \mathcal{P}$ be any collection of distributions such that $\mathbf{T} = do[\mathbf{I}^{(a_0)}] \subseteq do[\mathbf{I}^{(a_l)}]$ for $l \in [L]$, meaning for the baseline distribution all perfect interventions must be exactly on $\mathbf{T}$, and all other distributions must at least contain $\mathbf{T}$ in their perfect interventions. Let $\mathbf{Q} = \bigcup_{l \in [L]} \Delta\mathbf{Q}[\mathbf{I}^{(a_l)}, \mathbf{I}^{(0)}, \mathbf{T}, G^S]$ (Def. 3.1). It is assumed:*

1. *The probability density function of $\mathbf{V}$ is smooth and positive, i.e. $p_{\mathbf{T}}^{(a_l)}(\mathbf{v})$ is smooth and $p_{\mathbf{T}}^{(a_l)}(\mathbf{v}) > 0$ almost everywhere.*

2. *First-order discrepancy. If there exists $\{a'_1, \ldots, a'_{|\mathbf{Q}|}\} \subseteq \{a_1, \ldots, a_L\}$ such that $\forall \; V_q \in \mathbf{Q}, V_q \in \Delta\mathbf{Q}[\mathbf{I}^{(a'_q)}, \mathbf{I}^{(a_0)}, \mathbf{T}, G^S]$, then $\{\boldsymbol{\omega}_1(\mathbf{v}, a_1), \boldsymbol{\omega}_1(\mathbf{v}, a_2), \ldots, \boldsymbol{\omega}_1(\mathbf{v}, a_L)\}$ are linearly independent, where*

$$\boldsymbol{\omega}_1(\mathbf{v}, a_l) = \left( \oplus \left( \frac{\partial \log p_{\mathbf{T}}^{(a_l)}(\mathbf{v}) - \log p_{\mathbf{T}}^{(a_0)}(\mathbf{v})}{\partial v_q} \right)_{V_q \in \mathbf{Q}} \right) \tag{5}$$

3. *Second-order discrepancy. Let a set $\mathcal{E}$ consist of pairs of $(V_p, V_q)$ such that $(V_p, V_q)$ appears at least in one $\Delta\mathbf{Q}$ and $V_p$ is connected with $V_q$ conditioning on $\mathbf{V} \backslash \{V_p, V_q\}$ in $G_{\overline{\mathbf{T}}}(\mathbf{Q})$. Namely,*

$$\mathcal{E} = \{\epsilon_j = \{V_k, V_r\} \mid$$
$$\text{(i)} \; \exists a_l, \{V_p, V_q\} \in \Delta\mathbf{Q}^{(a_l),(a_0)}; \tag{6}$$
$$\text{(ii)} \; V_p \text{ is d-connected to } V_q \text{ conditioned on } \mathbf{V}^{tar} \backslash \{V_p, V_q\} \text{ in } G_{\overline{\mathbf{T}}}(\mathbf{V}^{tar})\},$$

*If there exists $\{a'_1, \ldots, a'_{2|\mathbf{Q}|+|\mathcal{E}|}\} \in \{a_1, \ldots, a_L\}$ such that $\forall \; V_q \in \mathbf{Q}, V_q \in \Delta\mathbf{Q}^{(a'_i),(a_0)}], V_q \in \Delta\mathbf{Q}^{(a'_{|Q|+i}),(a_0)}$ and for all $\epsilon_j \in \mathcal{E}, \epsilon_j \subseteq \Delta\mathbf{Q}^{(a'_{2|\mathbf{Q}|+j}),(a_0)}$, then $\{\boldsymbol{\omega}_2(\mathbf{v}, a_1), \boldsymbol{\omega}_2(\mathbf{v}, a_2), \ldots, \boldsymbol{\omega}_2(\mathbf{v}, a_L)\}$ are linearly independent, where*

$$\boldsymbol{\omega}_2(\mathbf{v}, a_l) = \left( \oplus \left( \frac{\partial \log p_{\mathbf{T}}^{(a_l)}(\mathbf{v}) - \log p_{\mathbf{T}}^{(a_0)}(\mathbf{v})}{\partial v_q} \right)_{V_q \in \mathbf{Q}}, \right.$$
$$\oplus \left( \frac{\partial^2 \log p_{\mathbf{T}}^{(a_l)}(\mathbf{v}) - \log p_{\mathbf{T}}^{(a_0)}(\mathbf{v})}{\partial v_q^2} \right)_{V_q \in \mathbf{Q}},$$
$$\left. \oplus \left( \frac{\partial^2 \log p_{\mathbf{T}}^{(a_l)}(\mathbf{v}) - \log p_{\mathbf{T}}^{(a_0)}(\mathbf{v})}{\partial v_p v_q} \right)_{(V_p, V_q) \in \mathcal{E}(G_{\overline{\mathbf{T}}}(\mathbf{Q}))} \right) \tag{7}$$

$\square$

**Lemma 1.** *Assumption 4 almost surely holds.* $\square$

*Proof.* We will prove the first-order discrepancy and second-order discrepancy almost surely hold, which means the situations where first-order discrepancy and second-order discrepancy do not hold have Lebesgue measure 0.

We first consider the first-order discrepancy. Denote $\{\boldsymbol{\omega}_1(\mathbf{v}, a_1), \boldsymbol{\omega}_1(\mathbf{v}, a_2), \ldots, \boldsymbol{\omega}_1(\mathbf{v}, a_L)\}$ as $\mathbf{A}$. And every entry in $\mathbf{A}$ is

$$a_{lq} = \frac{\partial \log p_{\mathbf{T}}^{(a_l)}(\mathbf{v}) - \log p_{\mathbf{T}}^{(a_0)}(\mathbf{v})}{\partial v_q} \tag{14}$$

According to Eq. (3), we know $\log p_{\mathbf{T}}^{(a_l)}(\mathbf{v}) - \log p_{\mathbf{T}}^{(a_0)}(\mathbf{v})$ is a function of only variables in $\Delta\mathbf{Q}$. Thus, if $V_q \notin \Delta\mathbf{Q}[\mathbf{I}^{(a_l')}, \mathbf{I}^{(a_0)}, \mathbf{T}, G^S]$, $a_{lq} = 0$; if $V_q \in \Delta\mathbf{Q}[\mathbf{I}^{(a_l')}, \mathbf{I}^{(a_0)}, \mathbf{T}, G^S]$, we assume $a_{lq}$ follows a standard normal distribution, which means the non-zero entries in matrix $\mathbf{A}$ are randomly sampled and are not fine-tuned. Thus, to prove this lemma, it is equivalent to prove if there exists $\{a_1', \ldots, a_{|\mathbf{Q}|}'\} \subseteq \{a_1, \ldots, a_L\}$ such that $\forall\ V_q \in \mathbf{Q}, V_q \in \Delta\mathbf{Q}[\mathbf{I}^{(a_q')}, \mathbf{I}^{(a_0)}, \mathbf{T}, G^S]$, the row of $\mathbf{A}$ are almost surely linear independent. W.O.L.G, we let $\{a_1' = a_1, \ldots, a_{\mathbf{Q}}' = a_L\}$. Then, it is equivalent to prove that $\mathbf{A}$ is a full rank matrix.

In order to prove that $\mathbf{A}$ is a full-rank matrix, we prove that the determinant of $\mathbf{A}$ is almost surely non-zero. Since $\forall\ V_q \in \mathbf{Q}, V_q \in \Delta\mathbf{Q}[\mathbf{I}^{(a_q)}, \mathbf{I}^{(a_0)}, \mathbf{T}, G^S]$, there exists $\mathbf{A}$ such that $\det(\mathbf{A})$ is non-zero, and then $\det(\mathbf{A})$ is non-trivial. Based on a simple algebraic lemma in [70], the subset of $\{\mathbf{A} \mid \det(\mathbf{A}) = 0\}$ of the real space has Lebesgue measure 0. Then $\det(\mathbf{A}) = 0$ almost surely holds.

The second-order discrepancy proof is similar. Denote $\{\boldsymbol{\omega}_2(\mathbf{v}, a_1), \boldsymbol{\omega}_2(\mathbf{v}, a_2), \ldots, \boldsymbol{\omega}_2(\mathbf{v}, a_L)\}$ as $\mathbf{B}$. And every entry of $\mathbf{B}$ is

$$a_{lq} = \frac{\partial \log p_{\mathbf{T}}^{(a_l)}(\mathbf{v}) - \log p_{\mathbf{T}}^{(a_0)}(\mathbf{v})}{\partial v_q}, q \leq |\mathbf{Q}|$$

$$a_{lq} = \frac{\partial^2 \log p_{\mathbf{T}}^{(a_l)}(\mathbf{v}) - \log p_{\mathbf{T}}^{(a_0)}(\mathbf{v})}{\partial v_q^2}, |\mathbf{Q}| + 1 \leq q \leq 2|\mathbf{Q}| \tag{15}$$

$$a_{l\epsilon} = \frac{\partial^2 \log p_{\mathbf{T}}^{(a_l)}(\mathbf{v}) - \log p_{\mathbf{T}}^{(a_0)}(\mathbf{v})}{\partial v_p \partial v_q}, 2|\mathbf{Q}| + 1 \leq \epsilon \leq 2|\mathbf{Q}| + |\boldsymbol{\mathcal{E}}|, \{V_p, V_q\} \in \boldsymbol{\mathcal{E}}$$

According to Eq. (3), we know $\log p_{\mathbf{T}}^{(a_l)}(\mathbf{v}) - \log p_{\mathbf{T}}^{(a_0)}(\mathbf{v})$ is a function of only variables in $\Delta\mathbf{Q}$. Thus, if $V_q \notin \Delta\mathbf{Q}[\mathbf{I}^{(a_l')}, \mathbf{I}^{(a_0)}, \mathbf{T}, G^S]$, $a_{lq} = 0$; if $V_q \in \Delta\mathbf{Q}[\mathbf{I}^{(a_l')}, \mathbf{I}^{(a_0)}, \mathbf{T}, G^S]$, we assume $a_{lq}$ follows a standard normal distribution, which means the non-zero entries in matrix $\mathbf{B}$ are randomly sampled and are not fine-tuned. If $\{V_p, V_q\} \not\subseteq \Delta\mathbf{Q}[\mathbf{I}^{(a_l')}, \mathbf{I}^{(a_0)}, \mathbf{T}, G^S]$, $a_{l\epsilon} = 0$; if $\{V_p, V_q\} \subseteq \Delta\mathbf{Q}[\mathbf{I}^{(a_l')}, \mathbf{I}^{(a_0)}, \mathbf{T}, G^S]$, we assume $a_{l\epsilon}$ follows a standard normal distribution, which means the non-zero entries in matrix $\mathbf{B}$ are randomly sampled and are not fine-tuned. Following the same discussion above, the subset of $\{\mathbf{B} \mid \det(\mathbf{B}) = 0\}$ of the real space has Lebesgue measure 0. Then $\det(\mathbf{B}) = 0$ almost surely holds. $\square$

### C.2 Distribution comparison - Proof of Proposition 1

**Proposition 1 (Distribution Comparison).** *Consider a pair of collections ASCMs $\mathcal{M}$ and $\widehat{\mathcal{M}}$ that matches with the distribution $\mathcal{P}$ resulting from interventions $\Sigma$ and LSD $G^S$. Consider two distributions $P^{\Pi^{(j)}}(\mathbf{X}; \sigma^{(j)})$ and $P^{\Pi^{(k)}}(\mathbf{X}; \sigma^{(k)})$. Suppose $\mathrm{perf}(\mathbf{T})$ is in both intervention sets, then,*

$$\sum_i^d \log p_{\mathbf{T}}^{(j)}(v_i \mid \mathbf{pa}_i^{\mathbf{T}+}) - p_{\mathbf{T}}^{(k)}(v_i \mid \mathbf{pa}_i^{\mathbf{T}+}) = \sum_i^d \log p_{\mathbf{T}}^{(j)}(\widehat{v}_i \mid \widehat{\mathbf{pa}}_i^{\mathbf{T}+}) - \log p_{\mathbf{T}}^{(k)}(\widehat{v}_i \mid \widehat{\mathbf{pa}}_i^{\mathbf{T}+}), \tag{2}$$

*where $p_{\mathbf{T}}^{(j)}(\cdot)$ and $p_{\mathbf{T}}^{(k)}(\cdot)$ are density functions.* $\square$

*Proof.* According to the ASCM definition Def .2.1, the mapping from $\mathbf{V}$ to $\mathbf{X}$, and the mapping $\mathbf{X}$ to $\widehat{\mathbf{V}}$ can be expressed as:

$$\widehat{\mathbf{V}} = \widehat{f}_{\mathbf{X}}^{-1}(\mathbf{X}) = \widehat{f}_{\mathbf{X}}^{-1}(f_{\mathbf{X}}(\mathbf{V})) \tag{16}$$

Then based on the change variable formula, we have

$$p(\mathbf{v}) = p(\widehat{\mathbf{v}})|\mathbf{J}_\phi| \tag{17}$$

where $\phi = \widehat{f}_{\mathbf{X}}^{-1} \circ f_{\mathbf{X}}$ and $\mathbf{J}_\phi$ is the Jacobian matrix of $\phi$. Leveraging the factorization in Eq. 1 and taking log of the above equation,

$$\sum_{i=1}^d \log p_{\mathbf{T}}(v_i \mid \mathbf{pa}^{\mathbf{T}+}) = \sum_{i=1}^d \log p_{\mathbf{T}}(\widehat{v}_i \mid \widehat{\mathbf{pa}}^{\mathbf{T}+}) + \log |\mathbf{J}_\phi| \tag{18}$$

Subtract the above factorization of density function induced by $\mathbf{I}^{(j)}$ and $\mathbf{I}^{(k)}$, and we have Eq.( 2). $\square$

Eq 2 naturally gives a connection from $\mathbf{V}$ to $\widehat{\mathbf{V}}$. Comparing two factorization for Fig. 4(c), the connection connections are made from $P(v_1), p(v_2 \mid v_1), p(v_3 \mid v_2, v_1), P(v_4 \mid v_3)$ or $P(v_1), p(v_3), p(v_2 \mid v_1, v_3), P(v_4 \mid v_3)$.

## C.3 Invariant factors - Proof of Proposition 2

**Proposition 2** (**Invariant Factors**). *Consider two distributions $P^{(j)}, P^{(k)} \in \mathcal{P}$ with intervention targets $\sigma^{(j)}$ and $\sigma^{(k)}$ containing* $\mathrm{perf}(\mathbf{T})$. *Construct the changed variable set $\Delta\mathbf{V}[\mathbf{I}^{(j)}, \mathbf{I}^{(k)}, G^S]$ (for short $\Delta\mathbf{V}$) with target sets $\mathbf{I}^{(j)}, \mathbf{I}^{(k)}$ as follows: (1) $V_l \in \Delta\mathbf{V}$ if $V_l^{\pi_l, \{b_l\}, t_l} \in \mathbf{I}^{(j)}$ but $V_l^{\pi_l', \{b_l\}, t_l'} \notin \mathbf{I}^{(k)}$, or vice versa; (2) $V_l \in \Delta\mathbf{V}$ if i) $S^{\Pi^{(j)}, \Pi^{(k)}}$ point to $V_l$ and ii) $V_l^{\pi_l, \{b_l\}, t_l} \notin \mathbf{I}^{(j)} \cup \mathbf{I}^{(k)}$. If $V_i \in \mathbf{V} \backslash \mathbf{C}(\Delta\mathbf{V})$, then $p_{\mathbf{T}}^{(j)}(v_i \mid \mathbf{pa}_i^{\mathbf{T}+}) = p_{\mathbf{T}}^{(k)}(v_i \mid \mathbf{pa}_i^{\mathbf{T}+})$ (denoted invariant factors).* $\qquad\square$

*Proof.* Consider an arbitrary order. Based on the proposition, $\Delta\mathbf{V}[\mathbf{I}^{(j)}, \mathbf{I}^{(k)}, G^S]$ includes all variables that the mechanism $f_V$ or exogenous $U$ possibly change when the intervention changes from $\mathbf{I}^{(k)}$ to $\mathbf{I}^{(j)}$. In other words, for any $V_l \in \mathbf{V} \backslash \Delta\mathbf{V}[\mathbf{I}^{(j)}, \mathbf{I}^{(k)}, G^S]$, $f_{V_l}$ and exogenous $U_l$ are invariant.

Let $V_i \in \mathbf{V} \backslash \mathbf{C}(\Delta\mathbf{V})$. $\mathbf{Z} = \mathbf{C}(V_i) \cup \mathbf{Pa}^{\mathbf{T}+}$. We have $\mathbf{Z} \subseteq \mathbf{V} \backslash \mathbf{C}(\Delta\mathbf{V})$ according to the definition of C-component.

According to the definition of $\mathbf{Pa}_i^{\mathbf{T}+}$, we know $\mathbf{Pa}_i^{\mathbf{T}+} \backslash \mathbf{Z} = \mathbf{Pa}(\{V_i\} \cup \mathbf{Z})$. Now reconsider the distribution $P^{\Pi^{(j)}}(V_i \mid \mathbf{Pa}_i^{\mathbf{T}+}; \sigma^{(j)})$ and $P^{\Pi^{(k)}}(V_i \mid \mathbf{Pa}_i^{\mathbf{T}+}; \sigma^{(k)})$,

$$P_{\mathbf{T}}^{\Pi^{(j)}}(V_i \mid \mathbf{Pa}_i^{\mathbf{T}+}; \sigma^{(j)}) = P_{\mathbf{T}}^{\Pi^j}(V_i, \mathbf{Z} \mid \mathbf{Pa}_i(\{V_i\} \cup \mathbf{Z}); \sigma^{(j)}) / P_{\mathbf{T}}^{\Pi^{(j)}}(\mathbf{Z} \mid \mathbf{Pa}_i(\{V_i\} \cup \mathbf{Z}); \sigma^{(j)})$$

$$(19)$$

$$P_{\mathbf{T}}^{\Pi^{(k)}}(V_i \mid \mathbf{Pa}_i^{\mathbf{T}+}; \sigma^{(k)}) = P_{\mathbf{T}}^{\Pi^{(k)}}(V_i, \mathbf{Z} \mid \mathbf{Pa}_i(\{V_i\} \cup \mathbf{Z}); \sigma^{(k)}) / P_{\mathbf{T}}^{\Pi^{(k)}}(\mathbf{Z} \mid \mathbf{Pa}_i(\{V_i\} \cup \mathbf{Z}); \sigma^{(k)})$$

$$(20)$$

Since the mechanism and exogenous variables of $V_i$ and $\mathbf{Z}$ are invariant, both the nominators and denominators are the same. Namely,

$$P_{\mathbf{T}}^{\Pi^j}(V_i, \mathbf{Z} \mid \mathbf{Pa}_i(\{V_i\} \cup \mathbf{Z}); \sigma^{(j)}) = P_{\mathbf{T}}^{\Pi^k}(V_i, \mathbf{Z} \mid \mathbf{Pa}_i(\{V_i\} \cup \mathbf{Z}); \sigma^{(k)}) \tag{21}$$

$$P_{\mathbf{T}}^{\Pi^{(j)}}(\mathbf{Z} \mid \mathbf{Pa}_i(\{V_i\} \cup \mathbf{Z}); \sigma^{(j)}) = P_{\mathbf{T}}^{\Pi^{(k)}}(\mathbf{Z} \mid \mathbf{Pa}_i(\{V_i\} \cup \mathbf{Z}); \sigma^{(k)}) \tag{22}$$

which implies the density functions are invariant,

$$p_{\mathbf{T}}^{(j)}(v_i \mid \mathbf{pa}_i^{\mathbf{T}+}) = p_{\mathbf{T}}^{(k)}(v_i \mid \mathbf{pa}_i^{\mathbf{T}+}) \tag{23}$$

$$\square$$

## C.4 ID $\Delta\mathbf{Q}$ w.r.t Canceled Factors - Proof of Proposition 3 and Lemma 2

**Proposition 3** (**ID the $\Delta\mathbf{Q}$ set w.r.t Canceled Variables**). *Consider variables $\mathbf{V}^{tar} \subseteq \mathbf{V}$. Let $\mathcal{P}_{\mathbf{T}} = \{P^{(a_0)}, P^{(a_1)}, \ldots, P^{(a_L)}\} \subseteq \mathcal{P}$ be a collection of distributions such that (1) $\forall \ l \in [L]$, $\mathbf{T} = \mathrm{Perf}[\mathbf{I}^{(a_0)}] \subseteq \mathrm{Perf}[\mathbf{I}^{(a_l)}]$ [18]; (2) $\bigcup_{l \in [L]} \Delta\mathbf{Q}[\mathbf{I}^{(a_l)}, \mathbf{I}^{(a_0)}, \mathbf{T}, G^S] = \mathbf{V}^{tar}$; (3) there exists $\{a_1', \ldots, a_{d'}'\} \subseteq \{a_1, \ldots, a_L\}$ such that for all $V_i^{tar} \in \mathbf{V}^{tar}, V_i^{tar} \in \Delta\mathbf{Q}[\mathbf{I}^{(a_i')}, \mathbf{I}^{(a_0)}, \mathbf{T}, G^S]$, where $d' = |\mathbf{V}^{tar}|$. Then, $\mathbf{V}^{tar}$ is ID w.r.t $\mathbf{V} \backslash \mathbf{V}^{tar}$.* $\qquad\square$

*Proof.* **We denote $\mathbf{V}^{tar}$ as $\mathbf{Q}$ for convenience.** Notice that the Assumption 4 will be used in the proof.

---

[18]Recall we use the notation $\mathrm{Perf}[\mathbf{I}]$ to denote that all variables that have perfect interventions on $\mathbf{I}$.

Comparing $P^{\Pi^{(a_l)}}(\mathbf{V}; \sigma^{(a_l)})$ with $P^{\Pi^{(a_0)}}(\mathbf{V}; \sigma^{(a_0)})$, we have

$$\sum_{V_i \in \hat{\mathbf{V}}} \log p_{\mathbf{T}}^{(a_l)}(v_i \mid \mathbf{pa}_i^{\mathbf{T}+}) - p_{\mathbf{T}}^{a_0)}(v_i \mid \mathbf{pa}_i^{\mathbf{T}+}) = \sum_{V_i \in \hat{\mathbf{V}}} \log p_{\mathbf{T}}^{(a_l)}(\widehat{v}_i \mid \widehat{\mathbf{pa}}_i^{\mathbf{T}+}) - \log p_{\mathbf{T}}^{(a_0)}(\widehat{v}_i \mid \widehat{\mathbf{pa}}_i^{\mathbf{T}+})$$

$$= \log p_{\mathbf{T}}^{(a_l)}(\widehat{\mathbf{v}}) - \log p_{\mathbf{T}}^{(a_0)}(\widehat{\mathbf{v}}) \tag{24}$$

from Eq. (3).

Notice that the left side only involves variables in $\mathbf{Q} = \bigcup_{l \in [L]} \Delta \mathbf{Q}[\mathbf{I}^{(a_l)}, \mathbf{I}^{(a_0)}, \mathbf{T}, G^S]$ based on the Def. 3.1. Thus, for any $Z \in \mathbf{V} \backslash \mathbf{Q}$,

$$\forall l \in [L], \frac{\partial \log p_{\mathbf{T}}^{(a_l)}(\widehat{\mathbf{v}}) - \log p_{\mathbf{T}}^{(a_0)}(\widehat{\mathbf{v}})}{\partial \widehat{z}} = 0 \tag{25}$$

Take partial of the above equation w.r.t. $Z$, we have:

$$\forall l \in [L], 0 = \sum_{v_i \in \mathbf{V}} \frac{\partial \log p_{\mathbf{T}}^{(a_l)}(\widehat{\mathbf{v}}) - \log p_{\mathbf{T}}^{(a_0)}(\widehat{\mathbf{v}})}{\partial \widehat{v}_i} \frac{\partial \widehat{v}_i}{\partial z} \qquad \text{(Chain Rule)} \tag{26}$$

$$= \sum_{v_q \in \mathbf{Q}} \frac{\partial \log p_{\mathbf{T}}^{(a_l)}(\widehat{\mathbf{v}}) - \log p_{\mathbf{T}}^{(a_0)}(\widehat{\mathbf{v}})}{\partial \widehat{v}_q} \frac{\partial \widehat{v}_q}{\partial z} \qquad \text{(Eq.( 25))} \tag{27}$$

Eq. (27) is a linear system for unknowns $\{\partial \widehat{V}_q / \partial Z\}_{V_q \in \mathbf{Q}}$. When distribution changes sufficiently, namely under Assumption 4, the row factor of the coefficient matrix of the linear system is linearly independent. When $L \geq |\mathbf{Q}|$ (implied by condition [3]), the matrix is full rank, thus,

$$\forall V_q \in \mathbf{Q}, \frac{\partial \widehat{v}_q}{\partial z} = 0 \tag{28}$$

Recall that $V_q = \phi_{V_q}(\mathbf{V})$. For any $Z \in \mathbf{V} \backslash \mathbf{Q}$, Eq.( 28) holds. Thus, $\mathbf{Q}$ is enough to be the input of $\phi_{V_q}$, which means there exists $V_q = \phi_{V_q}(\mathbf{Q})$.

$\square$

**Lemma 2.** *Consider variables $\mathbf{V}^{tar} \subseteq \mathbf{V}$ and $Z \in \mathbf{V} \backslash \mathbf{V}^{tar}$. Suppose $\mathbf{Mem} = \{V_j \in \mathbf{V}^{tar} \mid V_j \text{ is ID w.r.t. } Z\}$. Consider, $\mathcal{P}_{\mathbf{T}}$ and its corresponding intervention targets that hold conditions [1-2] in Prop. 3. If the new version of the condition [3] is also satisfied:*

*[4] there exists $\{a'_1, \ldots, a'_{|\mathbf{V}^{tar}|}\} \subseteq \{a_1, \ldots, a_L\}$ such that for all $V_i^{tar} \in \mathbf{V}^{tar} \backslash \mathbf{Mem}, V_i^{tar} \in \Delta \mathbf{Q}[\mathbf{I}^{(a'_i)}, \mathbf{I}^{(a_0)}, \mathbf{T}, G^S]$.*

*then $\mathbf{V}^{tar}$ is ID w.r.t $Z$.* $\square$

*Proof.* For all $V_m \in \mathbf{Mem}$, $\partial V_m / \partial Z = 0$. Thus, the unknown in Eq.( 27) exclude $\frac{\partial v_m}{\partial z}$. Then, when [3'] holds, the system will have zero solutions and Eq.( 28) will hold. $\square$

## C.5 ID within $\Delta$Q set - Proof of Proposition 4 and Lemma 3

The next result provides us with an additional way of disentangling latent variables within the same $\Delta \mathbf{Q}$-factor leveraging second-order conditions and conditional independence. We will first prove the following stronger result.

**Definition 6.1** (Markov Network)**.** Let $M_V$ be the Markov network over variables $\mathbf{V}$ with vertices $\{V_i\}_{i=1}^n$ and $\mathcal{E}(M_V)$ denote the set of edges. An edge $(V_i, V_j)$ is added to $\mathcal{E}(M_V)$ if $V_i \not\perp\!\!\!\perp V_j | \mathbf{V} \backslash \{V_i, V_j\}$. $\square$

**Proposition 6** (**ID of variables within $\Delta Q$ sets**). *Consider the variables $\mathbf{V}^{tar} \subseteq \mathbf{V}$. Define $\mathcal{E}$ as the set of edges within the Markov Network of $G_{\overline{\mathbf{T}}}(\mathbf{V}^{tar})$ that are contained within a $\Delta \mathbf{Q}$ set.*

$$\mathcal{E} = \{\epsilon_j = \{V_k, V_r\} \Big|$$

$$(i) \ \exists a_l, \{V_k, V_r\} \subseteq \Delta \mathbf{Q}^{(a_l),(a_0)};$$

$$(ii) \ V_k \text{ is d-connected to } V_r \text{ conditioned on } \mathbf{V}^{tar}\backslash\{V_k, V_r\} \text{ in } G_{\overline{\mathbf{T}}}(\mathbf{V}^{tar})\}, \tag{29}$$

*For any pair of $V_i, V_j \in \mathbf{V}^{tar}$ such that $V_i \perp\!\!\!\perp V_j \mid \mathbf{V}^{tar}\backslash\{V_i, V_j\}$ in $G_{\overline{\mathbf{T}}}(\mathbf{V}^{tar})$, if there exists $\mathcal{P}_{\mathbf{T}} = \{P^{(a_0)}, P^{(a_1)}, \ldots, P^{(a_L)}\} \subseteq \mathcal{P}$ that satisfies conditions (1-2) in Prop. 3 and the following condition (3').*

*(3') Enough changes occur across distributions, i.e., Formally, there exists $\{a'_1, \ldots, a'_{2d'+|\mathcal{E}|}\} \in \{a_1, \ldots, a_L\}$ such that for all $V_i^{tar} \in \mathbf{V}^{tar}$, i) $V_i^{tar} \in \Delta \mathbf{Q}^{(a'_i),(a_0)}$, ii) $V_i^{tar} \in \Delta \mathbf{Q}^{(a'_{d'+i}),(a_0)}$, and iii) for all $\epsilon_j \in \mathcal{E}, \epsilon_j \subseteq \Delta \mathbf{Q}^{(a'_{2d'+j}),(a_0)}$, where $d' = |\mathbf{V}^{tar}|$*

*then, $V_i$ is ID w.r.t. $V_j$.* □

*Proof.* **We denote $\mathbf{V}^{tar}$ as Q for convenience.** Notice that Assumption 4 will be used in the proof. From Eq. 3, we have

$$\sum_{V_i \in \tilde{\mathbf{V}}} \log p_{\mathbf{T}}^{(a_l)}(v_i \mid \mathbf{pa}_i^{\mathbf{T}+}) - p_{\mathbf{T}}^{a_0}(v_i \mid \mathbf{pa}_i^{\mathbf{T}+}) = \sum_{V_i \in \tilde{\mathbf{V}}} \log p_{\mathbf{T}}^{(a_l)}(\widehat{v}_i \mid \widehat{\mathbf{pa}}_i^{\mathbf{T}+}) - \log p_{\mathbf{T}}^{(a_0)}(\widehat{v}_i \mid \widehat{\mathbf{pa}}_i^{\mathbf{T}+})$$

$$= \log p_{\mathbf{T}}^{(a_l)}(\widehat{\mathbf{v}}) - \log p_{\mathbf{T}}^{(a_0)}(\widehat{\mathbf{v}}) \tag{30}$$

Notice that the left side only involves variables in $\mathbf{Q} = \bigcup_{l \in [L]} \Delta \mathbf{Q}[\mathbf{I}^{(a_l)}, \mathbf{I}^{(a_0)}, \mathbf{T}, G^S]$ based on the Def. 3.1.

We first argue that if $V_i \perp\!\!\!\perp V_j | \mathbf{Q}\backslash\{V_i, V_j\}$ in $G_{\overline{\mathbf{T}}}$ then $V_i \notin \mathbf{Pa}_j^{\mathbf{T}+}, V_j \notin \mathbf{Pa}_i^{\mathbf{T}+}$ and $V_i, V_j \notin \mathbf{Pa}_m^{\mathbf{T}+}$ for any $V_m \in \mathbf{Q}$.

First, since $V_i \perp\!\!\!\perp V_j | \mathbf{Q}\backslash\{V_i, V_j\}$, $V_i$ and $V_j$ cannot be directly connected by edges in $G_{\overline{\mathbf{T}}}$, which implies $V_i \notin \mathbf{C}(V_j)$ and $V_i \notin \mathbf{Pa}^{\mathbf{T}+}(V_j)$. Also, the outgoing edge from $V_i$ and $V_j$ cannot point to the same C-component. Otherwise, the path is active from $V_i$ and $V_j$ is active when conditioning on other variables (collider structure). Thus, $V_i \notin \mathbf{Pa}_j^{\mathbf{T}+}, V_j \notin \mathbf{Pa}_i^{\mathbf{T}+}$ and $V_i, V_j \notin \mathbf{Pa}_k^{\mathbf{T}+}$ where $V_k \in \mathbf{Q}$. This implies $V_i$ and $V_j$ will not appear to the same factor $p_{\mathbf{T}}^{(a_l)}(v_m \mid \mathbf{pa}_m^{\mathbf{T}+})$ for any $V_m \in \tilde{\mathbf{V}}$. Thus,

$$\frac{\partial^2 \log p_{\mathbf{T}}^{(a_l)}(v_m \mid \mathbf{pa}_m^{\mathbf{T}+})}{\partial v_i v_j} = 0 \tag{31}$$

Thus, for any pair of $V_k, V_r$ such that $V_k \perp\!\!\!\perp V_r | \mathbf{Q}\backslash\{V_k, V_r\}$,

$$\forall l \in [L], \sum_{V_m \in \tilde{V}} \frac{\partial^2 \log p_{\mathbf{T}}^{(a_l)}(\widehat{v}_m \mid \mathbf{pa}_m^{\mathbf{T}+})}{\partial \widehat{v}_k \widehat{v}_r}$$

$$= \frac{\partial^2 \log p_{\mathbf{T}}^{(a_l)}(\widehat{\mathbf{v}}) - \log p_{\mathbf{T}}^{(a_0)}(\widehat{\mathbf{v}})}{\partial \widehat{v}_k \widehat{v}_r} = 0 \tag{32}$$

On the other hand, when either $V_k$ or $V_r$ is in $\mathbf{Q}\backslash\Delta \mathbf{Q}^{(a_l),(a_0)}$ for $l \in [L]$,

$$\forall l \in [L], = \frac{\partial^2 \log p_{\mathbf{T}}^{(a_l)}(\widehat{\mathbf{v}}) - \log p_{\mathbf{T}}^{(a_0)}(\widehat{\mathbf{v}})}{\partial \widehat{v}_k \widehat{v}_r} = 0 \tag{33}$$

since

$$\frac{\partial p_{\mathbf{T}}^{(a_l)}(\widehat{\mathbf{v}}) - \log p_{\mathbf{T}}^{(a_0)}(\widehat{\mathbf{v}})}{\partial \widehat{v}_k} = 0 \quad \text{or} \quad \frac{\partial p_{\mathbf{T}}^{(a_l)}(\widehat{\mathbf{v}}) - \log p_{\mathbf{T}}^{(a_0)}(\widehat{\mathbf{v}})}{\partial \widehat{v}_r} = 0 \tag{34}$$

Upon Eq. (31), taking the second partial derivative on both sides of Eq. (30), the left side will be 0, and then $\forall\, l \in [L]$, we have

$$0 = \sum_{V_k, V_r \in \mathbf{Q}} \frac{\partial \log p_{\mathbf{T}}^{(a_l)}(\widehat{\mathbf{v}}) - \log p_{\mathbf{T}}^{(a_0)}(\widehat{\mathbf{v}})}{\partial \widehat{v}_k \widehat{v}_r} \frac{\partial \widehat{v}_k}{\partial v_i} \frac{\widehat{v}_r}{\partial v_j} \qquad \text{Chain Rule}$$

(35)

$$= \frac{\partial^2 \log p_{\mathbf{T}}^{(a_l)}(\widehat{\mathbf{v}}) - \log p_{\mathbf{T}}^{(a_0)}(\widehat{\mathbf{v}})}{\partial \widehat{v}_i^2} \frac{\partial \widehat{v}_i}{\partial v_i} \frac{\partial \widehat{v}_i}{\partial v_j} + \frac{\partial^2 \log p_{\mathbf{T}}^{(a_l)}(\widehat{\mathbf{v}}) - \log p_{\mathbf{T}}^{(a_0)}(\widehat{\mathbf{v}})}{\partial \widehat{v}_j^2} \frac{\partial \widehat{v}_j}{\partial v_i} \frac{\partial \widehat{v}_j}{\partial v_j}$$

$$+ \sum_{V_q \in \mathbf{Q}} \frac{\partial^2 \log p_{\mathbf{T}}^{(a_l)}(\widehat{\mathbf{v}}) - \log p_{\mathbf{T}}^{(a_0)}(\widehat{\mathbf{v}})}{\partial \widehat{v}_q^2} \frac{\partial \widehat{v}_q}{\partial v_i} \frac{\partial \widehat{v}_q}{\partial v_j}$$

$$+ \sum_{V_q \in \mathbf{Q}} \frac{\partial \log p_{\mathbf{T}}^{(a_l)}(\widehat{\mathbf{v}}) - \log p_{\mathbf{T}}^{(a_0)}(\widehat{\mathbf{v}})}{\partial \widehat{v}_q} \frac{\partial^2 \widehat{v}_q}{\partial v_i v_j}$$

$$+ \sum_{(V_k, V_r) \in \boldsymbol{\mathcal{E}}} \frac{\partial \log p_{\mathbf{T}}^{(a_l)}(\widehat{\mathbf{v}}) - \log p_{\mathbf{T}}^{(a_0)}(\widehat{\mathbf{v}})}{\partial \widehat{v}_k \widehat{v}_r} \frac{\partial \widehat{v}_k}{\partial v_i} \frac{\widehat{v}_r}{\partial v_j} \qquad \text{Eq. (32) and (32)}$$

(36)

Eq.( 36) is also a linear system. When distribution changes sufficiently, namely under Assumption 4, the row factor of the coefficient matrix of the linear system is linearly independent. When $L \geq 2|\mathbf{Q}| + \delta_{\not\perp}$ (implied by condition 4), the matrix is full rank, thus,

$$\frac{\partial \widehat{v}_i}{\partial v_i} \frac{\partial \widehat{v}_i}{\partial v_j} = 0, \quad \frac{\partial \widehat{v}_j}{\partial v_i} \frac{\partial \widehat{v}_j}{\partial v_j} = 0 \tag{37}$$

Then we have

$$\frac{\partial \widehat{v}_i}{\partial v_j} = 0, \quad \frac{\partial \widehat{v}_i}{\partial v_j} = 0 \tag{38}$$

up to a permutation of $V_i$ and $V_j$. This implies that $V_i$ is ID w.r.t $V_j$ and $V_j$ is ID w.r.t $V_i$.

When $\Delta \mathbf{Q}^{(a_1),(a_0)} = \cdots = \Delta \mathbf{Q}^{(a_L),(a_0)} \mathbf{V}^{tar}$

$\square$

Then we can get Prop. 4 directly from the above result.

**Proposition 4** (ID of variables within $\Delta \mathbf{Q}$ sets). *Consider the variables $\mathbf{V}^{tar} \subseteq \mathbf{V}$, $\mathcal{P}_{\mathbf{T}}$ that satisfies conditions (1) in Prop. 3 and $\Delta \mathbf{Q}^{(a_l),(a_0)} = \mathbf{V}^{tar}$, for $l \in [L]$. For any pair of $V_i, V_j \in \mathbf{V}^{tar}$ such that $V_i \perp\!\!\!\perp V_j | \mathbf{V}^{tar} \backslash \{V_i, V_j\}$ in $G_{\overline{\mathbf{T}}}(\mathbf{V}^{tar})$, $V_i$ is ID w.r.t. $V_j$ if $L \geq 2|\mathbf{V}^{tar}| + \delta_{\not\perp}$, where $\delta_{\not\perp}$ is the number of pair $V_k, V_r \in \mathbf{V}^{tar}$ such that $V_k$ and $V_r$ are connected given $\mathbf{V}^{tar} \backslash \{V_k, V_r\}$ in $G_{\overline{\mathbf{T}}}(\mathbf{V}^{tar})$.* $\square$

*Proof.* Taking $\Delta \mathbf{Q}^{(a_1),(a_0)} = \cdots = \Delta \mathbf{Q}^{(a_L),(a_0)} \mathbf{V}^{tar}$, the condition (2) is satisfied. When $L \geq 2|\mathbf{V}^{tar}| + \delta_{\not\perp}$, condition (3) is satisfied. Thus, Prop. 4 holds. $\square$

**Lemma 3** (ID of variables within $\Delta \mathbf{Q}$ sets). *Consider variables $\mathbf{V}^{tar} \subseteq \mathbf{V}$. For any pair of $V_i, V_j \in \mathbf{V}^{tar}$ such that $V_i \perp\!\!\!\perp V_j | \mathbf{V}^{tar} \backslash \{V_i, V_j\}$ in $G_{\overline{\mathbf{T}}}(\mathbf{V}^{tar})$, let $\mathbf{Mem}_i$ be a list of variables in $\mathbf{Q}$ that have been ID w.r.t. $V_i$ and let $\mathbf{Mem}_j$ be a list of qvariables in $\mathbf{Q}$ that have been ID w.r.t. $V_j$. If there exists $\mathcal{P}_{\mathbf{T}}$ that satisfies conditions [1-2] in Prop. 3 and the following condition [4'].*

[4'] *(Enough changes occur across distributions) Let $\mathbf{Q}^{re} = \mathbf{V}^{tar} \backslash (\mathbf{Mem}_i \bigcup \mathbf{Mem}_j)$ and $d' = |\mathbf{Q}^{re}|$. And*

$$\boldsymbol{\mathcal{E}}_{ij} = \{\epsilon_j = \{V_k, V_r\} \mid i) \ \exists a_l, \{V_k, V_r\} \in \Delta \mathbf{Q}^{(a_l),(a_0)};$$
$$\text{ii) } V_k \text{ is connected to} V_r \text{ conditioning } \mathbf{V}^{tar} \backslash \{V_k, V_r\} \text{ in } G_{\overline{\mathbf{T}}}(\mathbf{V}^{tar}) \qquad (13)$$
$$\text{iii) } V_k, V_r \notin \mathbf{Mem}_i \cup \mathbf{Mem}_j\}$$

*there exists $\{a'_1, \ldots, a'_{2d'+|\boldsymbol{\mathcal{E}}|}\} \in \{a_1, \ldots, a_L\}$ such that for all $Q_i \in \mathbf{Q}^{re}, Q_i \in \Delta \mathbf{Q}^{(a'_i),(a_0)}], Q_i \in \Delta \mathbf{Q}^{(a'_{d'+i}),(a_0)}$ and for all $\epsilon_l \in \boldsymbol{\mathcal{E}}_{ij}, \epsilon_l \subseteq \Delta \mathbf{Q}^{(a'_{2d'+l}),(a_0)}$.*

*, then $V_i$ is ID w.r.t $V_j$.* $\qquad\square$

*Proof.* The unknown in the linear system

$$\frac{\partial \widehat{v}_q}{\partial v_i}\frac{\partial \widehat{v}_q}{\partial v_j} = 0, \tag{39}$$

if $V_p$ is ID w.r.t $V_i$ or $V_q$ is ID w.r.t $V_j$.

$$\frac{\partial^2 \widehat{v}_q}{\partial v_i v_j} = 0 \tag{40}$$

If $V_q$ is ID w.r.t $V_i$ or $V_j$. Even these terms are excluded in [4'], the system still has the zero solutions.

$\qquad\square$

### C.6 ID-reverse of existing disentangled variables - Proof of Proposition 5

The next Proposition provides an additional tool to achieve identifiability and leverages the fact that other variables may have previously been disentangled and independence relationships in the factorization.

**Proposition 5** (**ID of canceled variables w.r.t. $\Delta\mathbf{Q}$ sets**). *Suppose $\mathbf{\Psi}$ contains $\mathrm{perf}(\mathbf{T})$. Given $\mathbf{V}\backslash V^{tar}$ is ID w.r.t. a single variable $V^{tar}$, $V^{tar}$ is ID w.r.t. $\mathbf{V}\backslash V^{tar}$ if $V^{tar} \perp\!\!\!\perp \mathbf{V}\backslash V^{tar}$ in $G_{\overline{\mathbf{T}}}$.* $\qquad\square$

*Proof.* We first introduce a lemma for distribution preserving from [20].

**Lemma 4** (Lemma 2 of [20]). *Let $A = C = R$ and $B = \mathbb{R}^n$. Let $f : A \times B \to C$ be differentiable. Define differentiable measures $P_A$ on $A$ and $P_C$ on $C$. Let $\forall b \in B$, $f(\cdot, b) : A \to C$ be measure-preserving. Then $f$ is constant in $b$.*

Denote $\mathbf{V}\backslash\mathbf{V}^{tar}$ as $\mathbf{Z}$. $V^{tar} \perp\!\!\!\perp \mathbf{Z}$ in $G_{\overline{\mathbf{T}}}$ implies that

$$P_{\mathbf{T}}(\mathbf{V}) = P_{\mathbf{T}}(V^{tar})P_{\mathbf{T}}(\mathbf{Z}) \tag{41}$$

With the change of variable formulation and taking log:

$$\log p_{\mathbf{T}}(\mathbf{v}^{tar}) + \log p_{\mathbf{T}}(\mathbf{z}) = \log p_{\mathbf{T}}(\widehat{\mathbf{v}}^{tar}) + \log p_{\mathbf{T}}(\widehat{\mathbf{z}}) + \log |\mathbf{J}_\phi| \tag{42}$$

Since $\mathbf{Z}$ is ID w.r.t $V^{tar}$, $\partial\widehat{\mathbf{Z}}/\partial\mathbf{V}^{tar} = 0$. In other words, the elements $\partial\phi_Z/\partial\mathbf{V}^{tar} = 0$ for every $Z \in \mathbf{Z}$ in Jacobian matrix are 0, where $\phi_Z$ is a function mapping from $\mathbf{V}$ to $\widehat{Z}$. Then

$$\log |\mathbf{J}_\phi| = \log |\mathbf{J}_{\mathbf{Z}}| + \log |\mathbf{J}_{\mathbf{V}^{tar}}| \tag{43}$$

where $|\mathbf{J}_{\mathbf{Z}}| = \begin{bmatrix} \partial\phi_{Z_1}/\partial z_1 & \partial\phi_{Z_1}/\partial z_2 & \dots & \partial\phi_{Z_1}/\partial z_{d-1} \\ \partial\phi_{Z_2}/\partial z_1 & \partial\phi_{Z_2}/\partial z_2 & \dots & \partial\phi_{Z_2}/\partial z_{d-1} \\ \vdots & \vdots & \ddots & \vdots \\ \partial\phi_{Z_{d-1}}/\partial z_1 & \partial\phi_{Z_{d-1}}/\partial z_2 & \dots & \partial\phi_{Z_{d-1}}/\partial z_{d-1} \end{bmatrix}$ and $\log |\mathbf{J}_{\mathbf{V}^{tar}}| = |\partial\phi_{V^{tar}}/\partial v^{tar}|$.

Again, since $\mathbf{Z}$ is ID w.r.t $\mathbf{V}^{tar}$, $\widehat{\mathbf{Z}} = \phi_{\mathbf{Z}}(\mathbf{Z})$. Thus,

$$\log p_{\mathbf{T}}(\mathbf{z}) = \log p_{\mathbf{T}}(\widehat{\mathbf{z}}) + \log |\mathbf{J}_{\mathbf{Z}}| \tag{44}$$

Subtracting this to Eq. (42)

$$\log p_{\mathbf{T}}(v^{tar}) = \log p_{\mathbf{T}}(\widehat{v}^{tar}) + \log |\mathbf{J}_{\mathbf{V}^{tar}}| \tag{45}$$

Denote $\phi_{\mathbf{V}^{tar}}(\mathbf{z}, \cdot)$ as $\phi^{\mathbf{z}}_{\mathbf{V}^{tar}}(\cdot)$, which is the function $\phi_{\mathbf{V}^{tar}}$ fixing value $\mathbf{Z} = \mathbf{z}$ mapping from $\mathbf{V}^{tar}$ to $\widehat{\mathbf{V}}$. This suggests for every $\mathbf{z}$,

$$P_{\mathbf{T}}(\widehat{V}^{tar}) = P_{\mathbf{T}}(\phi^{\mathbf{z}}_{V^{tar}}(V^{tar})) \tag{46}$$

Apply Lemma 2 of [20], $\phi_{V^{tar}}$ should be a constant regarding $\mathbf{Z}$. Thus,

$$\forall Z \in \mathbf{Z}, \frac{\partial V^{tar}}{\partial Z} = 0 \tag{47}$$

$\qquad\square$

## C.7 Soundness of LatentID Algorithm - Proof of Thm. 1

The following provides the proof of the soundness of our proposed graphical algorithm for determining whether or not two variables are disentangleable given a collection of distributions from multiple domains and interventions.

**Theorem 1** (**Soundness of CRID**). *Consider a LSD $G^S$ and intervention targets $\mathbf{\Psi}$. Consider the target variables $\mathbf{V}^{tar}$ and $\mathbf{V}^{en} \subseteq \mathbf{V} \backslash \mathbf{V}^{tar}$. If no edges from $\mathbf{V}^{tar}$ points to $\widehat{\mathbf{V}}^{en}$ in the output causal disentanglement map (CDM) from **CRID**, $G_{V,\widehat{V}}$, then $\mathbf{V}^{tar}$ is ID w.r.t $\mathbf{V}^{en}$.* $\qquad \square$

*Proof.* In **LatentID**, for each epoch, we iterate to choose $\mathbf{T}$ and the baseline distribution to execute procedure Alg. F.3 and Alg. F.6. Any time an edge is removed, Proposition 3 and/or 4 are applied. At the end of epoch, Alg. F.8 is executed and edges will be removed only if Proposition 5 is applied. Thus the edge removals are all sound. The algorithm will stop when no edge will be removed, and terminate giving the causal disentanglement map $G_{V,\hat{V}}$, which is a valid summary of what is disentangleable. $\qquad \square$

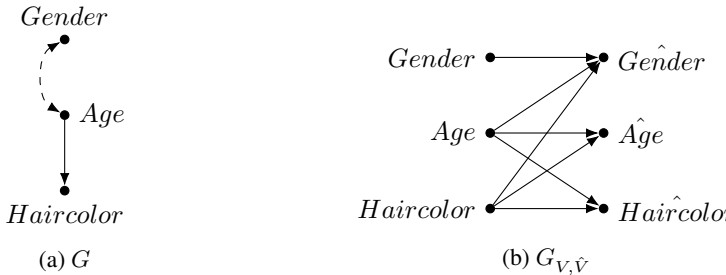

(a) $G$                         (b) $G_{V,\hat{V}}$

Figure S3: Latent causal graph and the desired causal disentanglement map.

# D  Examples and Discussion

## D.1  Additional Example Illustrating Motivation of Causal Disentangled Learning

In the introduction, we illustrated a medical example for why it is important to learn disentangled representations.

An additional motivating example can be seen through the lens of generating realistic face images [27]. Consider an image dataset of human faces. Based on our understanding of anatomy and facial expressions, we know that both $Gender$ and $Age$ are not causally related, while age does directly affect $HairColor$. There is a strong spurious correlation between age and gender, where there are many old males and young females in the dataset. In addition, let there be face images from both a senior and teen center building. The change in domain (i.e. population center) impacts the age distribution, as senior center faces are older than teen center faces. Given these images and knowledge of the latent causal graph, one would ultimately like to generate realistic face images given perturbations of $Age$. If the variable represen-

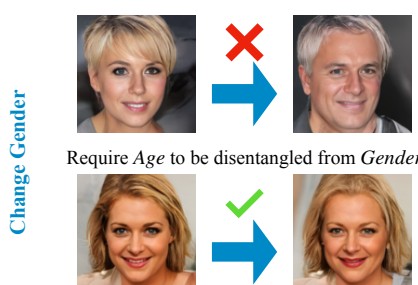

Require *Age* to be disentangled from *Gender*

Do not require *Age* being disentangled from *Haircolor*

Figure S2: The disentanglement requirements in face examples

tations are entangled, then it is possible for changes in age to also spuriously change gender. This is undesirable, and thus our goal is to achieve disentanglement of age and gender. Note that we do not require $Age$ to be disentangled from $HairColor$ necessarily since changing $Age$ and also simultaneously changing $Haircolor$ would be a realistic image generation. Here, we would seek a causal disentanglement map shown in Fig. S3.

If we could get the causal disentanglement map, then we know that when the representations are fully learned, we can intervene on $Age, an$ without changing the $Gender$ of the face. This motivates the need for a general approach to identifiability, compared to the scaling indeterminacy in Def. 6.5, which requires all variables to be disentangled from each other.

As another motivating example, consider a marketing company creating faces for a female product. The relevant latent factors are Gender $\leftrightarrow$ Age $\rightarrow$ Hair Color (see Appendix D.1 for details). If Gender and Age are entangled, changing Age might also alter Gender, which is undesirable. The company needs a model where Age is disentangled from Gender, while correlation with Hair Color is allowed. Our paper addresses the problem of determining whether a given set of input data and assumptions in the form of a LSD is sufficient to learn such a disentangled representation.

## D.2  Examples for non-Markovian Factorization

In this section, we centralize theoretical results in relation to the theory presented in this paper.

Unless specified, we denote the natural log as $\log$.

We first provide more discussion about non-Markovian factorization Eq. (1). First, the concept C-component is formally defined as follows:

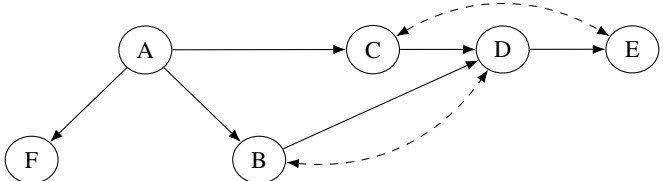

Figure S4: Causal graph with four C-components.

**Definition 6.2** (Confounded Component). Let $\{\mathbf{C_1}, \mathbf{C_2}, \ldots, \mathbf{C_k}\}$ be a partition over the set of variables $\mathbf{V}$, where $\mathbf{C_i}$ is said to be a confounded component (for short, $C$-component) of the selection diagram $G_V$ if for every $V_i, V_j \in \mathbf{C_i}$ there exists a path made entirely of bidirected edges between $V_i$ and $V_j$ in $G_V$, and $\mathbf{C_i}$ is maximal. $\qquad\square$

This construct represents clusters of variables that share the same exogenous variations regardless of their directed connections. The selection diagram in Figure 2 has a bidirected edge indicating the presence of unobserved confounders affecting the pairs $(V_1, V_2)$ and contains two C-components, namely, $\mathbf{C_1} = \{V_1, V_2\}, \mathbf{C_2} = \{V_3\}$.

Akin to parents within a Markovian SCM, the c-components play a fundamental role in factorizing the joint distribution of the observed variables $\mathbf{V}$.

Let $<$ be a topological order $V_1, \ldots, V_n$ of the variables $\mathbf{V}$ in $G^S$. Then define the $\mathbf{Pa}_i^{\mathbf{T}+} = \mathbf{Pa}(\{V \in \mathbf{C}(V_i) : V \leq V_i\}) \setminus \{V_i\}$. The $\mathbf{Pa}^+(V_i)$ set consists of the nodes in the same c-component that are "$\leq$" in topological order as $V_i$, their corresponding parents, minus the node $V_i$ itself. For instance, in Fig. S4, $Pa^+(E) = \{D, C, A\}$ and $Pa^+(D) = \{B, C, A\}$.

The general factorization formula Eq. (1) factorizes not only the joint observational distribution related to a causal graph, but also interventional distributions. With a perfect intervention on $\mathbf{T}$, the factorization follows the corresponding graph is $G_{\overline{\mathbf{T}}}$, where the incoming arrows towards $\mathbf{T}$ are cut. This factorization encompasses both Markovian and non-Markovian SCM models. When there are no bidirected edges in the diagram, $\mathbf{Pa}_i^{\mathbf{T}+}$ reduce to $\mathbf{Pa}$ in $F_{\mathbf{T}}$.

Next, we introduce the Markov blanket, a fundamental idea in characterizing certain conditional independences in a causal graph [71, 72].

**Definition 6.3** (Markov Blanket). Let $G$ be a causal graph over variables $\mathbf{V}$. A Markov blanket of a random variable $Y \in \mathbf{V}$ is any subset $V_1 \subseteq \mathbf{V}$ such that conditioned on $V_1$, $Y$ is independent of all other variables.

$$Y \perp\!\!\!\perp \mathbf{V} \backslash V_1 | V_1$$

$\qquad\square$

The Markov blanket is an important object that captures conditional independences between variables when conditioned on *all other variables* in the graph.

**Definition 6.4** ("Global" Markov property of DAGs [73]). Consider a joint probability distribution, $P$ over a set of variables $\mathbf{V}$ satisfies the **Markov property** with respect to a graph $G = (V \cup L, E)$ if the following holds for, $(\mathbf{X}, \mathbf{Y}, \mathbf{Z})$ disjoint subsets of V:

$$P(y|x, z) = P(y|z) \quad \text{if } Y \perp\!\!\!\perp X | Z \text{ in G (that is Y is d-separated from X given Z)}$$

$\qquad\square$

The global Markov property maps graphical structure in causal directed acyclic graphs (DAGs) to conditional independence (CI) statements in the relevant probability distributions from data. The distributions we consider $\mathcal{P}$ are considered Markov wrt the graph, thus mapping d-separations in the graph to conditional independences in the distributions. This allows us to leverage factorizations, such as the one presented in Section 2.

# E   Examples for Proposition 2

The following example illustrates more about the invariant factors.

**Example 14.** *(Example 1 continued.) Choose $P^{(1)}$ as the baseline and $\mathbf{T} = \{\}$. The factorization of $P(\mathbf{V})$ is $P(V_1)P(V_2 \mid V_1)P(V_3 \mid V_2)$. The changed variable set $\Delta\mathbf{V}[\mathbf{I}^{(2)}, \mathbf{I}^{(1)}, G^S] = \{V_3\}$ since the S-node points to $V_3$ in $G^S$ and $\Delta\mathbf{V}[\mathbf{I}^{(3)}, \mathbf{I}^{(1)}, G^S] = \{V_3\}$ since $V_3 \in \mathbf{I}^{(3)}$ while $V_3 \notin \mathbf{I}^{(1)}$. Thus, comparing $P^{(2)}$ and $P^{(3)}$ with the baseline $P^{(1)}$, $p(v_2 \mid v_1)$ and $p(v_1)$ are invariant factors while $p(v_3 \mid v_2)$ possibly changes.* ☐

## E.1   The detailed examples of Proposition 3 and 4

We show another example of using Proposition 3 to solve an ID task in Example 1.

**Example 15.** *(Example 14 continued.) Consider $\mathbf{V}^{tar} = \{V_2, V_3\}$, $\mathbf{V}^{en} = \mathbf{V}\backslash\{V_2, V_3\} = \{V_1\}$. When comparing $\{P^{(2)}, P^{(3)}\}$ with the baseline $P^{(1)}$, $\mathbf{T} = \sigma_M[\mathbf{I}^{(1)}] = \{\}$, and then*

$$\Delta\mathbf{Q}[\mathbf{I}^{(2)}, \mathbf{I}^{(1)}, \mathbf{T}, G^S] = \Delta\mathbf{Q}[\mathbf{I}^{(3)}, \mathbf{I}^{(1)}, \mathbf{T}, G^S] = \mathbf{V}^{tar} \tag{48}$$

*Thus, these two comparisons satisfy the three conditions in Prop.3. Because the number of compared distribution $\{P^{(2)}, P^{(3)}\}$ is 2, which is equal to $|\mathbf{V}^{tar}|$, then we know $\mathbf{V}^{tar}$ is ID w.r.t $\mathbf{V}^{en}$ by Prop. 3. This demonstrates that a variable $V_2$ can be disentangled from another variable that is in the C-component ($V_1$). See Appendix Ex. 16 for a detailed derivation.* ☐

Proposition 3 and 4 disentangle variables through comparing distributions. With enough distributions, one can build a linear system (illustrated in Appendix C.4 and C.5).

**Example 16.** *(details for Example 15). By comparing distribution resulting from $\sigma^{(2)}$ and $\sigma^{(3)}$ with the baseline $\sigma^{(1)}$,*

$$\begin{aligned}
\log p^{(2)}(v_3 \mid v_2) - \log p^{(1)}(v_3 \mid v_2) &= \log p^{(2)}(\widehat{v}_3 \mid \widehat{v}_2) - \log p^{(1)}(\widehat{v}_3 \mid \widehat{v}_2) \\
\log p^{(3)}(v_3 \mid v_2) - \log p^{(1)}(v_3 \mid v_2) &= \log p^{(3)}(\widehat{v}_3 \mid \widehat{v}_2) - \log p^{(1)}(\widehat{v}_3 \mid \widehat{v}_2)
\end{aligned} \tag{49}$$

*Taking the first order partial derivative w.r.t. $V_1$:*

$$\begin{aligned}
0 &= \frac{\partial \log p^{(2)}(\widehat{v}_3 \mid \widehat{v}_2) - \log p^{(1)}(\widehat{v}_3 \mid \widehat{v}_2)}{\partial \widehat{v}_2} \frac{\partial \widehat{v}_2}{\partial v_1} + \frac{\partial \log p^{(2)}(\widehat{v}_3 \mid \widehat{v}_2) - \log p^{(1)}(\widehat{v}_3 \mid \widehat{v}_2)}{\partial \widehat{v}_3} \frac{\partial \widehat{v}_3}{\partial v_1} \\
0 &= \frac{\partial \log p^{(3)}(\widehat{v}_3 \mid \widehat{v}_2) - \log p^{(1)}(\widehat{v}_3 \mid \widehat{v}_2)}{\partial \widehat{v}_2} \frac{\partial \widehat{v}_2}{\partial v_1} + \frac{\partial \log p^{(3)}(\widehat{v}_3 \mid \widehat{v}_2) - \log p^{(1)}(\widehat{v}_3 \mid \widehat{v}_2)}{\partial \widehat{v}_3} \frac{\partial \widehat{v}_3}{\partial v_1}
\end{aligned} \tag{50}$$

*In this system, notice that*

$$\log p^{(2)}(\widehat{v}_3 \mid \widehat{v}_2) - \log p^{(1)}(\widehat{v}_3 \mid \widehat{v}_2) = \log p^{(2)}(\widehat{v}_1, \widehat{v}_2, \widehat{v}_3) - \log p^{(1)}(\widehat{v}_1, \widehat{v}_2, \widehat{v}_3) \tag{51}$$

*Then since the coefficient is linear independent assumed in Assumption 4, we have*

$$\frac{\partial \widehat{v}_2}{\partial v_1} = 0, \frac{\partial \widehat{v}_3}{\partial v_1} = 0 \tag{52}$$

*Then $V_2 = \tau_2(V_2, V_3)$, and $V_3 = \tau_3(V_2, V_3)$.*

*First, this example shows we can disentangle two variables in the same C-component ($V_1, V_2$). Second, Compared with the baseline, one can disentangle variable $V$ with its descendants when soft interventions are given per node, and $V$ is considered to be still entangled with its ancestral (see Sec. F.6). The above result shows that it is possible to disentangle variables from their ancestors using only soft interventions. More interestingly, no intervention is performed on $V_2$ while we disentangle $V_2$ from $V_1$. Compared with [22], one can disentangle $V_1$ and $V_3$ using 10 distributions and we demonstrate 3 distributions are enough.* ☐

**Example 17.** *(details for Example 4). Choosing order $V_1 < V_3 < V_2 < V_4$.*

$$P(\mathbf{V}) = P(V_1)P(V_3)P(V_2 \mid V_1, V_3)P(V_4 \mid V_3) \tag{53}$$

*as the factorization. By comparing distribution resulting from $\sigma^{(2)}$ and $\sigma^{(3)}$ with the baseline $\sigma^{(1)}$,*

$$\log p^{(2)}(v_2 \mid v_1, v_3) - \log p^{(1)}(v_2 \mid v_1, v_3) = \log p^{(2)}(\widehat{v}_2 \mid \widehat{v}_1, \widehat{v}_3) - \log p^{(1)}(\widehat{v}_2 \mid \widehat{v}_1, \widehat{v}_3)$$

$$\log p^{(3)}(v_3) - \log p^{(1)}(\widehat{v}_3) + \log p^{(3)}(v_2 \mid v_1, v_3) - \log p^{(1)}(v_2 \mid v_1, v_3) =$$

$$\log p^{(3)}(\widehat{v}_1)(\widehat{v}_3) - \log p^{(3)}(\widehat{v}_3) + \log p^{(2)}(\widehat{v}_2 \mid \widehat{v}_1, \widehat{v}_3) - \log p^{(1)}(\widehat{v}_2 \mid \widehat{v}_1, \widehat{v}_3) \tag{54}$$

$$\log p^{(4)}(v_1) - \log p^{(1)}(v_1) = \log p^{(4)}(\widehat{v}_1) - \log p^{(1)}(\widehat{v}_1)$$

*Taking the first order partial derivative w.r.t. $V_4$:*

$$0 = h_{2,1}\frac{\partial \widehat{v}_1}{\partial v_4} + h_{2,2}\frac{\partial \widehat{v}_2}{\partial v_4} + h_{2,3}\frac{\partial \widehat{v}_3}{\partial v_4}$$

$$0 = h_{3,1}\frac{\partial \widehat{v}_1}{\partial v_4} + h_{3,2}\frac{\partial \widehat{v}_2}{\partial v_4} + h_{3,3}\frac{\partial \widehat{v}_3}{\partial v_4} \tag{55}$$

$$0 = h_{4,1}\frac{\partial \widehat{v}_1}{\partial v_4}$$

*where*

$$h_{2,i} = \frac{\partial \log p^{(2)}(\widehat{v}_2 \mid \widehat{v}_1, \widehat{v}_3) - \log p^{(1)}(\widehat{v}_2 \mid \widehat{v}_1, \widehat{v}_3)}{\partial \widehat{v}_4} \quad for\ i = 1, 2, 3$$

$$h_{3,i} = \frac{\partial \log p^{(3)}(\widehat{v}_3) - \log p^{(1)}(\widehat{v}_3) + \log p^{(3)}(\widehat{v}_2 \mid \widehat{v}_1, \widehat{v}_3) - \log p^{(1)}(\widehat{v}_2 \mid \widehat{v}_1, \widehat{v}_3)}{\partial \widehat{v}_4} \quad for\ i = 1, 2, 3$$

$$h_{4,1} = \frac{\partial p^{(4)}(\widehat{v}_1) - \log p^{(1)}(\widehat{v}_1)}{\partial \widehat{v}_4}$$

$$\tag{56}$$

*Then since the coefficient is linear independent assumed in Assumption 4, we have*

$$\frac{\partial \widehat{v}_1}{\partial v_4} = 0, \frac{\partial \widehat{v}_2}{\partial v_4} = 0, \frac{\partial \widehat{v}_3}{\partial v_4} = 0 \tag{57}$$

*Then $V_1 = \tau_1(V_1, V_2, V_3)$, and $V_2 = \tau_2(V_1, V_2, V_3)$ and $V_3 = \tau_3(V_1, V_2, V_3)$.* $\square$

**Example 18.** *The factorization based on $G^S$ choosing $\mathbf{T} = \{\}$ is*

$$P(\mathbf{V}) = P(V_1)P(V_3)P(V_3 \mid V_1, V_2) \tag{58}$$

*By comparing distribution resulting from $\sigma^{(2)}$ and $\sigma^{(3)}$ with the baseline $\sigma^{(1)}$, for $j = 2, 3, 4, 5$*

$$\log p^{(j)}(v_1) + \log p^{(j)}(v_3) - \log p^{(1)}(v_1) - \log p^{(1)}(v_3)$$

$$= \log p^{(j)}(\widehat{v}_\cdot) + \log p^{(j)}(\widehat{v}_3) - \log p^{(1)}(\widehat{v}_1) - \log p^{(1)}(\widehat{v}_3) \tag{59}$$

*Taking the second order partial derivative w.r.t. $V_1, V_3$:*

$$0 = \frac{\partial^2 \log p^{(j)}(\widehat{v}_1)p^{(j)} - \log p^{(1)}(\widehat{v}_1)}{\partial \widehat{v}_1^2}\frac{\partial \widehat{v}_1}{\partial v_1}\frac{\partial \widehat{v}_1}{\partial v_3} + \frac{\partial^2 \log p^{(j)}(\widehat{v}_3)p^{(j)} - \log p^{(1)}(\widehat{v}_3)}{\partial \widehat{v}_3^2}\frac{\partial \widehat{v}_3}{\partial v_1}\frac{\partial \widehat{v}_3}{\partial v_3}$$

$$+ \frac{\partial \log p^{(j)}(\widehat{v}_1)p^{(j)} - \log p^{(1)}(\widehat{v}_1)}{\partial \widehat{v}_1}\frac{\partial^2 \widehat{v}_1}{\partial v_1 \partial V_3} + \frac{\partial \log p^{(j)}(\widehat{v}_3)p^{(j)} - \log p^{(1)}(\widehat{v}_3)}{\partial \widehat{v}_3}\frac{\partial^2 \widehat{v}_3}{\partial v_1 \partial v_3}$$

$$\tag{60}$$

*Then since the coefficient is linear independent assumed in Assumption 4, we have*

$$\frac{\partial \widehat{v}_1}{\partial v_1}\frac{\partial \widehat{v}_1}{\partial v_3} = 0, \frac{\partial \widehat{v}_3}{\partial v_1}\frac{\partial \widehat{v}_3}{\partial v_3} = 0 \tag{61}$$

*Then after permutation,*

$$\frac{\partial \widehat{v}_1}{\partial v_3} = 0, \frac{\partial \widehat{v}_3}{\partial v_1} = 0 \tag{62}$$

*which implies that $V_3$ is ID w.r.t $V_1$ and $V_1$ is ID w.r.t $V_3$.* $\square$

The following example shows how Proposition 5 achieves disentanglement for the ID task in Example 1.

**Example 19.** *(Example 15 (continued).) Let $P^{(4)}$ with intervention target $\mathbf{I}^{(4)} = \{V_2^{1,\{1\},\text{do}}\}$ be another distribution added to the original setting. Consider $\mathbf{V}^{tar} = \{V_1\}$. From Ex. 14, $\{V_2, V_3\}$ is ID w.r.t. $V_1$. Consider $\mathbf{T} = \{V_2\}$ (from $\mathbf{I}^{(4)}$). Since $V_1 \perp\!\!\!\perp \{V_2, V_3\}$, then $V_1$ is ID w.r.t $\{V_2, V_3\}$.* $\square$

## F Related Work Discussion

Disentangled representation learning aims to obtain approximations $\widehat{\mathbf{V}} = \{\widehat{V}_1, \ldots, \widehat{V}_d\}$ that separate the distinct, informative generative factors of variations [5] from the observations of $\mathbf{X}$ and inductive bias of $\mathcal{M}$. In other words, the learning goal is an unmixing function $\widehat{f}_X^{-1}$ that maps from $\mathbf{X}$ to $\widehat{\mathbf{V}}$ (namely $\widehat{\mathbf{V}} = \widehat{f}_X^{-1}(\mathbf{X})$), where $\widehat{V}_i$ is some transformation of $\mathbf{W} \subseteq \mathbf{V}$. The goal of disentangled representation learning is to have $\widehat{V}_i$ be a function only of $V_i$, i.e. $\mathbf{W} = \{V_i\}$. This is not always possible, and different assumptions, data and relaxed versions of disentanglement may be studied to theoretically ground representation learning. The disentangled representation learning tasks are studied with various assumptions and input. In the following, we discuss related tasks and identifiability results in context of this paper. We also present a few case studies on the nuances between Markovian and non-Markovian ASCM setting.

First, we review the main goal of identifiability in all prior works. It is what is known as scaling identifiability. A special case of our ID definition in Def. 2.3.

**Definition 6.5** (Scaling indeterminancy). Consider a collection of ASCM $\mathcal{M}$ that induces an LSD $G^S$ and a collection of distribution $\mathcal{P}$. We say $\mathbf{V}$ is identifiable up to scaling indeterminacy if for every $\widehat{\mathcal{M}}$ matches with the $G^S$ and $\mathcal{P}$, there exists functions $\{h_1, \ldots, h_d\}$ such that $\widehat{V}_i = \mathbf{h}_i(V_i), i \in [d]$, where $h_i$ is a diffeomorphism in $\mathbb{R}$. □

### F.1 Causal representation learning with unknown latent causal structure

In many prior works, the goal has been not only identifiability of the underlying latent variables, but also the discovery of the causal relationships among the latent variables [4, 22, 51]. That is, the latent causal graph is unknown. The work proposed in this paper is a foundation for the first step of causal representation learning, i.e. identifying the distributions of the latent causal variables. It would be interesting future work to explore how the results proposed in this paper extend to the case when the latent causal graph is unknown.

### F.2 Comparisons with other identifiability criterion

We also consolidate other definitions of identifiability from the literature using the notion of an ASCM. We have already defined identifiability up to scaling ambiguity in Def. 6.5.

**Corollary 2** (Scaling ID is a case in general ID). *Let $\mathcal{M}$ be a collection of ASCM with $G^S$ the LSD over the latent causal variables $\mathbf{V}$. If $\tilde{V} \subseteq \mathbf{V}$ is identifiable up to scaling indeterminacy, then it is identifiable wrt $\mathbf{V} \backslash \tilde{V}$.*

*Proof.* The proof follows from the application of Def. 2.3 and Def. 6.5. □

**Definition 6.6** (Identifiability up to ancestral mixtures [21]). Let $\mathcal{M}$ be a collection of ASCM with $G^S$ the LSD over the latent causal variables $\mathbf{V}$. We say a variable $\tilde{V} \in \mathbf{V}$ is identifiable up to ancestral mixtures if for every $\widehat{\mathcal{M}}$ matches with the $G^S$ and $\mathcal{P}$, there exists functions $\{h_1, \ldots, h_d\}$ such that $\widehat{V}_i = \mathbf{h}_i(\overline{\mathbf{Anc}}(V_i)), i \in [d]$. □

**Corollary 3** (Ancestral ID is a case in general ID). *Let $M$ be a collection of ASCM with $G$ the LSD over the latent causal variables $\mathbf{V}$. If $\tilde{V} \subseteq \mathbf{V}$ is identifiable up to ancestral mixtures, then it is identifiable wrt $\mathbf{V} \backslash \mathbf{Anc}(\tilde{V})$.*

*Proof.* The proof follows from the application of Def. 2.3 and Def. 6.6. □

The following definitions are inspired by the identifiability results from [22].

**Definition 6.7** (Intimate Neighbor Set). We say $\varepsilon_{M_G, V_i} := \{V_j \mid j \neq i,$ but $V_j$ is adjacent to $V_i$ and all other neighbors of $V_i$ in $M_G$. □

The intimate neighbor set for a variable dictates a set of neighbors that are adjacent to all of that variable's neighbors. It is used in the following definition from [22].

**Definition 6.8** (Identfiability up to intimate neighbor set of Markov Network [22]). Let $\mathcal{M}$ be a collection of ASCM with $G^S$ the LSD over the latent causal variables $\mathbf{V}$. We say a variable $\tilde{V} \in \mathbf{V}$ is identifiable up to intimate neighbors in the Markov Network if for every $\widehat{\mathcal{M}}$ matches with the $G^S$ and $\mathcal{P}$, there exists functions $\{h_1, \dots, h_d\}$ such that $\widehat{V_i} = \mathbf{h}_i(\varepsilon(M_G, V_i)), i \in [d]$, and $M_G$ is the Markov network of $G$ and $\varepsilon(M_G, V_i)$ is the intimate neighbor set of $V_i$ in $M_G$. $\qquad\square$

**Corollary 4** (Intimate Neighbor Markov Network ID is a case in general ID). *Let $M$ be a collection of ASCM with $G$ the LSD over the latent causal variables $\mathbf{V}$. If $\tilde{V} \subseteq \mathbf{V}$ is identifiable up to intimate neighbor set of the Markov Network, then it is identifiable wrt $\mathbf{V} \backslash \phi(MN(G); \tilde{V})$.*

*Proof.* The result follows from the application of Def. 2.3 and Def. 6.8. $\qquad\square$

Thus, we showed that each of these identifiability definitions imply a general ID for a non-trivial subset of latent variables $\tilde{\mathbf{V}} \subseteq \mathbf{V}$ with respect to $\mathbf{V}^{en} \subset \mathbf{V}$.

### F.3 Case study on challenges when disentangling variables in a non-Markovian setting

Prior results suggest that in a Markovian setting, given a perfect intervention on every node, the latent variables $\mathbf{V}$ are ID up to scaling indeterminacies according to Def. 6.5 [14, 21].

One would suspect that ID may still hold in non-Markovian ASCMs, but the following result states that even with one perfect intervention per node, it is not possible to disentangle latent variables within the same c-component.

**Lemma 5** (Challenges of identifability in non-Markovian causal models). *Consider the ASCM $M$ that induces the diagram $V_1 \leftrightarrow V_2$. Suppose the intervention set includes an observational distribution, and perfect interventions on both $V_1$ and $V_2$: $\Psi = \langle \sigma_{\{\}}, \sigma_M(\{V_1\}), \sigma_M(\{V_2\})\rangle$. Then $V_1$ is not ID w.r.t $V_2$ and vice versa.* $\qquad\square$

*Proof.* We prove this by construction of a counter-example.

Consider an ASCM $M^*$ that is constructed as follows:

$$\mathcal{F}^* = \begin{cases} V_1 \leftarrow U_{1,2} \\ V_2 \leftarrow U_{1,2} + U_{V_2} \\ X_1 \leftarrow V_1, X_2 \leftarrow V_2 \end{cases}$$
$$U_{1,2} \sim \mathcal{N}(0,1), U_Y \sim \mathcal{N}(0,3)$$
$$\sigma_{V_1} = P(\tilde{U_{V_1}}), \tilde{U}_{V_1} \sim \mathcal{N}(0,2)$$
$$\sigma_{V_2} = P(\tilde{U_{V_2}}), \tilde{U}_{V_1} \sim \mathcal{N}(0,7)$$

Consider a separate ASCM $M^{(1)}$ that is constructed as follows:

$$\mathcal{F}^{(1)} = \begin{cases} V_1^{(1)} \leftarrow -U_{1,2}^{(1)} \\ V_2^{(1)} \leftarrow 0.5U_{1,2}^{(1)} + 1.5U_Y \\ X_1 \leftarrow 1/3V_1^{(1)} + 2/3V_2^{(1)}, \\ X_2 \leftarrow 2/3V_1^{(1)} - 2/3V_2^{(1)} \end{cases}$$
$$U_{1,2}^{(1)} \sim \mathcal{N}(0,3), U_{V_2}^{(1)} \sim \mathcal{N}(0,1)$$
$$\sigma_{V_1} = P(\tilde{U_{V_1}}^{(1)}), \tilde{U}_{V_1}^{(1)} \sim \mathcal{N}(0,6)$$
$$\sigma_{V_2} = P(\tilde{U_{V_2}}^{(1)}), \tilde{U}_{V_2}^{(1)} \sim \mathcal{N}(0,7)$$

$M^*$ and $M^{(1)}$ induce the same observational distribution $P(\mathbf{X}) \sim \mathcal{N}(0, \begin{bmatrix} 1 & 1 \\ 1 & 4 \end{bmatrix})$, and interventional distributions $P(\mathbf{X}; \sigma_{V_1}) \sim \mathcal{N}(0, \begin{bmatrix} 2 & 0 \\ 0 & 4 \end{bmatrix}), P(\mathbf{X}; \sigma_{V_2}) \sim \mathcal{N}(0, \begin{bmatrix} 1 & 0 \\ 0 & 7 \end{bmatrix})$

However, $V_1^{(1)} = V_1 - V_2$, which implies $V_1^{(1)}$ is not ancestral mixture or rescaling of the original $V_1$. Therefore, $V_1$ is not identifiable up to ancestral mixtures, or rescaling. $\square$

### F.4 ID within c-components

Lemma 5 shows that even with one perfect intervention on each node, it is not possible to disentangle variables within the same c-component. The next lemma provides a means of doing so using two perfect interventions on the same node. This provides some intuition for the usefulness of perfect interventions in the **CRID** setting.

**Lemma 6** (Two perfect interventions can disentangle within a c-component). *Let $G^S$ be the LSD induced from a collection of ASCM $\mathcal{M}$. Suppose $V_i, V_j \in \mathbf{V}$ are in the same c-component, and there are $L + 1$ perfect interventions distributions $\mathcal{P}_{V_i} = \{P^{(a_0)}, P^{(a_1)}, \ldots, P^{(a_L)}\}$ such that $V_i \in do[\mathbf{I}^{(a_l)}]$ and $\Delta \mathbf{Q}[\mathbf{I}^{(a_l)}, \mathbf{I}^{(a_0)}, V_i, G^S]$ are equivalent (denoted as $\mathbf{Q}$) for $l \in [L]$. When $V_j \notin \mathbf{Q}$ and if $L \geq |\mathbf{Q}|$, $V_i$ is identifiable wrt $V_j$. When $V_j \in \mathbf{Q}$ and if $L \geq 2|\mathbf{Q}| + \delta_{\not\perp}$, $V_i$ is identifiable wrt $V_j$.*

*Proof.* The result follows from the application of Proposition 3 and Proposition 4. $\square$

**Example 20.** *In most simple case. Let's have $do[\mathbf{I}^{(j)}] = do[\mathbf{I}^{(k)}] = V_i$ and $\Delta \mathbf{Q} = V_i$. Let $V_i, V_j \in \mathbf{C}_k$ be two arbitrary latent variables in the same c-component. By comparing distributions, we have*

$$p_{V_i}^{(2)}(v_i) - p_{V_i}^{(1)}(v_i) = p_{V_i}^{(2)}(\widehat{v}_i) - p_{V_i}^{(1)}(\widehat{v}_i) \tag{63}$$

*Taking partial w.r.t. $V_j$, we have*

$$0 = \frac{p_{V_i}^{(2)}(\widehat{v}_i) - p_{V_i}^{(1)}(\widehat{v}_i)}{\widehat{v}_i} \frac{\widehat{v}_i}{v_j} \tag{64}$$

*which implies $\frac{\widehat{v}_i}{v_j} = 0$.*

Notice that this is not the only way to disentangle to variables in the C-Component. In Example 6, $V_1$ and $V_2$ are disentangled from each other without leveraging two perfect interventions.

### F.5 Case study on disentangling variables in a Markovian setting

This next example works out the algebraic derivations for analyzing Fig. 4(a). This derivation is provided to provide additional intuition on the theory presented in Section 3, and how these concepts apply in a simple 3-dimensional latent causal graph.

**Example 21** (Algebraic derivation of disentanglement in a simple 3-node chain graph). *Given the graph shown in Figure 4(a), we can factorize the joint observational distribution of the latent variables*

$$P(\mathbf{V}) = P(V_3|V_2)P(V_1|V_2)P(V_2) \tag{65}$$

*By the probability transformation formula, we can similarly write the distribution in terms of its estimated sources via function $\phi = \hat{f}_{\mathbf{X}}^{-1} \circ f_{\mathbf{X}}$ for its distribution Q.*

$$P(\mathbf{V}) = P(\phi_{V_3}(\mathbf{V})|\phi_{V_2}(\mathbf{V}))P(\phi_{V_1}(\mathbf{V})|\phi_{V_2}(\mathbf{V})P(\phi_{V_2}(\mathbf{V}))|det J_\phi| \tag{66}$$

*Now, consider the interventional distributions: $P(\mathbf{V}; \sigma_{V_3^{(1)}})$ and $P(\mathbf{V}; \sigma_{V_3^{(2)}})$. Here, we will use shorthand $\phi_i$ to indicate $\phi_{V_i}(\mathbf{V})$. Similarly, we can factorize the distribution $P(\mathbf{V}; \sigma_{V_3^{(1)}})$:*

$$P(\mathbf{V}; \sigma_{V_3^{(1)}})$$
$$= P(V_3|V_2; \sigma_{V_3^{(1)}})P(V_1|V_2; \sigma_{V_3^{(1)}})P(V_2; \sigma_{V_3^{(1)}}) \quad \textit{(Conditional independence)}$$
$$= P(\phi_3|\phi_2; \sigma_{3^{(1)}})P(\phi_1|\phi_2; \sigma_{V_3^{(1)}})P(\phi_2; \sigma_{V_3^{(1)}})|det J_\phi| \quad \textit{(Probability transformation formula)}$$

*Similarly, we can decompose the interventional distribution $P(\mathbf{V}; \sigma_{V_3^{(2)}})$. Now, comparing the log observational distribution with the log intervention $\sigma_{V_3^{(i)}}$, we get:*

$$\log p(\mathbf{V}; \sigma_{V_3^{(i)}}) - \log p(\mathbf{V})$$
$$= \log p(V_3|V_2; \sigma_{V_3^{(i)}}) + \log p(V_1|V_2; \sigma_{V_3^{(i)}}) + \log p(V_2; \sigma_{V_3^{(i)}})$$
$$- \log p(V_3|V_2) - \log p(V_1|V_2) - \log p(V_2)$$
$$= \log p(V_3|V_2; \sigma_{V_3^{(i)}}) - \log p(V_3|V_2)$$

*Where the last line applies the invariance of $P(V_i|V_j; \sigma_{V_k}) = P(V_i|V_j)$ if $(V_i \perp\!\!\!\perp V_k|V_j)_{G_{\overline{V_3}}}$. In the space mapped by $\phi$, we similarly get:*

$$\log p(\phi; \sigma_{V_3^{(i)}}) - \log p(\phi)$$
$$= \log p(\phi_3|\phi_2; \sigma_{3^{(i)}}) + \log p(\phi_1|\phi_2; \sigma_{V_3^{(i)}}) + \log p(\phi_2; \sigma_{V_3^{(i)}})$$
$$- \log p(\phi_3|\phi_2) - \log p(\phi_1|\phi_2) - \log p(\phi_2)$$
$$= \log p(\phi_3|\phi_2; \sigma_{3^{(i)}}) - \log p(\phi_3|\phi_2)$$

*When comparing the distributions of $\widehat{\mathbf{V}}$, interestingly the* log *of the determinant of the Jacobian cancels out. Combining the two, we get:*

$$\log p(V_3|V_2; \sigma_{V_3^{(u)}}) - \log p(V_3|V_2) = \log p(\phi_3|\phi_2; \sigma_{3^{(u)}}) - \log p(\phi_3|\phi_2) \tag{67}$$

*Taking the partial derivative now with respect to $V_1$, we get that the LHS equals 0 and the RHS becomes:*

$$0 = \frac{\partial}{\partial V_1} \log p(\phi_3|\phi_2; \sigma_{3^{(i)}}) - \log p(\phi_3|\phi_2)$$
$$= \frac{\partial \log p(\phi_3|\phi_2; \sigma_{3^{(i)}})}{\partial \phi_3} \frac{\partial \phi_3}{\partial V_1} + \frac{\partial \log p(\phi_3|\phi_2; \sigma_{3^{(i)}})}{\partial \phi_2} \frac{\partial \phi_2}{\partial V_1}$$
$$- \frac{\partial \log p(\phi_3|\phi_2)}{\partial \phi_3} \frac{\partial \phi_3}{\partial V_1} - \frac{\partial \log p(\phi_3|\phi_2)}{\partial \phi_2} \frac{\partial \phi_2}{\partial V_1} \quad \text{(Chain rule)}$$
$$= \frac{\partial \phi_3}{\partial V_1} \left( \frac{\partial \log p(\phi_3|\phi_2; \sigma_{3^{(i)}})}{\partial \phi_3} - \frac{\partial \log p(\phi_3|\phi_2)}{\partial \phi_3} \right)$$
$$+ \frac{\partial \phi_2}{\partial V_1} \left( \frac{\partial \log p(\phi_3|\phi_2; \sigma_{3^{(i)}})}{\partial \phi_2} - \frac{\partial \log p(\phi_3|\phi_2)}{\partial \phi_2} \right) \quad \text{(Collect terms)}$$

*Thus, we have two unknowns $\frac{\partial \phi_2}{\partial V_1}$ and $\frac{\partial \phi_3}{\partial V_1}$. Given the two interventions with different mechanisms on $V_3$ compared to the observational distribution, we have two equations that result in a 2-dimensional linear system. We are able to determine that $\frac{\partial \phi_2}{\partial V_1} = \frac{\partial \phi_3}{\partial V_1} = 0$ thus demonstrating that our approach disentangles $\hat{V}_3 = \phi_3(\mathbf{V})$ and $\hat{V}_2 = \phi_2(\mathbf{V})$ from $V_1$.* $\qquad\square$

### F.6 Comparing different identifiability results

In this section, we explicitly compare and discuss our work compared to a non-exhaustive list of related disentangled learning in the setting of causally related latent components. Different from previous literature, we do not make common assumptions such as (1) each intervention is applied to a single node [55]; (2) idle interventions (observational distribution) are present within each domain [21, 22]; (3) *exactly* one intervention is applied per node [13]; (4) *at least* one intervention is applied per node [21, 55].

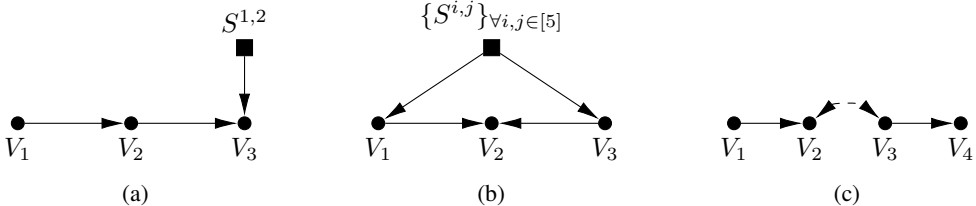

Figure S5: **Reproduced Fig. 4 for convenience.**

**Causal component analysis [21]**    The closest work to ours is [21], which also presupposes knowledge of the latent causal graph and focuses solely on learning the unmixing function and the distributions of the causal variables. In [21], the results emphasized the need for interventions that occur only on a single node in the latent causal graph. However, Lemma 5 demonstrates challenges that are not addressed in the prior work. In addition, in our work, we propose a more general concept of identifiability in Def. 2.3. As a result, Thm. 1 makes significantly weaker assumptions to still achieve identifiability. Exs.2-6 illustrate also the nuances addressed by our work, but not in [21].

Another interesting concept introduced by [21] is the "fat-hand" interventions, which intervene on groups of variables within different groups, and the concept of "block-identifiability".

Here, we illustrate some examples and discussion on how our work compares with that of [21] that also provides sufficient conditions for identifiability given a causal graph over the latent variables. One key difference between our work is that we do not assume Markovianity in the underlying SCM, whereas they do.

**Example** (Ex. 6 cont.).  *This example continues off of Ex. 6. Consider the motivating example in healthcare depicted in Fig. 2. In hospitals from different countries $\Pi^i$ and $\Pi^j$, drug treatment ($V_1$) affect length of ICU stay ($V_2$), and ultimately whether or not the patient lives or dies ($V_3$). Our task is to learn representations of the high-level latent variables ($V_1, V_2, V_3$) that are not collected given a collection of low-level input such as EMRs, imaging and bloodwork data (high-dimensional data $\mathbf{X}$). In existing work [21], there are no guarantees that variables $\{V_2, V_3\}$ are disentangled from their ancestor $V_1$ from soft interventions per nodes. However, Proposition 3 demonstrates two comparisons are enough to disentangle both $V_2$ and $V_3$ from their ancestor $V_1$.*  ☐

Even in the Markovian setting, where the LSG does not contain bidirected edges, our results can also guarantee identifiability in this setting.

**Example 22** ([21] approach).  *Given the graph shown in Figure 4(a), [21] requires an observational, and tuple of intervention sets $\mathbf{\Psi} = \langle\{\}, \{V_1\}, \{V_2\}, \{V_3\}\rangle$. Provided these four distributions, there is still no disentanglement of $\hat{V}_3$ with respect to any variables, $V_i \in \mathbf{V}$.*  ☐

**Causal Representation Learning from Multiple Distributions: A General Setting [22]**    Another approach to achieving disentanglement among the latent variables is similar to nonlinear-ICA, but leverages the conditional independence properties within a Markov Network of the causal graph. Then the proof strategy of [22] considers the second order derivative, which leverages the conditional independence constraints.

However, this results in a required $2d + |\mathcal{E}(M_G)| + 1$ number of distributions that satisfy Assump. 4. In addition, this strategy states that in a collider graph $V_1 \rightarrow V_2 \leftarrow V_3$, that $V_1$ is not ID wrt $V_2$, and $V_3$ is not ID wrt $V_2$.

Another example, continues off of Ex. 6.

**Example** (Ex. 6 cont.).  *This example continues off of Ex. 6. Consider the motivating example in healthcare depicted in Fig. 2. In hospitals from different countries $\Pi^i$ and $\Pi^j$, drug treatment ($V_1$) affect length of ICU stay ($V_2$), and ultimately whether or not the patient lives or dies ($V_3$). Our task is to learn representations of the high-level latent variables ($V_1, V_2, V_3$) that are not collected given a collection of low-level input such as EMRs, imaging and bloodwork data (high-dimensional data $\mathbf{X}$). According to [22], 10 distributions can disentangle $V_3$ from $V_1$ when $V_3 \perp\!\!\!\perp V_1 \mid V_2$. However, Proposition 3 demonstrates two comparisons are enough to disentangle both $V_2$ and $V_3$ from their ancestor $V_1$.*

**Linear ICA**    Linear ICA has been extensively studied over decades, and is applied in magnetic resonance imaging (MRI) [74], astronomy [75], image processing [76], finance [77] and document analysis [78]. In linear ICA settings, the generative factors are assumed to be independent of each other and the mixture function $f_{\mathbf{X}}$ is considered to be an invertible matrix $\mathbf{A} \in \mathbb{R}^{d \times d}$. Formally, the mechanism $\mathcal{F}$ and the distribution $P(\mathbf{U})$ of the true ASCM $\mathcal{M}^*$ are written as:

$$
\begin{cases}
V_j \leftarrow f_j(U_j), \forall j \in [d] \\
\mathbf{X} \leftarrow \mathbf{AV} \\
U_i \perp U_j, \forall i, j \in [d]
\end{cases}
\tag{68}
$$

Notice that $\mathbf{X}$ is $d$ dimensional variable here and $X_i \leftarrow \sum_{j=1}^{d} a_{ij} V_j = \mathbf{a}_i \mathbf{V}, \forall i \in [d]$. Given the observational distribution $P(\mathbf{X})$, the goal of linear tasks is to learn $\widehat{\mathbf{A}}$ such that $\widehat{V}_j$ is a scaling of a true underlying generative factors $V_i$, where $\widehat{\mathbf{V}} = \widehat{\mathbf{A}}^{-1} \mathbf{X}$. The scaling and permutation identifiability is defined as follows to denote the achievability of linear ICA tasks.

**Definition 6.9** (Scaling and Permutation Identifiability). The representation $\widehat{V}$ is said to be identifiable up to scaling and permutation $\mathbf{V}^{(2)} = \mathbf{CPV}^{(1)}$ if for every pair of ASCM $\mathcal{M}^{(1)}$ and $\mathcal{M}^{(2)}$ such that
(1) $P^{\mathcal{M}^{(1)}}(\mathbf{X}) = P^{\mathcal{M}^{(2)}}(\mathbf{X})$, $P^{\mathcal{M}^{(1)}}(\mathbf{X}; \sigma_{v_k}) = P^{\mathcal{M}^{(2)}}(\mathbf{X}; \sigma_{v_k})$;
(2) $\mathcal{M}^{(1)}$ and $\mathcal{M}^{(2)}$ are constrained by the modeling process in Eq. 68,
where $\mathbf{C} = \mathrm{diag}(c_1, \ldots, c_d)$ is a scaling diagonal matrix and $\mathbf{P}$ is a permutation matrix.    □

Def. 6.9 says that if every pair model $\mathcal{M}^{(1)}$ and $\mathcal{M}^{(2)}$ in linear ICA settings match the observational distributions, the generative variables can be transformed by permutation and scaling. This implies once one finds a proxy ASCM $\mathcal{M}$ that matches $P(\mathbf{X})$, $\widehat{V}$ is guaranteed to be a scale and permutation representation of the true generative variable if the identifiability is achieved. The next example illustrates ASCMs in linear ICA settings and Def. 6.9.

**Example 23** (ICA Identifiability Is Not Achieved). *We consider the three augmented generative processes $\mathcal{M}^*$, $\mathcal{M}^{(1)}$ and $\mathcal{M}^{(2)}$ with linear ICA constraints.*

$$
\begin{cases}
V_1 \leftarrow U_1, V_2 \leftarrow U_2 \\
X_1 \leftarrow V_1, X_2 \leftarrow V_2
\end{cases}
\qquad
\begin{cases}
V_1^{(1)} \leftarrow U_1, V_2^{(1)} \leftarrow U_2 \\
X_1 \leftarrow 2V_1^{(1)}, X_2 \leftarrow 0.5V_2^{(1)}
\end{cases}
\qquad
\begin{cases}
V_1^{(2)} \leftarrow U_1, V_2^{(2)} \leftarrow U_2 \\
X_1 \leftarrow \frac{\sqrt{2}}{2} V_1^{(2)} + \frac{\sqrt{2}}{2} V_2^{(2)} \\
X_2 \leftarrow \frac{\sqrt{2}}{2} V_1^{(2)} - \frac{\sqrt{2}}{2} V_2^{(2)}
\end{cases}
$$

$$
U_1, U_2 \sim \mathcal{N}(0, [1, 0; 0, 1]) \qquad U_1, U_2 \sim \mathcal{N}(0, [1/4, 0; 0, 4])
$$

$$
U_1, U_2 \sim \mathcal{N}(0, [1, 0; 0, 1])
$$

$$
\mathcal{M}^* \qquad\qquad\qquad \mathcal{M}^{(1)} \qquad\qquad\qquad \mathcal{M}^{(2)}
$$

*It is verifiable that $X_1, X_2 \sim \mathcal{N}(1, 0; 0, 1)$ induced by all three models. The latent generative variables in $\mathcal{M}^{(1)}$ are scaled and permuted representations of the true factors $\mathcal{M}^*$, namely $V_1^{(1)} = 2V_2^{(2)}$ and $V_2^{(1)} = 0.5V_2^{(1)}$. In other words, $V^{(1)}$ and $V^{(2)}$ distinctly represents $V_2$ and $V_1$ respectively. However, the representations $V_1^{(2)}$ and $V_2^{(2)}$ in $\mathcal{M}^{(2)}$ are mixture of true generative factors $V_1$ and $V_2$, i.e.,*

$$
\begin{aligned}
V_1^{(2)} &= \frac{2}{2} V_1 + \frac{2}{2} V_2 \\
V_2^{(2)} &= \frac{2}{2} V_1 - \frac{2}{2} V_2
\end{aligned}
\tag{69}
$$

*which implies this is not a scaling and permutation transformation. Thus, $\mathcal{M}^{(2)}$ demonstrates that the scaling and permutation identifiability is not achieved in this setting.*    □

The above example shows a famous result of linear ICA: the representations are not identifiable if generative factors follow a multi-gaussian distribution. This result comes from the symmetricity of gaussian distributions: any white gaussian variables are still white gaussian after an orthogonal transformation. However, orthogonal transformations are not guaranteed to be a scaling or permutation thus a proxy model may have generative factors that are mixtures of the true $\mathbf{V}$ ($\mathbf{V}^{(2)}$ in Example 23). Further, the identifiability result can be concluded as follows with the non-Gaussian assumption.

**Nonlinear ICA [7, 10]**    Compared to linear ICA, nonlinear ICA assumes the mixing function is a nonlinear bijective function (i.e. invertible and differentiable).

In linear ICA settings, the generative factors are assumed to be independent of each other and the mixture function $f_{\mathbf{X})}$ is considered to be an invertible matrix $\mathbf{A} \in \mathbb{R}^{d \times d}$. Formally, the mechanism $\mathcal{F}$ and the distribution $P(\mathbf{U})$ of the true ASCM $\mathcal{M}^*$ are written as:

$$\begin{cases} V_j \leftarrow f_j(U_j), \forall j \in [d] \\ \mathbf{X} \leftarrow \mathbf{f}_X(\mathbf{V}) \\ U_i \perp U_j, \forall i, j \in [d] \end{cases} \tag{70}$$

The traditional approaches for proving identifiability from [7, 8, 10] has the following settings:

- (Assumptions) A parametric exponential family is assumed in [7]. In addition, the causal assumptions of the latent variables is fully disconnected graph, where all variables are mutually independent. Our work assumes a nonparametric mixing model, and only requires the mixing function to be a bijection. In addition, we allow a non-Markovian causal model among the latent variables, which is the first to our knowledge to analyze identifiability in this general setting.

- (Data) Nonlinear ICA assumes that $2d + 1$ number of distributions with mechanism changes of the latent variables such that a version of the Assump. 4 holds. One instantiation of this in real-world data is time-series with non-stationary changes. Our work leverages arbitrary combinations of interventional data arising from multiple domains, and also does not necessarily require observational data.

- (Output) The focus of nonlinear ICA was typically on achieving disentanglement of latent variables up to scaling indeterminancy (Def. 6.5). Our work approaches the goal of identifiability from a more general setting according to Def. 2.3.

**Interventional causal representation learning [52]**    Another potentially promising approach to improving identifiability results lies in assuming a parametric form to the mixing function. [52] considers the setting of having a mixing function that is a composition of polynomial functions (i.e. a polynomial decoder).

Thus, [52] is able to achieve identifiability of latent variables up to an affine transformation:

$$\hat{V} = \mathbf{A}\mathbf{V} + c$$

where $\mathbf{A} \in \mathbb{R}^{d \times d}$ and $c \in \mathbb{R}^d$ make up an invertible affine transformation of the true latent variables $\mathbf{V}$. In our work, we consider a nonparametric form of the mixing function. However, future work could consider relaxing this assumption in the direction of a parametric mixing function with polynomial functions.

**Learning Causal Representations from General Environments: Identifiability and Intrinsic Ambiguity [25]**    In this paper, identifiability results for a linear ASCM with a linear mixing function is provided, with access to multi-distributional data arising from different environments.

In addition, they prove identifiability up to "surrounded-nodes" in [25, Thm. 3]. Specifically, any linear proxy model that is compatible with the observed distributions $\mathcal{P}$, the causal graph, and satisfies a few technical assumptions will achieve identifiability for each variable with respect to variables not in the surrounded-set. Similar to ancestral identifiability (Def. 6.6), surrounded-set disentanglement is a special case of our proposed identifiability definition (Def. 2.3). Our work proposes a graphical criterion and an algorithm for determining a causal disentanglement map, which may contain different disentanglements compared to a surrounded-set. Besides our notion of identifiability (goal), our paper also allows arbitrary distributions from multiple domains (input), and non-parametric non-Markovian ASCMs (assumptions).

**Definition 6.10** (Surrounded set from [79])**.** For two nodes, $V_i, V_j \in \mathbf{V}$ in graph G, we say that $V_i \in sur(V_j)$ if $V_i \in Pa_j$ and $Ch_j \subseteq Ch_i$.

**Identifying Linearly-Mixed Causal Representations from Multi-Node Interventions [80]** In this paper, the authors explore identifiability results in an ASCM with a linear mixing function, where interventions occur on multiple latent variables at the same time (i.e. multi-node interventions). Further, they assume that interventions are perfect interventions and sufficiently diverse, and have a sparse effect on the set of latent variables. Finally, their goal of identifiability is a full disentanglement, which is a special case of the general disentanglement we provide in Def. 2.3, where any variable may be ID w.r.t. a subset of latent variables.

Our paper also allows multi-node interventions within our graphical criterion (Props. 3, 4, and 5). In terms of the identifiability goal (output), ASCM model (assumptions), and distributions (input), our paper is more general notion of identifiability in the form of a causal disentanglement map (goal); our paper also allows arbitrary distributions (soft, and/or perfect interventions with same/different mechanisms) from multiple domains compared to only perfect interventions in a single domain that meet a sparsity constraint (input), and non-parametric non-Markovian ASCMs vs linear Markovian ASCMs (assumptions).

**Linear Causal Representation Learning from Unknown Multi-node Interventions [79]** In this paper, identifiability results are provided in a linear ASCM with a linear mixing function, where soft, or perfect interventions occur on multiple nodes. The authors establish full disentanglement results, or disentanglement up to ancestors, which is similar to the results demonstrated in "Causal Component Analysis" [21].

Our paper also allows multi-node interventions within our graphical criterion (Props. 3, 4, and 5). In terms of the identifiability goal (output), ASCM model (assumptions), and distributions (input), our paper is more general notion of identifiability in the form of a causal disentanglement map (goal); our paper also allows arbitrary distributions (soft, and/or perfect interventions with same/different mechanisms) from multiple domains (input), and non-parametric non-Markovian ASCMs vs linear Markovian ASCMs (assumptions).

**Learning Causal Representations from General Environments: Identifiability and Intrinsic Ambiguity [24]** In this paper, the authors propose a partial disentanglement goal in a linear, or non-parametric Markovian ASCM with a sparse causal structure.

Their notion of the disentanglement goal introduces so-called entanglement graphs (output), which is interestingly exactly what we call the causal disentanglement map. Though the proposed output is the same, the identifiability results are not the same even in the Markovian case.

In addition, in terms of the distributions leveraged (input), our work differs in considering arbitrary combinations of distributions (soft, or perfect, or observational) from heterogenous domains. In terms of modeling the ASCM (assumptions), our work considers completely non-parametric non-Markovian ASCMs instead of non-parametric Markovian ASCMs with sparse connectivity. In future work, we believe it will be interesting to explore the assumption of sparsity in the context of our work.

## G Experimental Results

### G.1 Synthetic data-generating process

We generate data according to latent causal diagrams shown in Fig. 4. Specifically, we analyze the chain graph $V_1 \to V_2 \to V_3$, and collider graph $V_1 \to V_2 \leftarrow V_3$ with different input distributions.

Each graph is constructed according to an ASCM, where the latent variables are related linearly:

$$V_i := \sum_{j \in Pa_i} \alpha_{i,j} V_j + \epsilon_i$$

where linear parameters are drawn from a uniform distribution $\alpha_{i,j} \sim U(-a, a)$, and the noise is distributed according to the standard normal distribution $\epsilon_i \sim \mathcal{N}(0, 1)$.

**Generating Multiple Domains** To generate a new domain, where $S^{i,j} \to V_i$ indicates a change in mechanism for $V_i$ due to the change in ASCMs between $M^i$ and $M^j$, we start from the first ASCM generated, and then we modify the distribution of the noise variable with a mean-shift.

**Generating Interventions Within Each Domain** To generate interventional datasets within each domain $\Pi^i \in \mathbf{\Pi}$, we modify the $\mathbf{M}^i \in M$ by additionally modifying the SCM, and shifting its mean for a variable. Therefore for distribution $k$ in $\Pi^i$, with perfect intervention $\mathbf{I}$, we will have:

$$V_k := \epsilon'_k, \quad \text{with } \epsilon'_k \sim \mathcal{N}(\mu_k, \sigma_k), \ \forall V_k \in \mathbf{I}$$

such that $\mu_k$ is not within $+/-1$ of any other distribution for variable $V_k \in \mathbf{V}$. This ensures the Assumption of Generalized Distribution Change (Assump. 4). With a soft intervention $\mathbf{J}$ that is not perfect:

$$V_k := \sum_{j \in Pa_k} \alpha_{i,j} V_k + \epsilon'_k, \quad \text{with } \epsilon'_k \sim \mathcal{N}(\mu_k, \sigma_k), \ \forall V_k \in \mathbf{J}$$

For each distribution over $\mathbf{V} \in \mathbb{R}^d$, we generate 200,000 data points resulting in $d \times 200,000$ data points in total for $N$ total distributions.

We modify the mean and the variance to ensure that the Assumption of distribution change is met (Assump. 4).

**Mixing function** In order to generate the low-level data $\mathbf{X}$, we will apply a mixing function $f_\mathbf{X}$ to the generated latent variables $\mathbf{V}$. Following [21, 51], to generate an invertible mixing function, we will use a multilayer perceptron $\mathbf{f_X} = \sigma \circ \mathbf{A}_M \circ ... \circ \sigma \mathbf{A}_1$, where $\mathbf{A}_M \in \mathbb{R}^{d \times d}$ for $m \in [1, M]$ denotes invertible linear matrices and $\sigma$ is an element-wise invertible nonlinear function. In our case, we will use the tanh functio as done in [81]:

$$\sigma(x) = tanh(x) + 0.1x$$

In addition, each sampled matrix $\mathbf{A}_i$ is re-drawn if $|\det \mathbf{A}_i| < 0.1$. This ensures that the linear maps are not ill-conditioned and close to being singular. Once the mixing function is drawn for a given simulation, it is fixed across all domains and interventions according to Assump. 4, and then $\mathcal{P}$ is drawn according to all ASCMs instantiated.

### G.2  Image Editing Using Disentangled Representations

We demonstrate qualitatively that the generalized disentanglement proposed in this work is valuable for downstream tasks, such as counterfactual image editing [27]. Consider the graph shown in Fig. S8. Specifically, we use our learned proxy model to generate initial images and perform interventions on learned representations $\widehat{\mathbf{V}}$ to edit images. We generate initial image samples from observational distribution of $\Pi_1$, and then perturb the relevant representations with random Gaussian noise to edit the image. This is done for the color of the bar, color of the digit, and the digit representations. If the learned $\widehat{\mathbf{V}}$ satisfy the CRID output disentanglement,

1. editing the color of the digit ($\sigma_{\widehat{V}_2}$) should keep the original digit and writing style but may change the color of the bar since $V_2$ has a causal effect on $V_3$.

2. editing the color of the bar ($\sigma_{\widehat{V}_3}$) should keep the original digit and writing style but may change the color of the digit since $V_3$ is not disentangled with $V_2$.

3. editing digit ($\sigma_{\widehat{V}_1}$) may change all variables since no disentanglement of $V_1$ is claimed by CRID.

The editing results are shown in Fig. S7. All editing results are aligned with the CDM output as expected, which are illustrated above. Specifically, Fig. S7(a) shows the learned VAE-NF model can change the color of the bar without arbitrarily changing the digit, or writing style. Fig. S7(b) shows the learned VAE-NF model can change the color of the digit without arbitrarily changing the digit, or writing style. Finally, Fig. S7(c) shows the learned VAE-NF model did not learn a disentangled representation for "digit". When perturbing the representation for digit, sometimes the digit does not change, while the color of the bar, color of the digit, or the writing style changes. This experiment also demonstrates one usage of CRID. Before training a model that is potentially computationally and time-intensive, one can leverage CRID to determine if their input data and input assumptions are sufficient for learning a relevant disentangled representation for their downstream task.

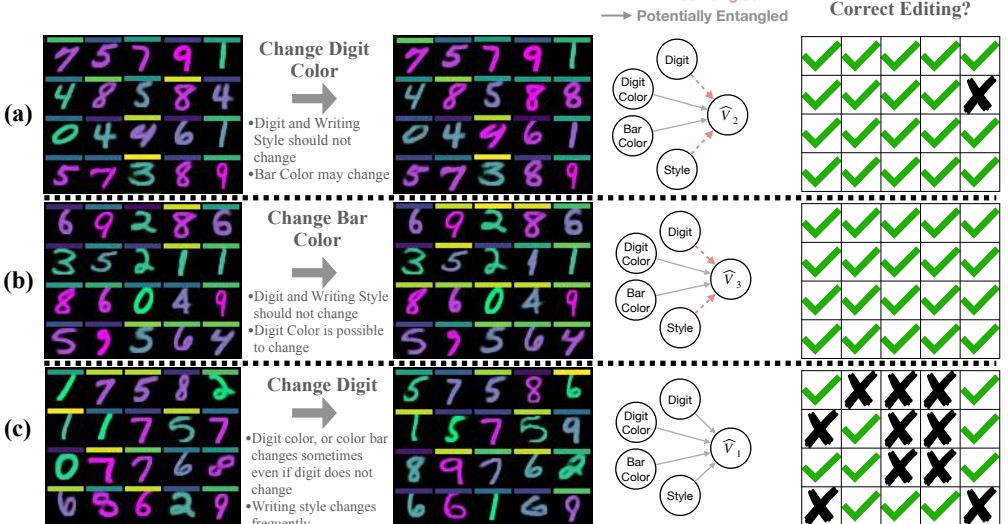

Figure S7: **Editing the image using the learned representations** - The representation of the color of the digit (a), color of the bar (b), and the digit (c) is perturbed. Only (a) and (b) show robust editing due to the learned representation being relatively disentangled as predicted by CRID.

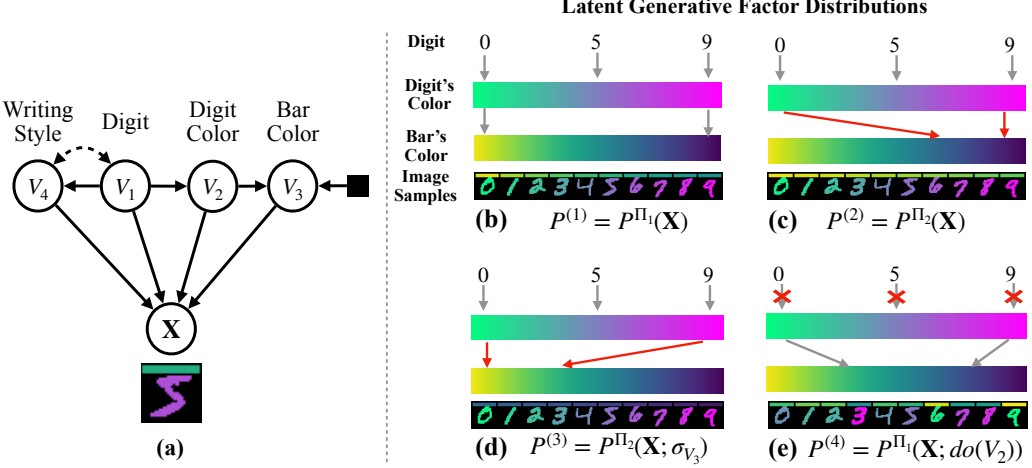

Figure S8: Color MNIST with bar data generation. The ground-truth LSD is shown in (a). Four distributions are generated with observations in domain $\Pi_1$ (b), observations in domain $\Pi_2$ (c), soft intervention on the color-bar ($V_3$) in $\Pi_1$ (d), and a perfect intervention on the color-digit ($V_2$) in $\Pi_1$.

### G.3 Model

We train invertible MLPs with normalizing flows. The parameters of the causal mechanisms are learned while the causal graph is assumed to be known. We leverage the implementation in [21], and extend it for our experiments.

The encoder is trained with the following objective that estimates the inverse function $f^{-1}$, and the latent densities $P(\mathbf{V})$ reproducing the ground-truth up to certain mixture ambiguities (c.f. Lemmas 3, 6). The encoder parameters is estimated by maximizing the likelihood..

**Normalizing flows**   We use a normalizing flows architecture [82] to learn an encoder $\mathbf{g}_\theta : \mathbb{R}^d \to \mathbb{R}^d$. Therefore, the observations $\mathbf{X}$ will be the result of an invertible and differentiable transformation:

$$\mathbf{X} = \mathbf{g}_\theta(\mathbf{V})$$

Specifically, $g_\theta$ will comprise of Neural Spline Flows [83] with a 3-layer feedforward neural network with hidden dimension 128 and a permutation in each flow layer.

**Base distributions**  Normalizing flows require a base distribution. We leverage one baseline distribution per sampled dataset, $(\hat{p}_\theta^k)_{k \in [d]}$ over the base noise variables $\mathbf{V}$. The conditional density of any variable is given by:

$$\hat{p}_\theta^k(v_i | \mathbf{Pa_i}) = \mathcal{N}\bigg( \sum_{j \in Pa_i} \hat{\alpha}_{i,j} v_j, \hat{\sigma}_i \bigg)$$

where the parameters are replaced by their corresponding counterparts if there is a change-in-domain, or an intervention applied. When a perfect intervention is applied, we have that:

$$\hat{p}_\theta^k(v_i) = \mathcal{N}(\hat{\mu}_i, \hat{\sigma}_i \bigg)$$

### G.4  Training details

We use the ADAM optimizer [84].We start with a learning rate of 1e-4. We train the model for 200 epochs with a batch size of 4096.

The learning objective is expressed as:

$$\theta^* = \arg \max_\theta \sum_{k=0}^{N} \big( \frac{1}{n_k} \sum_{n=1}^{n_k} \log p_\theta^k(\mathbf{X}^{(k)}) \big)$$

where $n_k$ represents the size of the dataset $P^k$, which is 200,000 in our simulations. We perform 10 training runs over different seeds for each experiment, and show the distributions of the mean-correlation coefficient (MCC). Using the output of Alg. 1, we compare variables that are expected to be entangled and disentangled. We use NVIDIA H100 GPUs to train the neural network models.

### G.5  Evaluation metrics

The output of our trained model is $\hat{\mathbf{V}} = g_\theta(\mathbf{X})$, which is a d-dimensional representation. We will compare this representation with our ground-truth latent variable distributions $\mathbf{V}$ by computing the mean correlation coefficients (MCC) between the learned and ground-truth latents. We expect there to be an overall lower MCC for variables that are predicted to be disentangleable by Alg. 1 relative to variables that are not deemed disentangleable.

Note that our algorithm is not shown to be complete, so there may be variables that are disentangled at the end of our training process that are not captured by the output of Alg. 1. Characterizing when this occurs and coming up with a complete theoretical characterization of disentanglement is a line for future work.

For the evaluation, we follow a standard evaluation protocol taken in prior work [18]. We expect low MCC values when predicting variables that are disentangled, and higher MCC values when predicting variables that are still entangled.

### G.6  Limitations

A major limitation of normalizing flows is that the input and output dimensions of the encoder must be the same. This is due to the fact that we wish to constrain the layers to be invertible transformations. It is easy to define invertible transformations for the same input/output dimensions, but it is non-trivial to do so when input/output dimensions vary widely.

Besides the technical limitations of the implementation, it is important to note that our theoretical results are asymptotic results. The theory claims we can achieve ID when the neural network is trained to zero error. However, in practice, this is not always simple to do and may require hyperparameter tuning and a very large sample size.

For example, when we consider Fig. 6, we observe that the disentanglement of (b,c) is significantly better than (a,d). In the experiment involving the collider graph from Fig. 4(b), we sample four distributions each with 200,000 samples, and thus we have almost 2x the data points compared to the settings in Fig. 6(a,c). We illustrate this point to emphasize that there is no correct way to set the sample sizes, hyperparameters, or model architecture as each simulation will be different. We chose a sample size, model architecture, and default hyperparameters based on prior literature [21] instead of biasing our experimental results by tuning significantly for each simulation.

### G.7 Discussion of Results

In Fig. S9, we show the MCC values for each learned latent representation $\hat{\mathbf{V}}$ and the corresponding ground-truth latents $\mathbf{V}$ for the three different LSDs shown in Fig. 4. Based on the causal disentanglement map (CDM) output from the CRID algorithm, the disentangled variables are shown in red, while the entangled variables are shown in gray.

In Fig. S9(a), the $MCC(\hat{V_3}, V_1)$ is low relative to the $MCC(\hat{V_3}, V_3)$, which is predicted by the CRID algorithm's CDM output (right plot). This suggests that $V_1$ is disentangled from $V_3$. In addition, we observe that all MCC values wrt $\hat{V_1}$ are relatively similar, which makes sense as we do not obtain any disentanglement wrt $V_1$ (left plot). CRID also predicts that $V_2$ is ID wrt $V_1$ (middle plot). However, we observe quite a large range of MCC values, possibly due to variance, default hyperparameter settings, or insufficient sample size. Importantly, this experiment verifies that two soft interventions on $V_3$ in the chain graph of Fig. 4(a) can ID $V_3$ wrt $V_1$, whereas previous literature suggested that $V_3$ is not ID wrt $V_1$ because $V_1 \in \mathbf{Anc}(V_3)$ [21].

In Fig. S9(b), we now have an observational, two soft interventions on $V_3$, and a perfect intervention on $V_2$. In addition to ID $V_2$ wrt $V_3$ (middle plot), we are also able to obtain full disentanglement of $V_1$ from $\{V_2, V_3\}$ (left plot). Interestingly, we are able to fully disentangle the representation for $V_1$ without intervening on it. This is the first theoretical (and empirical) result to our knowledge that shows this in a causal representation learning setting.

In Fig. S9(c), we have an observational and four interventional distributions applied on $\{V_1, V_3\}$ all with different mechanisms. We observe that $V_1$ and $V_3$ are fully disentangled. $MCC(\hat{V_3}, V_3) > MCC(\hat{V_3}, \{V_1, V_2\})$, and $MCC(\hat{V_1}, V_1) > MCC(\hat{V_1}, \{V_2, V_3\})$. CRID does not predict disentanglement for the $V_2$ representation (middle plot), yet interestingly we still see some disentanglement. [21] analyzes a similar setup using "fat-hand interventions", and the corresponding theory does predict $V_1$ and $V_3$ is ID wrt $V_2$. However, we also disentangle $V_1$ and $V_3$ from each other using many interventions. [22] presents a similar approach by leveraging $2d + |\mathcal{E}(M_G)| + 1$ distributions that "sufficiently change" (i.e. Assumption 4) to disentangle variables. However, the corresponding theory suggests that $V_1$ and $V_3$ are still entangled because they are adjacent in the Markov Network of G ($M_G$). These results demonstrate theoretically (and empirically) that $V_1$ and $V_3$ are in fact disentangled from each other in a fundamentally important causal graph (i.e. the collider).

In Fig. S9(d), we consider disentanglement in a non-Markovian LSD. We leverage two perfect interventions on $V_3$ (c.f. Lemma 6), and verify that even without observational distributions and the challenging setting of confounding among the latent variables, we can achieve disentanglement of $V_3$ wrt all other variables. $MCC(\hat{V_3}, V_3) > MCC(\hat{V_3}, \{V_1, V_2, V_4\})$, which is predicted by the CRID algorithm's CDM output (3rd plot from left). As expected, $V_1$ and $V_2$ are still fully entangled with all other variables (1st and 2nd plot from left).

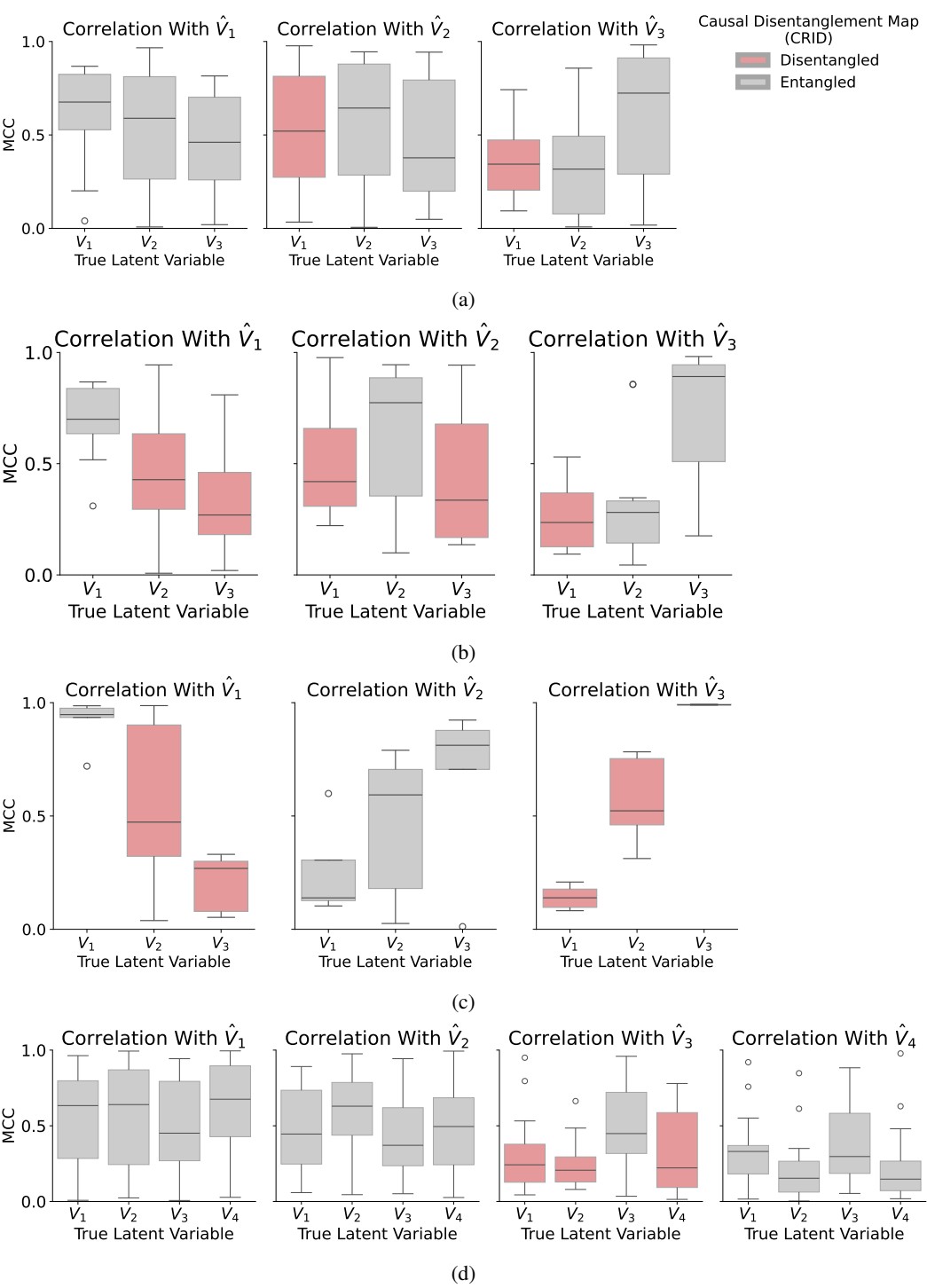

Figure S9: Mean correlation coefficient (MCC) of latent ground truth variables with the learned representation $\hat{\mathbf{V}}$, and expected disentanglement (red) according to the **CRID** algorithm. Each plot corresponds to an experimental setting using the graphs shown in Fig. 4: chain graph with two interventions on $V_3$ (a). chain graph with two interventions on $V_3$ and a perfect intervention on $V_2$ (b), collider graph with four interventions on $\{V_1, V_3\}$ (c) and the non-markovian graph with two perfect interventions on $V_3$ (d).

## H    Broader Impact and Forward-Looking Statements

The development of learning disentangled causal representations has the potential to improve our understanding of complex systems, and to help identify the generative factors for many important problems. By improving our ability to leverage observational and interventional data across multiple domains, this work could ultimately lead to more realistic generative AI. Beyond the machine learning and causal inference community, we expect that our results will enable fundamental contributions in various fields, including biology [85], epidemiology [86], economics [37] and neuroscience [38].

## I    Frequently Asked Questions

Q1. What's the learning goal of the paper? This work claims to be causal representation learning, but why do we not learn the structure over the latent variables while assuming it as given?

**Answer**.

Causal representation learning may comprise of two parts: i) learning the distributions of the latent variables and ii) learning the causal structure among these latent variables. Learning the distribution over latent variables is a non-trivial problem, especially in the context of non-Markovian ASCMs and the general multi-domain context. For example, consider nonlinear ICA, where the structure of the latent variables is the fully disconnected graph. It was shown to be non-ID with only iid data [9]. Although ID results eventually came about for nonlinear ICA, it was nontrivial to derive. In the same spirit, we seek to analyze the most general setting possible when assuming knowledge of the causal structure. This is analogous to the causal inference task of identification [87, 88], where the goal is to determine if a causal effect over observed variables is estimable given infinite data from some given distributions on the observed variables. Put similarly, our work's goal is to determine if a latent variable $V_i \in \mathbf{V}$ is disentangleable given infinite data from some given distributions over the observed variables $\mathbf{X}$. In traditional causal inference, when the causal graph is unknown, then one is typically interested in causal discovery, or structure learning of the graph over the observed variables given distributions over the observed variables. Future work may assume that even the latent causal structure is unknown, and pursue the structure learning of the LCG given distributions over the observed variables.

Q2. Is it reasonable to expect that the causal diagram is available? How do you get the graph?

**Answer**.

The assumption of the causal diagram is made out of necessity. Even existing methods is able to learn the casual diagram at the same time, however, the setting is more restricted. For example, the SCM should be Markovian and the intervention data per node should be given. In our setting, the underlying SCM can be non-Markovian and the given data can be any observational and interventional data from an arbitrary domain. In the general setting, even when the generative factors are all observed, learning the causal diagram task (structural learning task) is still difficult. Interestingly, recovering the full true diagram is even impossible, and existing works aim to recover an equivalence class of diagrams [32, 89–91]. Thus, in this general setting for causal representation learning, we first provide identification results given a causal diagram and leave structure learning for future work.

We follow closely to the disentangled representation learning works that assume the causal diagram is given. ICA/Nonlinear ICA assumes the diagram $G$ is given and restricts the setting where no edges are in $G$. Later, [18] assumes focus on disentangling the content variable from the style variable and assumes the knowledge of the diagram is given ($Content$ is the ancestral of $Style$). Recently, [21] focuses on the setting that the given diagram is Markovian. We extend the setting to non-Markovain settings. Notice that our generalization is not only related to diagram assumption but involves more general assumption, data, and output (please see Sec. 1, Tab. 1 and Tab. S1 for details.)

In practice, knowledge of the latent causal graph is typically provided by domain experts, or a modeling assumption. As an example of a realistic setting where the latent causal graph can be assumed, consider generating realistic face images [27]. Here, the latent causal structure comprises of Gender, Age, and Hair Color. Knowledge of the graph is provided due to our understanding of what comprises realistic changes in a face. For a detailed discussion on this, see Appendix Section D.1.

Q3. Why CRID (Alg. 1) only takes intervention targets $\mathbf{\Psi}$ and LSG $G^S$ as input? Do you need distributions $\mathcal{P}$? If not, how do you learn representations?

**Answer**. CRID leverages the intervention targets $\mathbf{\Psi}$ and the LSG $G^S$ to determine the invariant and changing factors when considering the generalized factorization of probability distributions Markov relative to the provided graph. These invariant and changing factors are what give rise to the theory we develop in Section 3. The CRID algorithm leverages this theory to provide an identifiability algorithm, which answers the question: If we fully learn a representation $\hat{\mathbf{V}}$ (given the diagram and the distributions), which variables are expected to be disentangled with which variables? This is an asymptotic question and assumes the representation is fully learned.

To fully learn the representations, one can search a proxy model that matches $\mathcal{P}$ and $\mathcal{G}^S$ and the $\hat{\mathbf{V}}$. Then the proxy model is the learned representation. We do this in the Experiments Section, but note we do not claim that this method of learning the representations is superior to any prior work. Specifically, we implement an approach to train a neural model that is compatible with the diagram to match the given distribution based on normalizing flows. Recently, many graphical constraints proxy neural models have been proposed, and they are trained to fit the given distribution for causal representation learning and downstream tasks [20, 27, 55, 92–94]. Without our work, one can still try to use these models to learn representations. However, there is no guarantee about how these learned representations is entangled with each other. Our work is the first one to provide general answers for this identification problem. This process can be compared with the identification and estimation problem in classic causal inference. The identification of a specific query given a causal diagram can be answered in symbolic ways [87, 95–99], and then if the query is identifiable, one can take the distribution (or data) as input and use estimation methods to obtain the estimated query. Without the identifiability result, there are no guarantees for the estimation.

Q4. Why not just use observational distributions in each domain as the baseline in the CRID algorithm described in Section 4?

**Answer**. One may surmise that this is not efficient and propose to choose the observational distribution in each domain alternatively. However, we argue that this enumeration is needed from two perspectives. First, the observational distributions, namely the idle interventions, are not always given. Second, comparing with observational distributions is not guaranteed to offer diverse $\Delta\mathbf{Q}$ sets. For example, consider intervention targets $\mathbf{I}^{(1)} = \{\}^{\Pi_1}, \mathbf{I}^{(2)} = \{V_1^{\Pi_1,[1]}, V_2^{\Pi_1,[1]}\}, \mathbf{I}^{(3)} = \{V_1^{\Pi_1,[1]}, V_2^{\Pi_1,[2]}\}$ all applied to the same domain $\Pi_1$. Choosing $\mathbf{T} = \{\}$ and comparing $\mathbf{I}^{(2)}$ and $\mathbf{I}^{(3)}$ with the idle intervention $\mathbf{I}^{(1)}$,

$$\Delta\mathbf{Q}[\mathbf{I}^{(2)}, \mathbf{I}^{(1)}, \mathbf{T}] = \Delta\mathbf{Q}[\mathbf{I}^{(3)}, \mathbf{I}^{(1)}, \mathbf{T}] = \{V_1, V_2\}. \tag{71}$$

Comparing $\mathbf{I}^{(1)}$ and $\mathbf{I}^{(3)}$ with the idle intervention $\mathbf{I}^{(2)}$,

$$\Delta\mathbf{Q}[\mathbf{I}^{(1)}, \mathbf{I}^{(2)}, \mathbf{T}] = \Delta\mathbf{Q}[\mathbf{I}^{(3)}, \mathbf{I}^{(2)}, \mathbf{T}] = \{V_2\}. \tag{72}$$

Then using Proposition 3, it is possible to disentangle $V_2$ from $V_1$ with the latter choice. This demonstrates that the observational distribution is not always necessarily the best baseline. Furthermore, consider the challenge of disentangling $V_1$ from $V_2$ in the LCG $V_1 \leftarrow\!-\!-\!-\!\rightarrow V_2$. As Lemma 6 demonstrates, one can compare two perfect intervention distributions on $V_1$ to achieve ID of $V_1$ wrt $V_2$. In this case, one would not even need the observational distribution.

Q5. Why distinguish domains and interventions? Are they not the same thing?

**Answer**. The literature has typically conflated domains and interventions in the context of causal inference.

Many examples across scientific disciplines demonstrate that the notions of domain/environment and interventions are distinct. For example, when making inferences about humans based on data from bonobos, this distinction becomes clear. The difference between the two species is depicted as the environment/domain in this context. A scientist might perform an intervention on a bonobo's kidney (specifically, what we're representing as $Z$), and try to determine the effect of medication ($X$) on fluid equilibrium in the body ($Y$). Although we could intervene on $Z$ in bonobos and observe its effect on $X$ and $Y$,

our ultimate goal might be to understand the effect of $X$ on $Y$ in humans. It's generally invalid to conflate these two qualitatively different indices, a point first noted by [62] in the context of transportability analysis. The distinct environments exist regardless of any intervention, such as medication. Also, an intervention on kidney function is different across the two species. [62] formalized this setting, introducing clear semantics for the S-nodes (environments) that essentially offer a combined representation for both environments. With this foundation, we can now address the more general problem of analyzing data generated from interventions across multiple domains in the latent space.

We point the reader to Appendix Section A.3 for a discussion and some examples of how CRID leverages this distinction. More recently,

In addition, we provide the following example that we hope further motivates the necessity of distinguishing interventions and domains.

**Example 24** (Disentangled representation with interventions in different domains). *Consider a ASCM, $\mathcal{M}$ over domains bonobos ($\Pi^1$) and humans ($\Pi^2$) that induces the causal chain $V_1 \rightarrow V_2 \rightarrow V_3 \leftarrow S^{1,2}$. The latent variables are sun exposure ($V_1$), Age ($V_2$), Hair Color ($V_3$). Sun exposure causes aging over time, and aging causes changes in hair color. Hair color looks different across species, which is represented by the S-node. We collect images of their faces, $\mathbf{X} = f_X(V_1, V_2, V_3)$. Assume we are able to collect images in two interventional settings $\{V_1\}^{\Pi_1}$ and $\{V_1\}^{\Pi_2}$, where we modify the level of sun exposure each participant is exposed to.*

*Now, assume we ignore the domain index, and simply treat these two distributions as interventional, since we are intervening on $V_1$. Then prior results would state that a soft (or perfect) intervention on $V_1$ allows it to be disentangled from $V_3$. However, this is incorrect.* ☐

Q6. Is the relaxation of Markovianity important? Since all $\mathbf{V}$ are already latent, can one regard the confounding $\mathbf{U}$ as $\mathbf{V}$ to transfer the model in the non-Markovianity setting to a Markovanity model?

**answer**
Yes the distinction between Markovianity and non-Markovianity is important both qualitatively and quantitatively.

Qualitatively, consider the following example in healthcare, where one has access to high-dimensional T1 MRI scans. Let the LCG comprise of Drug Treatment $\rightarrow$ Outcome, but they are confounded by socioeconomic status (Drug Treatment $\leftarrow$----$\rightarrow$ Outcome). The drug treatment and outcome are visually discernable on the MRI. However, socioeconomic status does not directly impact how the MRI appears, except through how it impacts the drug treatment efficacy or outcome. The socioeconomic status is therefore an unaccounted confounder in the LCG, and it is important to model this spurious association. If unaccounted for, one may assume that it is possible to disentangle Drug Treatment and Outcome leveraging existing ID results in the literature [11, 13, 14, 21, 22] even if the results do not apply in this setting.

Regarding modeling, an ASCM with confounding cannot be reduced to a Markovian ASCM. Although $\mathbf{U}$ and $\mathbf{V}$ are both latent, every $\mathbf{U}$ is not the direct parents of $\mathbf{X}$, which means $\mathbf{U}$ cannot be uniquely determined by value of $\mathbf{X}$. Take the example where $V_1 \leftarrow$----$\rightarrow V_2$ is the LCG $G$. Since $U_{12}$ does not point to $\mathbf{X}$, we cannot let $U_{12}$ be another latent generative factor $\mathbf{V}$.

Regarding results, we point the reader to Lemma 5, where it is shown that even with one perfect interventions per node, it is not possible to disentangle variables within the same c-component. This in contrast with results in the Markovian setting, where it is shown in [21] that one perfect intervention per latent variable allows us to achieve full identifiability of every latent variable up to scaling indeterminacies.

More broadly, it is noteworthy that transitioning causal reasoning from Markovian to non-Markovian settings was not trivial. For example, it is known that interventional distributions, such as $P(y \mid do(x))$, are always identifiable from the causal graph and observational distribution in Markovian settings in all models. Moving to non-Markovian settings, the celebrated do-calculus is developed primarily to address the decision problem of whether an interventional distribution can be uniquely computed from a combination of causal assumptions (in the form of a causal diagram) and the observational distribution [61].

Naturally, the issue of non-identifiability is much more acute in this setting, due to the existence of unobserved confounding.

