# OpenReview forum: "Disentangled Representation Learning in Non-Markovian Causal Systems"
_NeurIPS.cc/2024/Conference — NeurIPS 2024 poster_

### Official Review · Reviewer_fxDa · 2024-07-07

**Soundness:** 4
**Presentation:** 3
**Contribution:** 2
**Rating:** 5
**Confidence:** 3

**Summary:**

This paper attempts to disentangle latent variables from high-dimensional observational data (e.g., EEG data) as much as possible, based on knowledge of the causal structure between latent factors, high-dimensional observational data obtained by different interventions in several domains, and knowledge of the differences between the domains of those interventions.

**Strengths:**

* Authors introduce a relaxed notion of disentanglement, which allows checking if only a set of variables of interest is disentangled with other sets of variables. As it relaxes the need for all variables to be disentangled individually, it has the potential to deal with situations where existing methods can not disentangle in principle.
* Can also deal with situations where there are unobserved confounding factors in the graph.

**Weaknesses:**

[W1] The practicality of the proposed method is questionable, as it requires structural knowledge of observed data in several domains with different interventions and how they differ.

[W2] It is questionable what new value will be created by the new disentanglement notion (see Question 1). It may be a natural extension of the research stream on disentanglement, but rather the value should be clearly stated for readers outside of the specific research topic.

**Questions:**

[Q1] How can the relaxed disentangled estimation be used? For example, if V2 and V3 can be disentangled from other variables as Vtar, can we predict the outcome if we intervene on V2 and V3, even though V2 and V3 are entangled?

[Q2] What are the advantages of considering unobserved confounding factors? In this problem, V is also unobserved and is to be estimated from observed data. Therefore, as a simple approach, if such confounding factors are also considered as the target V of the disentanglement, the causal graph may be assumed to be a DAG. However, it must not collapse by f_X and there must be an inverse function. In other words, the proposed method has the advantage that it does not need to be able to be estimated by inverse mapping with respect to unobserved confounding factors. My question is: is that essentially important? And are there other advantages of allowing unobserved confounding factors in the latent causal graph?

**Limitations:**

See [W1].

---

> ### Author Rebuttal · Authors · 2024-08-07
>
> > W#1 “The practicality of the proposed method is questionable…”
>
> We address this question directly in Appendix I. FAQ Q2 (pg 46) and provide some details in **author’s rebuttal section II**.
>
> Firstly, expert knowledge can help infer causal structures. For instance, in Appendix D.1 (pg 31), it's known that "Age" directly affects "Hair Color" in faces.
>
> Beyond the CRL setting, the causal inference literature has decades of work demonstrating the utility of imposing causal assumptions in the form of a causal graph, which can be traced back to its origins (e.g., see Pearl’s original paper in 1995 [2]). We highlight a few [1-4]. [1-2] demonstrates that it is possible to identify causal effects given only observational data and knowledge of the causal graph. [3-4] shows how to estimate counterfactual quantities. [5] discusses external validity and the transportability of causal effects across domains. Similar to these prior works, our paper demonstrates how to leverage the knowledge of the latent causal structure (assumptions), along with the data distributions (input) to achieve a generalized notion of identifiability (output).
>
> Finally, our work complements methods that learn causal graphs. One can leverage our work, and propose further assumptions that enable simultaneously learning the latent causal structure.
>
> Further insights are available in FAQ Q1 (pg 46).
>
> [1] Pearl. Causality: Models, Reasoning, and Inference. 2000.
>
> [2] Pearl, Judea. “Causal Diagrams for Empirical Research.”1995.
>
> [3] Pearl, "Probabilities of causation: three counterfactual interpretations and their identification," 1999.
>
> [4] Pearl. "Direct and indirect effects." 2001.
>
> [5] Bareinboim and Pearl, "Causal inference and the data-fusion problem," 2016.
>
> > W#2 “...what new value…disentanglement notation”
>
> First, we argue that relaxing full disentanglement to achieve causal disentanglement has practical value. Often, full disentanglement of all variables isn't necessary. For example, in the medical scenario with EEG data (L65-82), clinicians may only need to disentangle drug treatment from seizures, not all variables. Achieving full disentanglement typically requires specific distributions, such as single-node interventions for each latent variable, which are often unavailable or hard to obtain. Thus, allowing arbitrary combinations of distributions and focusing on partial disentanglement is more practical in real-world settings.
>
> To help readers outside the specific research topic understand the impact of learning partially disentangled representations, we have added to the text:
>
> “As another example to motivate learning a representation that only disentangles a subset of variables, consider a marketing company generating face images for a female product. The relevant latent factors are Gender $\leftrightarrow$ Age $\rightarrow$ Hair Color. If Gender and Age are entangled, altering Age could also change Gender, which is undesirable. The company needs a model where Age is disentangled from Gender but can remain correlated with Hair Color. Thus, the company doesn’t need to ensure disentanglement between Age and Hair Color. Our paper addresses the generalized identifiability problem of determining the sufficient training data for learning such a disentangled representation.”
>
>
> > Q#1 “How can the relaxed disentangled estimation be used?”
>
> Consider a motivating example as described in Appendix D.1 pg 32, where Gender $\leftrightarrow$ Age $\rightarrow$ Haircolor. If Age is disentangled from Gender, but is entangled with Haircolor, then after editing the representation of Age, we would expect that Gender does not arbitrarily change, while Haircolor may change. This flexibility may be useful in the context where the user does not care about the entanglement of Haircolor and Age, but does care a lot about keeping faces the same gender during image generation (i.e. not spuriously turning a woman's face into a male face, and vice-versa).
>
> In the context of your example, if we have V1 -> V2 -> V3 -> V4, and V2 and V3 can be disentangled from Vtar = \{V1, V4\}, then one can predict that variables in Vtar will not arbitrarily change when intervening on V2 and/or V3, unless they are causal descendants of V2 and/or V3. For example, perturbing V2 will translate into changes downstream of V2, such as V4 (and V3 since we assume they are still entangled). However, perturbing V2 will not spuriously change V1 since we assume in this example that the representation of V2 is disentangled from V1.
>
> > Q#2 “advantage…unobserved confounding factors”
>
> The non-Markovianity applies to the underlying SCM over V, not the ASCM including X. Unobserved confounders (U) are not part of V and do not generate X. We expand on this in the **author’s rebuttal IV.**
>
> Unobserved confounders are crucial as they allow for causal relationships within V that cannot be encoded by a Markovian model but exist in the real world. For concreteness, consider V1 and V2 generating X. In Markovian settings, three causal relationships are considered: (1) V1 and V2 are independent; (2) V1 -> V2; (3) V1 <- V2. Now consider that there exists a common cause, U, which both affects V1 and V2 (i.e., V1 <- U -> V2). This cannot be encoded through the Markovian language. Here, U is not a parent of X and does not mix with V1 and V2 to generate X. Introducing unobserved confounders is necessary for modeling general causal relationships in CRL.
>
> For concreteness, consider the discussion in Q1, where Gender and Age are spuriously correlated because there are many old males and young females in the dataset due to face images being collected in a non-diverse setting. This unobserved confounding though does not manifest itself in the face image, except through the causal factors Gender and Age. The causal relationships between generative factors Gender and Age (V) are assumed from domain knowledge, but they cannot be assumed independent or to have a direct effect on each other.

---

> > ### Author Response · Authors · 2024-08-12
> >
> > Dear Reviewer fxDa,
> >
> > We believe we have addressed your concerns, but since the discussion period is ending soon, we would like to double check if we could provide any further clarification or help to provide other comments about our work. We hope to be able to engage in a thoughtful and constructive exchange.
> >
> > Thank you again for spending the time to read our paper, and providing feedback.
> >
> > Authors of the paper #13821

---

> > > ### Comment · Area_Chair_Y6Z2 · 2024-08-13
> > >
> > > Dear reviewer,
> > >
> > > As the author-reviewer discussion period is coming toward an end on Aug 13, it would be helpful to have you read and acknowledge the author rebuttal, and/or update your review if your opinion of the paper has changed. Thank you!
> > >
> > > Best,
> > >
> > > Area chair

---

> ### Comment · Reviewer_fxDa · 2024-08-14
>
> Thank you for your answer. For W1, I still think the applicability is fairly limited. For W2 and Qs, the authors' response makes sense, thus I raised my score.
>
> > Gender and Age are spuriously correlated because there are many old males and young females in the dataset due to face images being collected in a non-diverse setting. This unobserved confounding though does not manifest itself in the face image, except through the causal factors Gender and Age.
>
> This is an example of selection bias rather than confounding.

---

> > ### Author Response · Authors · 2024-08-14
> >
> > Dear Reviewer fxDa,
> >
> > Thank you again for reading our paper and providing feedback during the process.
> >
> > >> Gender and Age are spuriously correlated because there are many old males and young females in the dataset due to face images being collected in a non-diverse setting. This unobserved confounding though does not manifest itself in the face image, except through the causal factors Gender and Age.
> >
> > > This is an example of selection bias rather than confounding.
> >
> > This example was intended to illustrate that there are spurious associations between Gender and Age not accounted for. In this example, the confounder is the place in which the data was collected.
> >
> > More broadly, evaluation is non-trivial in causal inference since a dataset under observational and experimental conditions are needed, one in which the system has access and the other with ground truth. These two datasets are almost never available in the real world. Perhaps surprisingly, it’s not unprecedented that a trick based on selection is used to synthesize confounding, or spurious association between variables. For example, check Kallus and Zhou, NeurIPS’18,  Zhang and Bareinboim, NeurIPS’19, to cite a few.  Since the comment was posted at the very last minute, we will keep this message short, in particular, the philosophical implications of this note.
> >
> > Having said that, we will add a note in the example to be more explicit of the role of the confounder.
> >
> > Authors of the paper #13821

---

### Official Review · Reviewer_zg4p · 2024-07-11

**Soundness:** 4
**Presentation:** 3
**Contribution:** 4
**Rating:** 7
**Confidence:** 2

**Summary:**

The authors propose a method called CRID for Disentangled Representation Learning in non-Markovian Causal settings (i.e. the latent variables are dependent and there might be unobserved confounders). The two sets of variables A and B are understood to be disentangled if the learned representations of A are a function of all variables excluding those in B, which is a more general definition than identifiability up to scaling, permutation etc. common in the literature. The method assumes that we have different domains such that the mixing function (which we are aiming to "undo" for disentanglement) and the causal graph are shared between the domains, while the mechanisms in the structural causal model (SCM) vary across the domains. We also assume a set of interventions applied across the domains. The authors theoretically derive a graphical criterion for disentanglement by (as I understood) first separating the variables potentially affected by a confounder (\Delta Q) from other variables, and second by considering disentanglement within the confounded variables. CRID is a algorithm based on these graphical disentanglement criteria.

**Strengths:**

+ Interesting and well-motivated problem
+ Through and rigorous theoretical analysis
+ A general setting not requiring particularly strong assumptions and thus potentially applicable to a variety of scenarios

**Weaknesses:**

- Experiments are limited to synthetic datasets with a handful of variables and fairly simple causal graphs
- The text is very dense with technical terms, abbreviations, definitions and results making it a bit hard to follow. It seems almost that there are too many results for a single conference paper

**Questions:**

- What do you think about the scalability of CRID to real-world data (e.g. similar to Fig. 1) with complex confounders and mixing functions? Do you think the proposed algorithm is applicable to such cases, and if not what do you think are the weak point to be addressed?

**Limitations:**

The limitations are adequately addressed (NeurIPS checklist)

---

> ### Author Rebuttal · Authors · 2024-08-07
>
> ## New Experiment Details
>
> The images are a modification of the popular MNIST dataset with a bar and color added (to both the digit and the bar). Each digit in the original database contains many examples with different writing styles. The goal is to demonstrate that a disentangled representation can robustly modify the color of the bar and digit without spuriously changing other latent factors.
>
> Exp details: Data is generated according to Fig. 2 (in **author’s rebuttal pdf**), where the digit ($V_1$) causes the color of the digit ($V_2$), which in turn causes the color of the bar ($V_3$). Each digit is written in its own style ($V_4$), and is confounded by the setting in which the writer wrote the digit. With observational data from two domains, soft intervention on the distribution of color-bar, and a hard intervention on the color-digit, CRID outputs that $V_2,V_3$ are disentangled from $V_1,V_4$, a new identifiability result compared to the prior literature. Fig. 3 verifies this quantitatively since the correlation of the representations w.r.t. disentangled variables are lower than w.r.t. entangled variables. Since this evaluation requires the ground-truth of latent factors while we do not have access to the truth writing style ($V_4$), and thus only show correlations w.r.t. the others. However, this can be verified qualitatively through Fig. 4. Upon modification of $V_2$ and $V_3$’s representation, we note that robust changes in only the color-digit/color-bar are observed, while no spurious changes in the digit or writing style take place.
>
> > W#1 “Experiments are limited...”
>
> We want to clarify that the proposed work’s goal is to **identify** the presence of disentanglement in fully learned representations. The experiments in this work are used for verifying our identifiability results.  Compared with similar identifiability literature, there are many works that focus on the theoretical side (identifiability) and only use pure simulation data to verify their theoretical results (e.g., see [1-8]).
>
> Having said that and based on your suggestion,  we have included an additional experiment on hand-written digit images to highlight the potential impact of learning disentangled representations.
>
> [1] Hyvarinen et al., "Nonlinear ICA using auxiliary variables and generalized contrastive learning," 2019.
>
> [2] Khemakhem et al., "Variational autoencoders and nonlinear ICA: A unifying framework," 2020.
>
> [3] Liang et al., "Causal component analysis," 2024.
>
> [4] Zhang et al., "Causal representation learning from multiple distributions: A general setting," 2024.
>
> [5] Varici et al., "General identifiability and achievability for causal representation learning," 2024.
>
> [6] von Kügelgen et al., "Nonparametric identifiability of causal representations from unknown interventions," 2024.
>
> [7] Varici et al., "Score-based causal representation learning with interventions," 2023.
>
> [8] Sturma et al., "Unpaired multi-domain causal representation learning," 2024.
>
>
> > W#2 “The text is very dense”
>
> Thank you for noting the many results we try to introduce in this work. We also agree that there are numerous concepts that must be introduced in order to arrive at the theoretical contributions. The submission includes a table of the various notations in Appendix A.1 (pg 16), and also Ex. 1 to introduce the notation (L198-203).
>
> To further help the introduction of the theory, we will add a paragraph describing the assumptions informally to help visualize the critical technical assumptions in a conceptual manner. We anticipate this will help readers conceptualize the technical content better. In addition, we will add additional examples that conceptualize the notation and definitions.
>
> > Q#1 “What do you think about the scalability of CRID to real-world data…”
>
> CRID Algorithm Applicability: Yes, the CRID algorithm produces a causal disentanglement map (CDM) that indicates what can be disentangled based on data and latent causal structure assumptions.
>
> Let us elaborate. CRID answers: “Given my distributions and causal assumptions, what is identifiable?” It generates a Causal Disentanglement Map from the selection diagram and interventional targets but does not estimate or learn representations, thus handling complex confounders among latent variables, and any mixing function. Note the identifiability of variables may change with different confounders.
>
> Weaknesses: CRID determines identifiability but does not estimate representations with complex confounders and mixing functions. Estimating representations involves building a proxy ASCM model compatible with the selection diagram and observed distributions.
>
> Existing prior work (and our work Def. 1) suggests that the mixing function is a diffeomorphism, making normalizing flows a common approach. However, these require matching dimensions between latent variables and data, which can be problematic with high-dimensional data. Future research could focus on models bridging low-dimensional latent factors and high-dimensional data.
>
> Summary: Our paper focuses on identifiability, not on developing new models for learning representations. In the **author's rebuttal with pdf**, we show a disentangled learning experiment on a “causal” MNIST dataset to inspire further research into learning disentangled representations using our theoretical framework.

---

> > ### Comment · Reviewer_zg4p · 2024-08-12
> >
> > Thank you very much for the thorough rebuttal! I confirm my positive view of this paper, though I think I am less knowledgeable than other reviewers about the technical details, so these other reviews should have a higher weight than mine. However, I still think that a conference paper might be not the best way to publish these results (e.g. a longer journal article might be more suitable as it provides more space to introduce the background and the theory which could help improve the clarity of presentation).

---

> > > ### Author Response · Authors · 2024-08-14
> > >
> > > Dear Reviewer zg4p,
> > >
> > > Thank you again for spending the time reading our paper and providing valuable feedback during the process. We are glad you have a positive view of our paper. We re-affirm that we will have more conceptual discussion and examples to ground the technical content in the main text using our additional page and appendix.
> > >
> > > Authors of the paper #13821

---

### Official Review · Reviewer_SjE4 · 2024-07-13

**Soundness:** 3
**Presentation:** 3
**Contribution:** 3
**Rating:** 5
**Confidence:** 3

**Summary:**

This paper tackles the problem of causal disentangled representation learning in non-Markovian causal systems, which include (1) unobserved confounding and (2) arbitrary distributions from various domains. The authors introduce new graphical criteria for disentanglement and an algorithm to map which latent variables can be disentangled based on combined data and assumptions. Their method is theoretically sound and validated through simulation experiments.

**Strengths:**

The problem that this work studies is vital in the causal representation learning community, especially the generalization of non-Markovian causal systems as well as the partial disentanglement based on the target variables.

The theoretical analysis is good with informative supporting examples.

**Weaknesses:**

There is a gap between the inspiring example provided in the paper and the experiment results. In the Introduction, the authors mentioned EEG data and in Appendix D1, the authors mentioned face examples. But the experiments are purely on simulation data, which leaves a gap between the theoretically sound method and the real problem solutions. Can the authors provide some reason why they didn't apply the proposed method on those inspiring tasks or what would be the blocking issue to do so?

The simulation result, especially the figures, are not supported with detailed discussion, at least in the main text. For example, in Figure 5, the authors claim the comparison of MCC values leads to some conclusion on which variables are disentangled or not. However, in the figure, the error bars or the confidence intervals are heavily overlapped. Can the authors explain to what extent we can trust the output of those models?

Minor issue:
- Typo: line 127 f_x: R^{d} \to R^{n} should be R^{m} to match the dimension of X.

**Questions:**

See questions in Weaknesses

**Limitations:**

Yes.

---

> ### Author Rebuttal · Authors · 2024-08-07
>
> > W#1 “There is a gap …experiments”
>
> We want to clarify that the proposed work aims to **identify** if a learned representation will exhibit disentanglement. This is a separate question from **estimating** the disentangled representations, and **using** said representations in a downstream task, such as realistic data simulations.
>
> The contributions on identification are of a theoretical nature and work for both low and high-dimensional mixture X. To illustrate, the CRID algorithm is proposed to identify the disentanglement of learned representations. CRID is a pure symbolic method that takes the diagram and intervention targets as input but does not require any training procedure. Consider the EEG example in the introduction (L65-82). With the input $G^S$ and intervention targets described in Ex.1, we can obtain the causal disentanglement map as shown in Figure 4 (please see Ex. 6 for details.)
>
> The experiments in the paper are used for verifying the identifiability results. In the literature, there are many causal disentangled representation learning works that focus mainly on the theoretical problem of identifiability and also only use simulation data to verify their theoretical results [1-9].
>
> We understand that seeing more than pure simulated data would be compelling, which led us to include an additional experiment on hand-written digit images. We hope this can illustrate the potential impact of learning disentangled representations. In particular, the images are a modification of the popular MNIST dataset with a bar and color added (to both the digit and the bar). Each digit in the original database contains many examples with different writing styles. The goal is to demonstrate that a disentangled representation can robustly modify the color of the bar and digit without spuriously changing other latent factors.
>
> Exp details: Data is generated according to Fig. 2 (in **author’s rebuttal pdf**), where the digit ($V_1$) causes the color of the digit ($V_2$), which in turn causes the color of the bar ($V_3$). Each digit is written in its own style ($V_4$), and is confounded by the setting in which the writer wrote the digit. With observational data from two domains, soft intervention on the distribution of color-bar, and a hard intervention on the color-digit, CRID outputs that $V_2,V_3$ are disentangled from $V_1,V_4$, a new identifiability result compared to the prior literature. Fig. 3 verifies this quantitatively since the correlation of the representations w.r.t. disentangled variables are lower than w.r.t. entangled variables. Since this evaluation requires the ground-truth of latent factors while we do not have access to the truth writing style ($V_4$), and thus only show correlations w.r.t. the others. However, this can be verified qualitatively through Fig. 4. Upon modification of $V_2$ and $V_3$’s representation, we note that robust changes in only the color-digit/color-bar are observed, while no spurious changes in the digit or writing style take place.
>
> [1] Hyvarinen et al., "Nonlinear ICA using auxiliary variables and generalized contrastive learning," 2019.
>
> [2] Khemakhem et al., "Variational autoencoders and nonlinear ICA: A unifying framework," 2020.
>
> [3] Liang et al., "Causal component analysis," 2024.
>
> [4] Zhang et al., "Causal representation learning from multiple distributions.." 2024.
>
> [5] Varici et al., "General identifiability and achievability for causal representation learning," 2024.
>
> [6] von Kügelgen et al., "Nonparametric identifiability of causal representations from unknown interventions," 2024.
>
> [7] Varici et al., "Score-based causal representation learning with interventions," 2023.
>
> [8] Sturma et al., "Unpaired multi-domain causal representation learning," 2024.
>
> [9] Lachapelle et al., "Disentanglement via Mechanism Sparsity Regularization…" 2022.
>
> > W#2 “The simulation result, … not supported with detailed discussion…”
>
> The simulation results are obtained over 10 different random seed runs with a fixed set of hyperparameters, such as learning rate, MLP normalizing flow architecture, and max epochs. Note: disentangled representation learning theory (in general and in our paper) holds in the setting of infinite data and perfect optimization. However, given finite data and numerical optimization, this will generally not be perfect. Thus we ran multiple trials to verify that on average disentanglement was achieved. We also point out that our paper does not claim to propose a robust method for **learning** the representations, which is a separate problem.
>
> Re MCC (correlation) values and how to interpret: According to ID definition (Def. 2.3, pg 5), if there exists a function $\tau$ such that $\widehat{V}_j = \tau(V^{en})$, then $V_j$ is disentangled from $\mathbf{V} \backslash V^{en}$. Thus the simulation results show MCC values between $V_k$ and $\widehat{V}_j$ are relatively lower when $V_j$ is disentangled from $V_k$. In contrast, MCC values will be higher on average when variables are entangled. The MCC values in Fig. 5 show on average that variables that were expected to be disentangled by CRID were in fact relatively less correlated to their entangled counterparts.
>
> For instance, Fig. 5(a) (in manuscript) considers a causal chain $V_1\rightarrow V_2\rightarrow V_3$. Based on prior work [1], with a single-node soft intervention on each variable, one can achieve disentanglement up to ancestors; i.e. $V_3$ would still be entangled with $V_1$. However, in line with the motivation of this paper, what if one does not have a single-node intervention per latent variable? Figure 5(a) demonstrates that with two $V_3$ soft interventions, one can disentangle $V_3$ from $V_1$ as evidenced by the average correlation of $\widehat{V}_3$ with V1 (MCC=0.3) being lower than the correlation of $\widehat{V}_3$ and $V_3$ (MCC=0.8).
>
> We have added text to the Experiments section describing the above setting to provide more discussion.

---

> > ### Comment · Reviewer_SjE4 · 2024-08-10
> >
> > Thank you. I've read your rebuttal, responses, and the other reviews. I will keep my score for now since it is already positive.
> >
> > Also, can the authors elaborate more about
> >
> > > We also point out that our paper does not claim to propose a robust method for learning the representations, which is a separate problem.
> >
> > with respect to my previous question
> >
> > > However, in the figure, the error bars or the confidence intervals are heavily overlapped. Can the authors explain to what extent we can trust the output of those models?

---

> > > ### Author Response · Authors · 2024-08-10
> > >
> > > Thank you for the reply, we are happy to elaborate more on your question. We start by providing some context to ground our response. There are two tasks involved in disentangled causal representation learning (CRL): (1) identification and (2) estimation.
> > >
> > > First, when the causal variables $\mathbf{V}$ are fully observed, observational data alone cannot always estimate causal effects, even with infinite data [1]. Identification determines if the causal graph's assumptions allow observational data to estimate the desired effect [1]. Pearl’s celebrated do-calculus, for example, solves the problem of identifiability – linking a causal quantity to the input distribution (Biometrika 1995). If identifiable, estimation methods can then use observational data to evaluate the target effect. Without identification guarantees, estimation is not applicable [1].
> > >
> > > Armed with the understanding of the dichotomy between identification and estimation, we turn our attention to the CRL setting discussed in our paper. Consider a learned representation $\widehat{V}$ and the true latent set of variables $\mathbf{V}$. When the causal variables $\mathbf{V}$ are unobserved, one would like to learn a representation $\widehat{V}$ that is “similar” to $\mathbf{V}$. The notion of similarity appears in different forms, including i) full disentanglement: $\widehat{V}_i = \tau(V_i)$ for every $i$, ii) ancestral disentanglement: $\widehat{V}_i = \tau(Anc(V_i))$, and iii) general disentanglement (proposed in this work): $\widehat{V}_i = \tau(V^{en})$, where $V^{en}$ is a set of entangled variables w.r.t. $\widehat{V}_i$.
> > >
> > > Identification in the context of CRL is concerned with the problem: given input assumptions (e.g., causal graph), input distributions, and the desired disentanglement goal, is it possible to learn a representation that satisfies these properties? Estimation here addresses the problem of developing estimators that learn representations ($\widehat{V} = f^{-1}_{X}(X)$) given finite data, once identification is satisfied. Many prior works consider one of these tasks [3-10].
> > >
> > > Both identification and estimation are highly non-trivial, fundamental tasks within CRL. If one estimates representations without identification, there is no guarantee the output will be disentangled and any of the downstream inferences will be meaningful. For example, some prior works have studied estimation without considering identification (e.g. beta-VAE) [3-5]. Fortunately, it is understood since Locatello et al. [6] that in general, identifiability is not guaranteed for CRL, so additional understanding and formal conditions are needed (e.g., [2,7-9]).
> > >
> > > Given the discussion above, the proposed work primarily studies the identification task [specific contributions, L86-93] with the goal of returning a disentanglement map. Specifically, the proposed algorithm (CRID) determines identifiability given a general set of input assumptions, input data, and desired disentanglement goals. The experiments here are designed to empirically corroborate our theoretical results, which is a common approach taken in the identification literature [7, 9, to cite a few].  Specifically, we used synthetic data following the protocol in the field, employing a normalizing flow method to obtain representation $\widehat{V}$ and verify that CRID's disentanglement aligns with the relative MCC calculated w.r.t. the ground truth latent variables. We have not claimed the estimation method works for complex high-dimensional data, as our focus was on the identification task. Identification is crucial before estimation, and both are necessary for learning valid causal representations. To address the reviewer’s comments, we conducted more refined experiments, using a VAE-NF structure for the image dataset, which is more practical than previous approaches that mainly used synthetic data.
> > >
> > > **Summary.** In the spirit of the constructive exchange we have had so far, we note that we have answered all your questions and even performed new MNIST experiments upon your request (Fig. 3,4 in [pdf rebuttal](https://openreview.net/attachment?id=W6rCNuQqw3&name=pdf)). Of course, we would be happy to follow up in case you feel there are still unresolved issues. In terms of the paper’s significance, while there is always more work to do, we note that it provides the most general CRL identification result without requiring the Markov assumption in very general non-parametric settings. It also reduces the experimental design complexity in Markovian settings (e.g., see example L1278 in p. 39).  Considering this, we would add that we appreciate your kind words referring to your evaluation (“since it is already positive”), but we note that it may not be perceived as such by the system, since a score of 5 refers to a “weak accept”. We wholeheartedly appreciate the opportunity for engagement and would like to ask you to reconsider our paper based on the clarifications provided above and in the rebuttal.

---

> > > > ### Author Response · Authors · 2024-08-10
> > > >
> > > > References for reply to [SjE4](https://openreview.net/forum?id=uLGyoBn7hm&noteId=dDKhmvrVyF)
> > > >
> > > > [1] Pearl. Causality: Models, Reasoning, and Inference. 2000.
> > > >
> > > > [2] Hyvärinen. Nonlinear independent component analysis: Existence and uniqueness results. 1999.
> > > >
> > > > [3] Chen, et al. "Isolating sources of disentanglement in variational autoencoders." (2018).
> > > >
> > > > [4] Higgins, et al. "beta-vae: Learning basic visual concepts with a constrained variational framework." ICLR, 2017.
> > > >
> > > > [5] Kim, et al. "Disentangling by factorising." International conference on machine learning. PMLR, 2018.
> > > >
> > > > [6] Locatello, Francesco, et al. "Challenging common assumptions in the unsupervised learning of disentangled representations.", 2019.
> > > >
> > > > [7] Wendong et al. "Causal component analysis.", 2024.
> > > >
> > > > [8] Pawlowski, et al. "Deep structural causal models for tractable counterfactual inference.", 2020.
> > > >
> > > > [9] Zhang et al. "Causal representation learning from multiple distributions: A general setting.", 2024.
> > > >
> > > > [10] Xia, et al. "The causal-neural connection: Expressiveness, learnability, and inference.", 2021.

---

### Official Review · Reviewer_J7y2 · 2024-07-13

**Soundness:** 3
**Presentation:** 2
**Contribution:** 2
**Rating:** 5
**Confidence:** 4

**Summary:**

This paper extends the previous work on learning disentangled representation from heterogeneous observational data with known hidden structures (e.g., Causal Component Analysis). Unlike the previous work assuming Markov conditions, the proposed theory can deal with non-Markovian causal systems. The identifiability has been shown up to a causal disentanglement map.

**Strengths:**

1. The related background and many examples provided in the appendix significantly enhance the paper's readability for a general audience. This comprehensive background information ensures that even readers who may not be familiar with the specific domain can grasp the fundamental concepts and context of the research.

2. The theoretical results are articulated with commendable mathematical rigor.

3. The FAQ section in the appendix is particularly helpful in addressing potential confusion. It preempts some questions and concerns, providing clear and concise explanations.

**Weaknesses:**

1. I'm not completely sure about the significance of the theoretical results, especially compared to previous work. Since the hidden causal structure is already provided, the problem is easier than most existing works in causal representation learning. Compared to nonlinear ICA, the final indeterminacy is a strict superset of the common point-wise indeterminacy. Therefore, the most interesting contribution should be considered alongside the most relevant work, i.e., causal component analysis, where both assume the hidden structure is known and aim to recover the distribution. Compared to causal component analysis, the extension from Markovian to non-Markovian settings is introduced. However, given that the hidden structure is already known (we can precisely locate those hidden confounders that make the system non-Markovian), this extension seems somewhat compromised. Although the paper extensively discusses its connection to previous works in the appendix (e.g., FAQ), I still feel that the unique contribution might not be significant enough.

2. The combination of assumptions 1 and 2 seems unrealistic. Since the considered setting is non-Markovian, the system allows the existence of unobserved confounders, suggesting that the number of latent variables is likely larger than the number of observed variables. Therefore, the invertibility assumption on the generating process, if all latent variables are considered, is generally not satisfied. To address this, assumption 2 has been introduced to ensure that all unobserved confounders are not part of the mixing procedure, making invertibility possible. However, this simplifies the problem significantly and might not hold in most real-world scenarios.

3. The organization of the paper could be improved. Currently, the most important part of the theoretical discussion, i.e., the motivation and interpretation of all assumptions needed for identifiability, is delayed to the appendix. As a result, it could be difficult for readers to clearly understand the technical contributions, especially given that the topic has been extensively studied in recent works.

(Minor) The current version of the main paper looks overly compact. For instance, the distance before and after section titles is too small. Please consider adjusting this, as it might be unfair given the hard page limits.

**Questions:**

1. In the FAQ part, it is mentioned that domains and interventions are distinct. I understand that they are quite different for hard interventions but I didn't fully get the difference when comparing domains and soft interventions, especially when the interventions are unknown and applied on multiple nodes.

2. In Table 1, it is stated that the proposed method could deal with general distributions, while previous works focus either on one intervention per node or multiple distributions. However, it seems that the identifiability result in this paper also needs changing distributions (sufficient changes) and soft interventions, as mentioned in the appendix. Given that soft interventions are a specific type of distributional change, I'm not sure whether the proposed theory actually generalizes the distributional assumption, especially when compared with previous work on causal representation learning with multiple distributions.

3. Are assumptions required in this paper strictly weaker than those required in causal component analysis? If so, are there any trade-offs?

**Limitations:**

The limitations have been explicitly discussed.

---

> ### Author Rebuttal · Authors · 2024-08-07
>
> We thank the reviewer for the feedback. We feel that a few misreadings of our work make the evaluation overly harsh, and hope you can reconsider the paper based on the clarifications provided below.
> ## W#1
> > W#1 (cont) “I'm not completely sure about the significance of the theoretical results, especially compared to previous work. Since the hidden causal structure is already provided, the problem is easier than most existing works in causal representation learning.”
>
> We respectfully disagree on this point. The latent causal structure is assumed in many prior causal inference literature and CRL works. Compared to existing works, our paper is not easier since we consider a more general setting. Please see **section II in the author's rebuttal** for further discussion about assuming the graph.
>
>
> > W#1 (cont) “Compared to causal component analysis [1], the extension from Markovian to non-Markovian settings is introduced.”
>
> Thanks for highlighting the Markovianity extension! However, our work extends causal component analysis across all three axes we mentioned in the global review: input-assumptions, input-data, and output-disentanglement goals, not just in terms of non-Markovianity.
>
> While we generalize to the non-Markovian setting, we also improve results within the Markovian context. The CRID algorithm encompasses all disentanglement results from [1]. For instance, in the Markovian diagram $V_1 \rightarrow V_2 \rightarrow V_3$, [1] requires one hard intervention per node plus observational data for full disentanglement, ensuring $V_i$ is disentangled from $V_j$ for all $i \neq j$. [1] shows ancestral disentanglement with soft interventions per node plus observational data, achieving disentanglement as: $V_1$ from $V_2, V_3$, and $V_2$ from $V_3$. In contrast, Appendix F.5 (pg 37) shows how we can disentangle a $V_3$ from its ancestors using only two soft interventions on $V_3$.
>
> > W#1 (cont) “However, given that the hidden structure is already known (we can precisely locate those hidden confounders that make the system non-Markovian), this extension seems somewhat compromised.”
>
> That’s a misconception, to see why the non-Markovian setting is reasonable and valuable, refer to the **author’s rebuttal Section IV**.
>
> ## W#2
> > W#2 (and W#1 cont.) “The combination of assumptions 1 and 2 seems unrealistic ...”
>
> Please see why the non-Markovain setting in our work is reasonable and valuable in **author’s rebuttal Section IV**.
>
> ## W#3
>
> > W#3 “The organization of the paper could be improved …”
>
> Thank you for pointing this out. We added a paragraph in Section 2 to introduce the assumptions and their conceptual ideas informally early on.
>
> ## Q#1-3
>
> > Q#1 “In the FAQ part, it is mentioned that domains and interventions are distinct.”
>
> We added a response to this issue in **Section II. of author rebuttal**. In addition, we provide an example illustrating the differences and possibly the issues that may arise in conflating the two.
>
> > Q#2  “In Table 1, it is stated that the proposed method could deal with general distributions, while previous works focus either on one intervention per node or multiple distributions. ”
>
> We want to clarify what we discuss in the manuscript: the current literature focuses on distributions from interventions per node or observational distributions from multiple domains. We handle the general case since our distributions come from arbitrary combinations of interventions and domains.
>
> > “However, it seems that the identifiability result in this paper also needs changing distributions (sufficient changes) and soft interventions, …, especially when compared with previous work on causal representation learning with multiple distributions.”
>
> Thank you for pointing out we leverage different distributions for disentanglement. However, requiring different distributions does not mean we do not generalize the input distributions. In fact, the challenge here is to determine how distributions change given arbitrary combinations of intervention targets and domains encoded in the selection diagram. Under Markovianity, determining how distributions change given soft interventions per node is relatively easier and less complex compared to interventions in different domains under non-Markobanity settings. This challenge is answered by Prop 2. In addition, our setting does not require observational data necessarily either. Finally, note that existing work explicitly specifies the required distributions beforehand, whereas CRID accepts as input whatever distributions a user has, and determines the identifiability of a learned representation. Taken together, the proposed work generalizes the input distributions assumed compared to prior work.
>
> > Q#3 “Are assumptions strictly weaker than those required in causal component analysis (CCA)?”
>
> First, expanding on our earlier point in Q2, CCA requires an intervention per node, whereas our work does not impose such distribution requirements for disentanglement. We offer a more general approach with the causal disentanglement map (CDM) and provide additional graphical criteria (Props. 3, 4, 5) for identifiability, even in cases where CCA may not apply. Second, our non-Markovanity cases are strictly weaker than Markovanity in CCA.
>
> In addition, our theory does not specify exact conditions for disentanglement because the goal is user-defined. Instead, we use graphical criteria to link assumptions in the latent causal graph with data distributions to determine expected disentanglement.
>
> > Q#3 (cont.) “If so, are there any trade-offs?”
>
> There is indeed a tradeoff in terms of how much disentanglement can be achieved, which we highlight in L65-82. Specifically, the more diverse distributions are given, the more disentanglement can be generally achieved. However, if they only care about disentanglement among a subset of variables, they may be able to design more targeted experiments, or data collection procedures to achieve a subset of full disentanglement.

---

> > ### Author Response · Authors · 2024-08-12
> >
> > Dear Reviewer J7y2,
> >
> > Thank you again for spending the time to read our paper, and providing feedback.
> >
> > We believe we have addressed your concerns raised in the review, but since the discussion period is ending soon, we would like to double check if we could provide any further clarification or help to provide other comments about our work.
> >
> > Authors of the paper #13821

---

> > > ### Comment · Area_Chair_Y6Z2 · 2024-08-13
> > >
> > > Dear reviewer,
> > >
> > > As the author-reviewer discussion period is coming toward an end on Aug 13, it would be helpful to have you read and acknowledge the author rebuttal, and/or update your review if your opinion of the paper has changed. Thank you!
> > >
> > > Best,
> > >
> > > Area chair

---

> > ### Comment · Reviewer_J7y2 · 2024-08-13
> >
> > Thanks a lot for the response. I have a few follow-up questions:
> >
> > 1. Does this imply that CCA addresses a special case of the problem in the paper under stronger assumptions?
> >
> > 2. Can we view the intervention as a specific form of domain change? If so, I am still not entirely convinced of the need to treat them separately.
> >
> > 3. Additionally, I couldn't find the updated part concerning W3. Did I overlook it?

---

> ### Author Response · Authors · 2024-08-13
> **Response to 1. and 2. (1/2)**
>
> Thank you for your response and follow-up questions.
>
> > 1. Does this imply that CCA addresses a special case of the problem in the paper under stronger assumptions?
>
> Yes, our problem setting is strictly more general in terms of the assumptions and input data (as discussed in Q3 of the rebuttal). Even though CCA is also nonparametric, it focuses on the Markovian setting, which is a special case of the non-Markovian setting we study in our paper.  Also, the results in CCA rely on input data that contains interventional data for every node. On the other hand, our result accepts arbitrary combinations of distributions from heterogeneous domains, where not all nodes need to be intervened on. Thus, the required input distributions in CCA are a special case of the proposed work.
>
> **Summary**: The disentanglement results in CCA can be derived using CRID. For further discussion on how CCA and our work relates, please refer to L1256-1290 (p. 39) in the submitted manuscript.
>
> > 2. Can we view the intervention as a specific form of domain change? If so, I am still not entirely convinced of the need to treat them separately.
>
> The answer is no, not in generality.
>
> Section III of the [author rebuttal](https://openreview.net/forum?id=uLGyoBn7hm&noteId=W6rCNuQqw3) discusses this issue. As another example, consider a modification of the Ex., where one treats a domain as an intervention. Consider the causal chain $S \rightarrow V1 \rightarrow V2 \rightarrow V3$. Assume a hypothetical setup, where one has observations in humans ($P^{\Pi_1}(X)$), and observations in bonobos ($P^{\Pi_2}(X)$), and two soft interventions on hair color in bonobos ($P^{\Pi_2}(X;\sigma_{V_3}), P^{\Pi_2}(X;\sigma_{V_3})$). Leveraging CRID, one can disentangle hair color ($V_3$) from sun exposure ($V_1$).
>
> If we ignore the domain index and consider any bonobo distribution as an intervention w.r.t. the observational distribution in humans, then the available distributions would be the following: $P(X), P(X;\sigma_{V_1}), P(X;\sigma_{V_1, V_3}), P(X;\sigma_{V_1, V_3})$. Interestingly, the original result that $V_3$ is ID w.r.t. $V_1$ cannot be obtained in this case since $V_1, V_2, V_3$ are all in \deltaQ set (Def. 3.1) ($P(V_1)$ and $P(V_3 \mid V_2)$ can both change) when comparing $P(X;\sigma_{V_1, V_3})$ with any other distribution. The lesson here is that it would be naive to ignore that distributions within the same domain contain invariances that can be leveraged.
>
> In addition, further discussion about this matter is provided in Q5 (FAQ) in p. 47 of the submitted manuscript. After all, the domain index is not only useful mathematically, but it formally encodes important information about the world. Finally, the distinction of interventions and domains is also useful and has far-reaching consequences in the context of causal discovery (structure learning), as noted in [1]. We would be happy to provide further clarifications.
>
> [1] Li, Adam, Amin Jaber, and Elias Bareinboim. "Causal discovery from observational and interventional data across multiple environments." Advances in Neural Information Processing Systems 36 (2023).

---

> ### Author Response · Authors · 2024-08-13
> **Response to 3. (2/2)**
>
> > 3. Additionally, I couldn't find the updated part concerning W3. Did I overlook it?
>
>  Since we could not upload the new manuscript, we copy the text here, which was added at the end of Section 2.
>
> “**Assumptions (Informal) and Modeling Concepts**
>
> Before discussing the main theoretical contributions, we informally state our assumptions and provide remarks to ground the ASCM model.
>
> **Assumptions**
> 1. Soft interventions without altering the causal structure
>
> Hard interventions cut all incoming parent edges, and soft interventions preserve them. However, more general interventions may arbitrarily change the parent set for any given node. We do not consider such interventions and leave this general case for future work.
>
> 2. Known-target interventions
>
> All interventions occur with known targets, reducing permutation indeterminacy for intervened variables.
>
> 3. Sufficiently different distributions
> Each pair of distributions $P^{(j)}, P^{(k)} \in \mathcal{P}$ are sufficiently different, unless stated otherwise. This is naturally satisfied if ASCMs and interventions are randomly chosen. Similar assumptions include the "genericity", "interventional discrepancy", and "sufficient changes" assumptions.
>
> **Remarks**
> 1. Mixing is invertible
>
> As a consequence of Def. 2.1, the mixing function $f_X$​ is invertible since it is defined as a diffeomorphism, ensuring that latent variables are uniquely learnable.
>
> 2. Confounders are not part of the mixing function
>
> According to Def. 2.1, latent exogenous variables U influence the high-dimensional mixture X only through latent causal variables V, so unobserved confounding does not directly affect the mixing function. For instance, in EEG data, sleep quality and drug treatment may influence EEG appearance, while socioeconomic status may confound sleep and drug treatment but not directly affect EEG. This idea is also present in prior work, such as nonlinear ICA, where independent exogenous variables $U_i$​ each point to a single $V_i$.
>
> 3. Mixing function is shared across all domains
>
> According to Def. 2.1, the mixing function $f_X$​ is the same for all ASCMs $M_i \in M$, enabling cross-domain analysis. If the mixing function varied across distributions, the latent representations would not be identifiable from iid data alone.
>
> 4. Shared causal structure
>
> As a consequence of Def. 2.1, each environment's ASCM shares the same latent causal graph, with no structural changes among latent variables. The assumption that there are no structural changes between domains can be relaxed and is considered in the context of inference when causal variables are fully observed, as discussed in (Bareinboim 2011). This is an interesting topic for future explorations, and we do not consider this avenue here.“

---

> > ### Comment · Reviewer_J7y2 · 2024-08-14
> >
> > Thanks for your further clarification and the updated part. I have raised my score.

---

> > > ### Author Response · Authors · 2024-08-14
> > >
> > > Thank you for the opportunity to clarify these issues and elaborate on our work. We will review this thread again and use it to improve the manuscript.

---

### Official Review · Reviewer_XxiF · 2024-07-30

**Soundness:** 4
**Presentation:** 2
**Contribution:** 3
**Rating:** 6
**Confidence:** 3

**Summary:**

This study considers the problem of causal representation learning in the presence of confounding, a direction that wasn’t previously explored. This translates to finding the unmixing function and disentangling a set of target variables from another given the knowledge of the latent causal graph and an input set of distributions in a setting which no longer is Markovian. More generally, this work addresses the question of whether a set of variables can be disentangled from another, given the knowledge of causal structure and a set of interventional distributions.

The authors provide a graphical criteria for that purpose accompanied by an algorithm that uses those criteria to identify which subsets are identifiable from another when given a set of intervention targets and a selection diagram. The graphical criteria are backed by theoretical guarantees and proofs, and simple experiments show how the model performs as a proof of concept. As confounding is a setting unexplored in prior work, there are no baselines in the experiments.

**Strengths:**

- The non-Markovian setting is a novel direction and has the potential to be relevant and important in bridging the gap between CRL and realistic applications to AI, however, the considered setting relies on the knowledge of the latent causal structure.
- The CRID algorithm is an interesting, general, and novel algorithm that enables us to understand the possibility of identifying any subset of variables from another, given intervention targets (though it requires access to the latent structure).

**Weaknesses:**

- The contributions seem a bit overstated, in particular, ii and iii in the abstract which consider arbitrary distributions from multiple domains, and a relaxed version of disentanglement. If I understand correctly, such distributions and relaxations of disentanglement are present in a decent recent body of work; Lachapelle et al (https://arxiv.org/pdf/2401.04890) and Multi-node interventional works (https://arxiv.org/pdf/2311.12267, https://proceedings.mlr.press/v236/bing24a, https://arxiv.org/abs/2406.05937), just to name a few, there are more references that I encourage the authors to have a look at and provide some comment as to the differences. The non-Markovianity aspect is indeed a contribution but the relaxations of disentanglement seem less novel (unless the authors could clarify any significant distinctions). The same issue w.r.t. overstatement applies to lines 63-64. Again in lines 172-197 the setup bears resemblance to the aforementioned works. It would be great if the authors could clarify the distinctions. The contributions are relaxed in lines 86-93, getting closer to what actually has been achieved by this work, which are different from the abstract and other parts of the manuscript.

- Although the setting considered is novel, I could not be convinced about the relevance or importance of this setup in the more realistic applications of CRL, mainly due to two reasons: i) Knowledge of the causal structure is quite a strong assumption that carries over to not many tasks, ii) if I understand correctly, the applicability of the proposed method relies somewhat on the availability of “do” interventions which themselves are not always easily obtainable, adding to the concern w.r.t. The relevance in the context of applying CRL more widely.

- I think the authors tried to convey some outlook on the relevance by describing the level of granularity and the causal variables, but that direction was cut short, and the experiments did not seem to add/support that storyline.

Remark: I am happy to change my score based on the discussion; the current score reflects that I have edged on the side of caution regarding the contributions and the relevance/applicability of the proposed method.

**Questions:**

Please see the weaknesses as well.

Also:
- It would be nice if the example in lines 24-25 could be expanded/elaborated on.
- Line 120 typo: modeling disentangled representation learning.
- “do” interventions: How much does the method rely on “do” interventions? We know it’s not easy and sometimes possible to obtain those in practice. How does this method go beyond such scenarios if “do” interventions are not available?

**Limitations:**

Please see the weaknesses as well.

- Also, what are the use cases of the proposed method under such assumptions? The EEG example is fine where the latent structure is known, but what is the broader implication of this method when that is not the case, and “do” interventions are not cheap either?
- Quoting the authors: “This brings us one step closer to building robust AI that can causally reason over high-level concepts when only given low-level data, such as images, video, or text.” Again, I could not be convinced that this work closes a large gap in the literature and the experiments are not completely in line with the quoted claim.

---

> ### Author Rebuttal · Authors · 2024-08-07
>
> > W#1 “The contributions seem a bit overstated…”
>
> Thank you for pointing out additional relevant works. Broadly, please refer to **Section I in the author’s rebuttal** and **Fig.1 in the pdf** for a global view of this setting. Specifically, we have added a comparison of these papers [1-4] in the Appendix and discuss them here.
> Input-Assumptions: Our paper operates in a nonparametric, non-Markovian setting. Among [1-4], only [1] considers a nonparametric approach but assumes sparsity in the graph and mechanism, unlike our work. All papers assume Markovianity.
> Input-Data: We accept various combinations of distributions (observational, soft, or hard interventions) across domains. [2] uses observational data across domains, [3] focuses on hard interventions in one domain, [4] considers interventions in one domain, and [1] includes interventions in one domain (and time series, which we do not discuss).
>
> Output: We target partial disentanglement, whereas [2-4] aim for full or structured disentanglement (e.g., up to ancestors). Our approach encompasses these as special cases, with notable differences from [1] as described above.
>
> > W#1 (cont.) “lines 63-64, 172-197”
>
> We reworded the sentences:
>
> L63: “In terms of the expected output, rather than aiming for a full disentanglement, or structured disentanglement (such as from ancestors, or surrounding nodes), we consider a more relaxed type of disentanglement, such as in Ex. 6”.
>
> L172-197 to compare to prior work: “Recent work has also considered a similar goal of generalized disentanglement  [1]. Our work still differs from theirs in the following ways: i) [assumptions] we model a completely nonparametric non-Markovian ASCM, whereas [1] assumes sparsity and a Markovian ASCM … (see Appendix F for more details).”
>
> [1] https://arxiv.org/pdf/2401.04890
>
> [2] https://arxiv.org/pdf/2311.12267
>
> [3] https://proceedings.mlr.press/v236/bing24a
>
> [4] https://arxiv.org/abs/2406.05937
>
> > W#2 and L#1 “i) Knowledge of the causal structure…”
>
> We recognize causal discovery/structure learning is an important problem within CRL and causal inference more broadly. Please check **the author's rebuttal Section II** for why we assume the causal graph and the justification about this assumption. To summarize, the causal diagram is a common assumption made in both causal inference and CRL literature since its inception. Causal inference as we study in AI arguably started in its modern form with the work of Judea Pearl in Biometrika 1995 where he introduced the causal graph assumption and developed the do-calculus, which highlights the tradeoff between assumptions and the types of conclusions one can draw.  Also note that achieving identifiability given the diagram, as discussed in this work, is essential for learning the diagram. We note that inferential and learning components already appear together in the literature, albeit not yet in the context of representations, such as [1-4]. Adding a representation layer on top of these works is certainly a challenging but very compelling research direction.
>
>
> [1] Zhang. "Causal Reasoning with Ancestral Graphs" (2008).
>
>
> [2] Jaber et al. “Causal Identification under Markov equivalence: Calculus, Algorithm, and Completeness” (2022).
>
>
> [3] Jaber et al. Identification of Conditional Causal Effects under Markov Equivalence (2019).
>
>
> [4] Jaber et al. Causal Identification under Markov Equivalence: Completeness Results. (2019).
>
> > W#2 (cont.), Q#3 and L#1 “ii) …relies somewhat on the availability of “do” interventions…”
>
> Thanks for mentioning this point and giving us the opportunity to clarify. To be clear, “do” interventions are not required by our work. In the text, “arbitrary distributions” implies that neither hard nor soft are necessarily required. Formally, `do(T) = {}` means there is no hard intervention. We have reworded L163-164 to be more clear: “When $\mathbf{I}^{(k)} = \{V_i^{\Pi^{(k)}, \{b\}}\}$, where $t$ is omitted, then the intervention is assumed to be soft. Thus when $do[I^{(k)}] = do[\{\}]$, it implies there are no variables with hard interventions in $\sigma^{(k)}$.”
>
> In summary, CRID accepts as input any arbitrary combination of “do” and “soft” interventions, including “soft” interventions alone. For concreteness, Ex. 4 in the paper provides an example precisely that does not rely on any hard intervention.
>
>
> > W#3 “I think the authors tried to convey some outlook...”
>
> The granularity discussion in the Introduction is to motivate why learning disentangled representations is important to the broader puzzle. The goal of this paper is foundational, and we show a more general setting where it is possible to learn disentangled representations. The simulation experiments corroborate the theory in settings that previously were thought that disentanglement could not be achieved. We also added a new image experiments shown in **author's rebuttal V**.
>
> > Q#1: It would be nice if the example in lines 24-25 could be expanded/elaborated on.
>
> We include in the introduction the following elaboration:
>
> "For example, images may capture a natural scene, where the causal variables are the objects within the scene while the pixels themselves are not the causal variables. Thus AI must disentangle the representations of latent causal variables given the pixels to capture correct causal effects for downstream tasks. Similarly, given written text describing this natural scene, the characters in the text are not the causal variables, but rather a low-level mixture of the underlying causal variables."
>
> Also please refer to a motivating example in Appendix D.1 (pg 32).
>
> > Q#3
>
> See response to W2(ii).
>
> > L#1
>
> See W#1 and W#2 responses.
>
> > L#2 “This brings us one step closer to building robust AI…”
>
> When we say “one step closer”, it does not mean we claim to solve the problem of “robust AI”, but rather the disentanglement results in the paper are a critical ingredient towards developing AI with causal awareness. We will reword the sentence.

---

> > ### Comment · Reviewer_XxiF · 2024-08-09
> >
> > Thank you for your efforts and the detailed rebuttals, and for adding the experiments on colouredMNIST.
> >
> > The authors' rebuttal addressed most of my concerns, and thus I raise my score to 6. The reason for not a higher score is a concern shared with the rest of the reviewers on the presentation which is often quite dense and lacks discussions and explanations throughout the technical details to give a clearer idea of the implications of the contribution and theoretical results as well as with the organization (not respecting the formatting limits). However, I find the technical contributions solid and interesting and hope the authors could revisit their presentation for the final version of their work.

---

> > > ### Author Response · Authors · 2024-08-09
> > >
> > > We appreciate your time and efforts in reading and providing feedback. We are thankful that you found our contribution and results solid and interesting. We understand we have many results (as noted by ZG4p), partially due to the fact that we are considering such a general setting. We will reformat the paper to introduce i) Figure 1 from the pdf rebuttal, which helps visually the various axes of identifiability in causal representation learning, and ii) add additional examples and discussion of assumptions using the extra page.

---

### Author Rebuttal · Authors · 2024-08-07

We thank the reviewers for providing valuable feedback. We address common questions here.

## I. Novelty and value? (XxiF, j7y2, fxDa)

As illustrated in Table 1 in the manuscript (L50-64) and Fig. 1 in pdf rebuttal, the literature can be interpreted through the following axes: (1) input assumptions; (2) input data; and (3) disentanglement goal. Though some prior works may be similar in one dimension, our submitted work considers and advances all three axes simultaneously.

**Input-Assumptions**: Prior work's additional assumptions (e.g., sparsity, Markovianity, linearity, polynomial) do not cover all situations. For instance, assuming Markovianity is restrictive due to common unobserved confounding in many realistic settings. Similarly, assuming linearity is unrealistic for high-dimensional data (e.g., images or text).

**Input-Data**: Prior work assumes data from single-domain interventions or multi-domain observations and often requires an intervention per node. Our work accepts any combination of interventional data from multiple domains (see III. about domains vs interventions).

**Output-Disentanglement Goal**: Prior work often focuses on conditions for full disentanglement. We aim for partial disentanglement, as in recent work [9] (thanks XxiF for sharing). This form includes full disentanglement and is more practical, as full disentanglement requires stronger assumptions and **more** targeted interventions or domain changes.

## II. Assuming the causal graph? (XxiF, j7y2, fxDa)

First, causal diagrams are common in causal inference tasks like identification, estimation, and transportability [1-2]. They may be obtained in realistic settings from human or expert knowledge (e.g., EEG example in Fig. 1, pg 3; Face example in Appendix D.1, pg 31).

Contextualizing our work, note there are two sub-tasks in CRL: (a) learning a representation of latent variables and (b) learning the diagram [3]. To address (a), many prior works assume more or less explicitly the diagram [3-7]. In addition, many works addressing (b) typically demonstrate as a first step that disentanglement among the latent variable representations is achievable given the diagram [8-10]. However, we also recognize that learning the diagram is a valuable subtask, and added to the conclusion: “Considering disentangled representation learning where the latent causal graph is unknown, or only partially known is interesting future work.”

[1] Pearl. Causality: Models, Reasoning, and Inference. 2000.

[2] Pearl. "Probabilities of causation: three counterfactual interpretations and their identification." 2022.

[3] Schölkopf et al. "Toward causal representation learning." 2021.

[4] Hyvarinen et al. "Nonlinear ICA using auxiliary variables and generalized contrastive learning." 2019.

[5] Locatello et al. “weakly-supervised disentanglement without compromises.” 2020.

[6] Shen et al. "Weakly supervised disentangled generative causal representation learning." 2022.

[7] Von Kügelgen et al. "Self-supervised learning with data augmentations provably isolates content from style." 2021.

[8] Lachapelle et al. "Nonparametric partial disentanglement via mechanism sparsity…" 2024.

[9] Squires et al. "Linear causal disentanglement via interventions." 2023.

[10] Zhang et al. "Causal representation learning from multiple distributions: A general setting." 2024.


## III. Why distinguish domains and interventions? (XxiF, j7y2)

[1] formalized multiple domains, introducing S-nodes (environments) for a combined representation of different environments. Conflating interventions and domains when comparing distributions is generally invalid [2-3]. See Appendix FAQ Q5 (pg 47).

_Ex. Comparing interventions without considering the domain is unsound:_
Consider an ASCM for bonobos ($\Pi^1$) and humans ($\Pi^2$) with the causal chain $V_1 \rightarrow V_2 \rightarrow V_3 \leftarrow S^{1,2}$, where $V_1$ is sun exposure, $V_2$ is age, and $V_3$ is hair color. Sun exposure affects aging, which in turn affects hair color, varying by species ($S$). We collect face images, $X = f_X(V_1, V_2, V_3)$, in two settings with altered sun exposure levels, ${V_1}^{\Pi_1}$ and ${V_1}^{\Pi_2}$. Ignoring the domain and treating these distributions as purely interventional would wrongly suggest that $V_1$ can be disentangled from $V_3$, which is incorrect given the input distributions and causal graph.

[1] Pearl. “Transportability of causal and statistical relations...” 2011

[2] Bareinboim et al. "Causal inference and the data-fusion problem." (2016)

[3] Li et al. “Characterizing and learning multi-domain causal structures ...” (2024)

## IV. Why non-Markovanian setting is valuable ? (j7y2, fxDa)

Unobserved confounders $U$, shown as bidirected edges, represent spurious associations among latent factors $V$ and are a major concern throughout the literature. In Markovian settings, exogenous factors $U_i$ are assumed independent for each endogenous factor $V_i$, and do not mix into $X$. For example, in $U_1\rightarrow V_1 \rightarrow X\leftarrow V_2\leftarrow U_2$, $U_1, U_2$ are not part of V, and do not affect X.

In non-Markovian settings, unobserved confounders $U$ are also not part of $V$ and do not mix into $X$, differing from endogenous confounders among $V1$ and $V2$ (e.g., $V1 \leftarrow V3 \to V2$, where ${V1, V2, V3} \to X$). In fact, Markovianity is stronger since it assumes independence, so non-Markovianity is a relaxation over the stringent requirement that the exogenous variations of each variable is independent. Technically speaking, one usually does not assume non-Markovianity, but relaxes Markovianity.  See the EEG ex. (Fig. 1, pg 3), Face ex. (Appendix D.1, pg 31), and Appendix FAQ Q6 (pg 48).

## V. Experiment on Images. (all reviewers)

We have added experiments on images verifying disentanglement we expect to see in Fig. 2-4 of pdf rebuttal. Further discussion is provided [here](https://openreview.net/forum?id=uLGyoBn7hm&noteId=9kDz0N5kNR)

---

### Decision · Program_Chairs · 2024-09-25

**Decision:**

Accept (poster)

**Comment:**

The paper addresses causal disentangled representation learning in non-Markovian causal systems, which include unobserved confounding and arbitrary distributions from various domains.

Pros:

+ Addresses an important problem in causal representation learning by extending to non-Markovian systems and partial disentanglement

+ Provides rigorous theoretical analysis with supporting examples

+ Introduces a more general definition of disentanglement, potentially applicable to a wider range of scenarios

Cons:

+ Limited empirical validation, with experiments only on synthetic datasets and simple causal graphs

+ Requires strong assumptions, including knowledge of the latent causal structure and access to multiple domains with interventions

+ Gap between theoretical contributions and practical applications